# A statistical framework for differential pseudotime analysis with multiple single-cell RNA-seq samples

Wenpin Hou [1,2], Zhicheng Ji[1,3], Zeyu Chen[4,5,6,7], E. John Wherry[4,5,6], Stephanie C. Hicks [1] ✉ & Hongkai Ji [1] ✉

Pseudotime analysis with single-cell RNA-sequencing (scRNA-seq) data has been widely used to study dynamic gene regulatory programs along continuous biological processes. While many methods have been developed to infer the pseudotemporal trajectories of cells within a biological sample, it remains a challenge to compare pseudotemporal patterns with multiple samples (or replicates) across different experimental conditions. Here, we introduce Lamian, a comprehensive and statistically-rigorous computational framework for differential multi-sample pseudotime analysis. Lamian can be used to identify changes in a biological process associated with sample covariates, such as different biological conditions while adjusting for batch effects, and to detect changes in gene expression, cell density, and topology of a pseudotemporal trajectory. Unlike existing methods that ignore sample variability, Lamian draws statistical inference after accounting for cross-sample variability and hence substantially reduces sample-specific false discoveries that are not generalizable to new samples. Using both real scRNA-seq and simulation data, including an analysis of differential immune response programs between COVID-19 patients with different disease severity levels, we demonstrate the advantages of Lamian in decoding cellular gene expression programs in continuous biological processes.

Single-cell RNA-sequencing (scRNA-seq) enables the dissection of complex cellular programs at single-cell resolution in biological samples with heterogeneous cell compositions. When cells in a sample come from a continuous biological process, computationally placing the cells along a pseudotemporal trajectory based on their progressively changing transcriptomes is a powerful approach to reconstructing the dynamic gene expression programs of the underlying biological process. This approach, also known as pseudotime analysis[1–3], is now widely used to study cell differentiation[4–6], immune responses[7,8], disease development[9–12], and many other biological systems with temporal dynamics. A systematic review and comparison of these methods can be found in a recent benchmark study[3]. The majority of existing methods were designed to infer gene expression changes along the reconstructed trajectory within one biological sample. However, scRNA-seq experiments today standardly generate data with multiple biological samples across multiple conditions. For

[1]Department of Biostatistics, The Johns Hopkins Bloomberg School of Public Health, Baltimore, MD 21205, USA. [2]Department of Biostatistics, Mailman School of Public Health, Columbia University, New York, NY 10032, USA. [3]Department of Biostatistics and Bioinformatics, Duke University School of Medicine, Durham, NC 27710, USA. [4]Department of Systems Pharmacology and Translational Therapeutics, Perelman School of Medicine, University of Pennsylvania, Philadelphia, PA 19104, USA. [5]Institute for Immunology, Perelman School of Medicine, University of Pennsylvania, Philadelphia, PA 19104, USA. [6]Parker Institute for Cancer Immunotherapy at University of Pennsylvania, Philadelphia, PA 19104, USA. [7]Department of Cancer Biology, Dana-Farber Cancer Institute, Boston, MA 02215, USA. ✉e-mail: shicks19@jhu.edu; hji@jhu.edu

example, a number of COVID-19 studies generated scRNA-seq data from multiple patients with differential disease severity levels[13–19]. Therefore, there is an increasing demand for methods that can simultaneously (i) take into account sample-to-sample variation and (ii) identify changes in pseudotemporal trajectories across conditions. To meet this demand, two challenges need to be addressed.

First, changes in pseudotemporal trajectories across conditions can occur in multiple ways, including (i) topological differences, such as a cell lineage along differentiation is lost (or added) in one sample group compared to another group, (ii) changes in the proportion (or density or abundance) of cells along a cell lineage across conditions, and (iii) changes in the gene expression itself along pseudotime across conditions. An ideal solution would address all three types of changes in one comprehensive statistical framework.

Second, in order to separate changes of biological interest (e.g. difference between treatment and control) from other biological or technical noises, it is important to account for naturally occurring sample-level variations, not of interest (e.g. sample-to-sample variation within the treatment or control group), unwanted technical variations (e.g. batch effects), and other uncertainties in the analysis (e.g. uncertainties of the inferred trajectory and pseudotime).

However, there currently does not exist a comprehensive integrative framework that identifies all three types of changes in pseudotemporal trajectories (topology, cell density, and gene expression) across experimental conditions with multiple samples per condition, while also accounting for sample-level variability.

Although there exist pseudotime analysis methods to detect changes in gene expression along pseudotime (e.g. Monocle[20–22], TSCAN[23], Slingshot[24]), in cell abundance along pseudotime (e.g. milo[25], DAseq[26]), and in trajectory lineages (e.g. tradeSeq[27]), most methods do not investigate changes across conditions. Almost all methods ignore sample-to-sample variation by either only analyzing cells from a single sample or treating cells from multiple samples as if they were from a single sample. For the latter, cells from different samples are usually integrated in a low-dimensional space by removing both biological and technical differences among samples, and a trajectory is then inferred to characterize dynamic cellular programs along pseudotime, without considering variability among samples.

Phenopath[28] and condiments[29] are two pseudotime methods capable of identifying changes across conditions. Condiments assumes that each condition has one sample and therefore does not consider sample-to-sample variation within each condition when each condition has multiple replicates. Ignoring sample-level variability can result in false discoveries not generalizable to new samples. Phenopath assumes gene expression changes linearly along pseudotime and cannot deal with arbitrary differences between conditions which may be non-linear functions of pseudotime. Moreover, it does not estimate sample-level variance separately from cell-level variance. Thus, similar to condiments, one cannot assess whether the observed difference between conditions is real or expected by chance based on the random sample-level variability within each condition. Although properly accounting for the variation across samples is important in multi-sample single-cell data, neither PhenoPath nor condiments can meet this need.

Pseudotime inference itself also has uncertainties. Recently, PseudotimeDE[30] has been proposed to account for pseudotime reconstruction uncertainties in single-sample pseudotime analysis via subsampling cells and permuting pseudotime. However, this approach does not consider multiple samples and therefore does not characterize variability and differences across samples.

To address these gaps, we introduce a comprehensive and integrative statistical framework, referred to as Lamian, for differential multi-sample pseudotime analysis. Lamian is named after a traditional Chinese hand-pulled noodle. The name is chosen based on the similarity between the process of making Lamian noodles and our

statistical model in which multi-sample single-cell data are described using multiple smooth noodle-like functional curves (Fig. S1). Given scRNA-seq data from multiple biological samples with known covariates, such as age, sex, sample type, and disease status, Lamian can be used to (1) construct pseudotemporal trajectories and evaluate the uncertainty of the topologies, (2) evaluate changes in the topological structure associated with sample covariates, (3) describe how gene expression and cell density change along the pseudotime, and (4) characterize how sample covariates modify the pseudotemporal dynamics of gene expression and cell density. Importantly, when identifying gene expression or cell density changes, Lamian accounts for variability across biological samples. As a result, Lamian is able to more appropriately control the false discovery rate (FDR)[31] when analyzing multi-sample data, a property not offered by other existing methods.

## Results

### Lamian: a statistical framework for differential pseudotemporal trajectory analysis in multiple samples

Lamian consists of four modules tackling different aspects of multi-sample pseudotime analysis (Fig. 1). The input for Lamian includes (1) a low-dimensional representation of cells, such as principal components (PCs) or other low-dimensional embeddings of the scRNA-seq data from multiple samples that have been harmonized into a common space using methods such as Seurat[32], Harmony[33] or scVI[34], (2) the normalized scRNA-seq gene expression matrices, and (3) sample-level metadata, such as covariate information corresponding to samples' biological groups, experimental conditions, and batch indicators for batch effect correction. We assume that the data harmonization is done by users and refer readers to a recent benchmark study[35] for guidelines on choosing the harmonization methods. Advantages of Lamian compared to existing methods (Table S1) include comprehensive solutions to evaluating tree topology uncertainty and differential topology and identifying gene expression and cell density changes associated with sample covariates while accounting for sample-level variability.

Module 1 of Lamian uses the harmonized data to construct a pseudotemporal trajectory and then quantifies the uncertainty of tree branches using bootstrap resampling. First, cells from all samples are jointly clustered (Fig. 1a), and the cluster-based minimum spanning tree (cMST) approach described in TSCAN[23] is used to construct a pseudotemporal trajectory. The tree can have multiple branches, allowing one to model multiple lineages of a dynamic process. Next, after users specify a tree node as the start of pseudotime or marker genes that should highly express at the start of pseudotime, Lamian will automatically enumerate all pseudotemporal paths and branches. Then, it evaluates the uncertainty of each branch by quantifying a metric we refer to as the *detection rate*, which is defined as the probability that a tree branch can be detected in repeated bootstrap samplings of cells (Fig. 1b). The advantages of using TSCAN to construct pseudotime include (i) the scalability of its cMST approach to a large number of cells (since the number of tree nodes in the spanning tree is determined by cell cluster number instead of cell number) and repeated bootstrap resamplings, (ii) the flexibility it provides to support both automatic and manual trajectory construction[23], and (iii) its overall competitive performance in multiple previous benchmark evaluations[3,36].

Module 2 of Lamian first identifies variation in tree topology across samples and then assesses if there are differential topological changes associated with sample covariates (Fig. 1b). For each sample, Lamian calculates the proportion of cells in each tree branch, referred to as *branch cell proportion*. Because a zero or low proportion can reflect the absence or depletion of a branch, changes in tree topology can be described using branch cell proportion changes. With multiple samples, Lamian characterizes the cross-sample variation of each

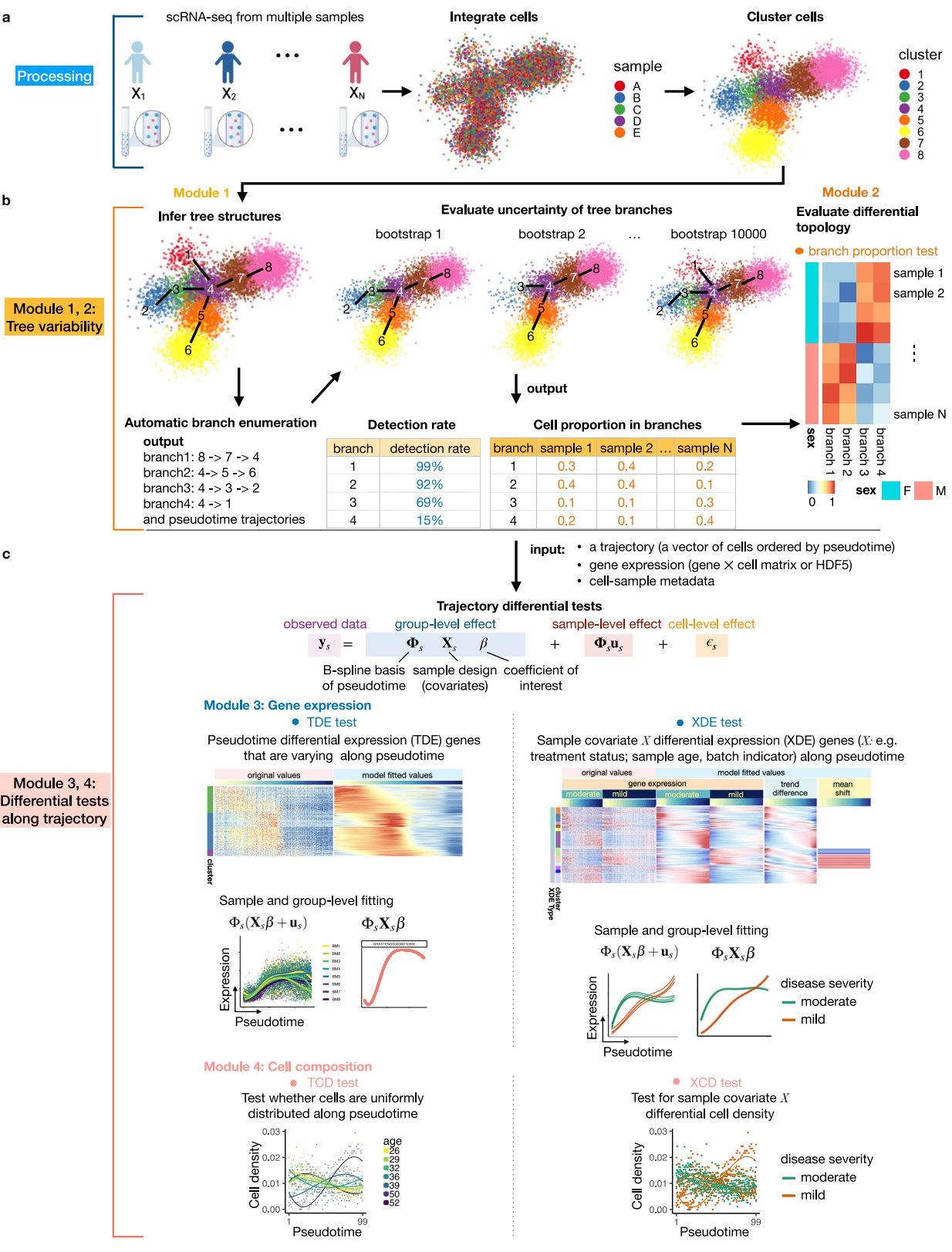

**Figure — Module overview**

branch by estimating the variance of the branch cell proportion across samples. Furthermore, regression models can be fit to test whether the branch cell proportion is associated with sample covariates. To facilitate convenient exploration of each individual branch, one can use a binomial logistic regression to evaluate covariate-associated branch cell proportion changes for each branch separately. Alternatively, users can also use a multinomial logistic regression to analyze

covariate-associated changes of cell proportion ratios between branches by considering all branches jointly (Supplementary Notes). These regression-based methods allow one to identify tree topology changes between different conditions, for example in a case-control cohort, accounting for sample-level variability. They are functions not provided by methods such as `PhenoPath`, `condiments`, and `PseudotimeDE`.

**Fig. 1 | Overview of** `Lamian` **: a statistical framework for differential pseudo-temporal trajectory analysis with multiple samples. a** Using integrated and harmonized scRNA-seq data across multiple samples, `Lamian` first groups cells into clusters. **b** Clustered data is used to infer a pseudotemporal tree structure followed by automatically enumerating all pseudotemporal branches (paths). The uncertainty of tree branches are quantified using a *detection rate* in bootstrap resampling framework (Module 1), followed by quantifying the variability of branches across samples and identifying differences in the branching structure across conditions (Module 2). **c** For each tree branch (pseudotemporal path), `Lamian` can identify two types of differential expression (DE): DE along pseudotime (TDE) and DE associated

with sample covariates (XDE) (Module 3). Similarly, `Lamian` can also identify changes in cell density along pseudotime (TCD) and associated with a sample covariate (XCD) (Module 4). Gene's or cell abundance's pseudotemporal patterns are modeled using the combinations of B-spline bases ($\Phi_s$) to allow non-linear patterns. The combination coefficients are decomposed into effects due to sample covariates ($X_s\beta$, where $X_s$ is the design matrix) and variation among samples with common covariate values ($\mathbf{u}_s$). Cell-level data $\mathbf{y}_s$ in sample $s$ are generated from the sample-level curve by adding cell-level random noise $\epsilon_s$. See Methods and Supplementary Notes for details.

---

Given a path or branch along a pseudotemporal trajectory, the scRNA-seq gene expression matrices from multiple samples, and sample-level covariate information, Module 3 of `Lamian` identifies differentially expressed (DE) genes using a functional mixed effects model (Fig. 1c). There are two types of DE tests. First, the *TDE test* evaluates whether a gene's activity as a function of pseudotime $t$, denoted as $f(t)$, is a constant ($H_0: f(t) = c$), with the goal to identify genes whose activities change along pseudotime ($H_1: f(t) \neq c$). Here, TDE refers to *pseudotime differential expression*. Second, the *XDE test* evaluates for each gene whether the pseudotemporal activity $f(t)$ is associated with a sample-level covariate, such as whether $f(t)$ is different between healthy and disease samples. Here, XDE refers to *covariate X differential expression*. Currently, existing pseudotime methods, such as `Monocle`, `Slingshot` and `TSCAN` only detect TDE, but not XDE. `PhenoPath` and `condiments` may detect XDE but do not account for sample-level variability in multi-sample studies. `Lamian` is an integrative framework to provide both TDE and XDE for multiple sample analyses. For each XDE gene, `Lamian` further evaluates whether the sample covariate shifts the mean of $f(t)$ (referred to as a *mean shift*) or changes the functional form of $f(t)$ (referred to as a *trend difference*) or both. Additionally, unsupervised clustering (*k*-means by default, Louvain and Gaussian mixture model clustering are provided as optional) is applied to DE genes to group and summarize major differential gene patterns. In all DE tests, `Lamian` accounts for sample-to-sample variation directly in its model framework, whereas the other methods do not. Consequently, `Lamian` is able to better control the false discovery rate (FDR)[31] compared to existing methods that ignore sample-to-sample variation which leads to identifying false discoveries that are not generalizable in new samples. By default, `Lamian` uses a permutation approach to determine statistical significance of the DE tests (Lamian.pm). This approach is more reliable but can be computationally slow. For fast computation, `Lamian` also provides an option to determine significance using the chi-squared distribution as the asymptotic null for the likelihood ratio statistics (Lamian.chisq). This option is fast but less accurate. It can be used when users want to run a quick initial analysis while waiting for more rigorous results from Lamian.pm, especially when dealing with a large dataset. Below `Lamian` refers to Lamian.pm unless otherwise specified.

Similar to gene expression, Module 4 of `Lamian` tests whether cells' density along pseudotime is uniformly distributed or not (*TCD test*), and if it is associated with a sample covariate (*XCD test*). This may be used to study dynamic processes, such as cell expansion in immune response or how disease changes the pseudotemporal cell density pattern.

In all differential analyses, unwanted technical variations such as batch effects or other confounding variables can be adjusted by regressing them out in the `Lamian` regression model.

**Lamian estimates tree topology stability and accurately detects differential tree topology**

We begin with illustrating Modules 1 and 2 of `Lamian` using a Human Cell Atlas (HCA)[37,38] 10x Genomics scRNA-seq dataset, referred to as HCA-BM, consisting of bone marrow samples from 8 donors (4 females and 4 males) and a total of 32,819 cells. Bone marrow contains

hematopoietic stem cells (HSCs) differentiating into different blood cell types, creating a natural branching structure. This dataset along with the existing biological knowledge about this system therefore can be used to demonstrate and evaluate Lamian's ability to analyze a trajectory with branches.

First, we construct the pseudotemporal trajectory and assess the tree topology stability (Module 1). Applying `TSCAN` to the `Seurat`-harmonized bone marrow data, we identified 6 cell clusters (Fig. 2a), which form a minimum spanning tree with three branches, corresponding to the three major lineages of HSC differentiation - myeloid, erythroid, and lymphoid (Fig. 2b). We confirmed these lineages with known marker genes (Figs. 2c, S2). Specifically, HSCs are mostly in cluster 5, as indicated by high *CD34* expression (Fig. 2c). By setting cluster 5 as the origin, we obtained three pseudotemporal paths (Fig. 2a: the path of cluster $5 \rightarrow 1$; $5 \rightarrow 6 \rightarrow 2$; $5 \rightarrow 3 \rightarrow 4$). `Lamian` uses repeated bootstrap sampling of cells along the branches to calculate a detection rate. In the HCA-BM data, these three branches can be detected in 93.8% ($5 \rightarrow 1$), 95.3% ($5 \rightarrow 6 \rightarrow 2$), and 61.5% ($5 \rightarrow 3 \rightarrow 4$) in all bootstrap samples (or with a detection rate = 0.938, 0.953 and 0.615), suggesting that they are real and can be reliably detected from data. Note that although `TSCAN` is scalable to a large number of clusters as tree nodes and can handle more complex tree structures, increasing tree complexity can introduce noise and produce many unreliable branches with low detection rates (Fig. S3). Therefore, we proceed with the three branches here as their presence is robustly supported by the available data and also consistent with known biology.

Next, we assess the variability in the branch cell proportions across samples and between conditions (Module 2). Using all 8 donors, the branch cell proportion is 41.1%, 48.4%, and 10.5% for the myeloid, erythroid, and lymphoid branches, respectively. Of note, the proportions show variation across donors (proportion Mean (SD) = 0.41 (0.10) for myeloid, 0.48 (0.11) for erythroid, 0.11 (0.01) for lymphoid). `Lamian` allows one to assess if there is a statistically significant difference in the tree topology (i.e. branch cell proportion) between two sample groups. As an example, comparing the branch cell proportion between male and female donors in the HCA-BM data by applying the binomial logistic regression to each branch did not show significant differences along the myeloid, erythroid, and lymphoid lineages ($p$-values = 0.35, 0.64, 0.94, respectively), suggesting that there is no significant change in tree topology between the two sexes (Fig. 2d). Using multinomial logistic regression showed similar results ($p$-values for odds ratios between male and female = 0.20, 0.39 for myeloid and lymphoid, respectively, using erythroid as the baseline since by default `Lamian` uses the most abundant branch as the baseline category in multinomial logistic regression).

To demonstrate the validity of `Lamian`'s topology stability and differential topology analysis, we performed two sets of simulations. In Simulation 1, we subsampled cells in the myeloid lineage in the HCA-BM data to reduce the myeloid cell number while retaining all cells in the erythroid and lymphoid lineages (Fig. 2e, f). As expected, decreasing the number of cells decreased the detection rate for the myeloid branch (Fig. 2g). For example, when 80% cells in the myeloid lineage were reduced, the detection rate dropped to 0.106 (Fig. 2e, g).

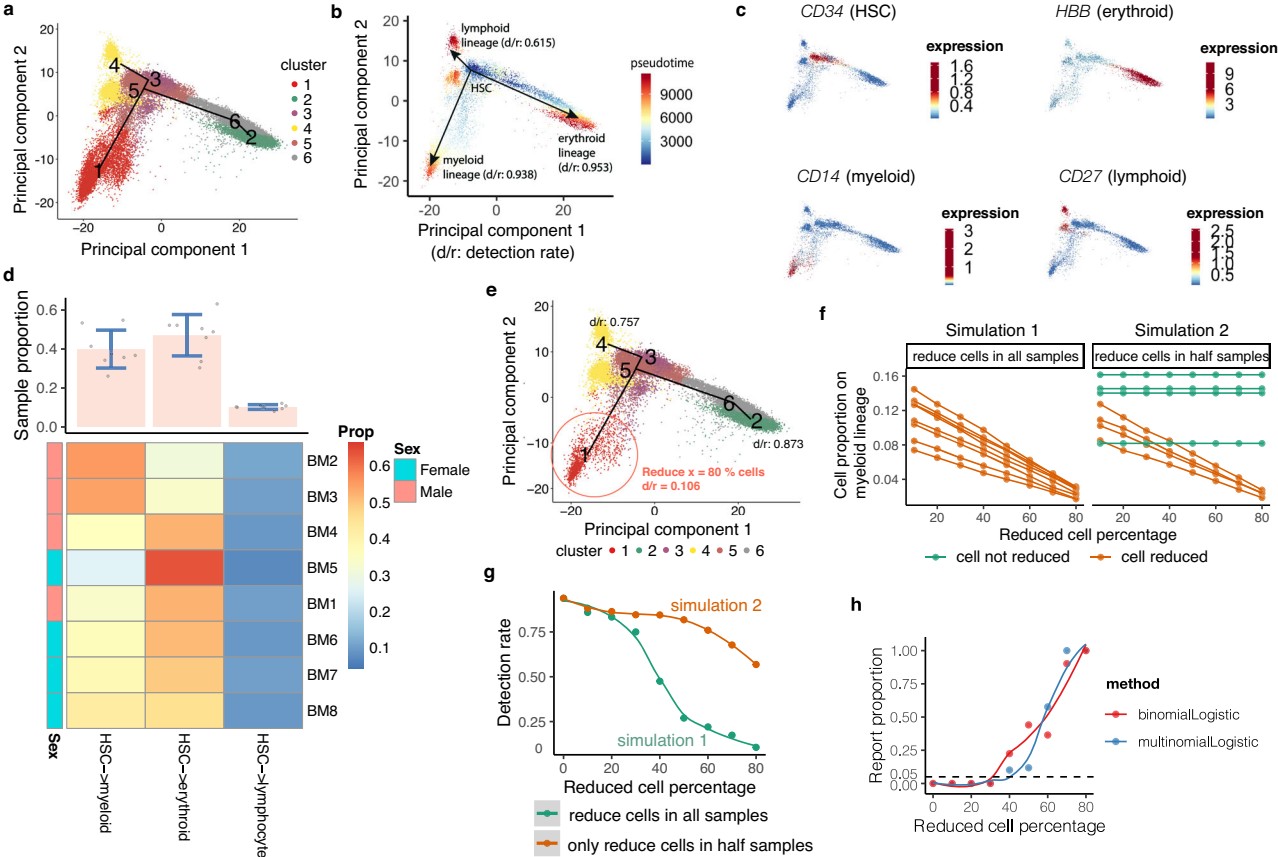

**Fig. 2 | Lamian estimates tree topology stability (Module 1) and tests differential tree topology between sexes (Module 2) in the HCA bone marrow data[37,38]. a** Inferred tree topology using eight integrated scRNA-seq bone marrow samples displayed in the first two principal components (PCs). Dots are cells colored by cluster labels ($k = 6$). **b** Similar to (**a**), but cells are colored by pseudotime. The estimated detection rates (d/r) are shown for three tree branches corresponding to three lineages of hematopoietic stem cell (HSC) differentiation. **c** Similar to (**a**), but cells are colored by the expression of lineage-specific marker genes. **d** Heatmap of sample-level branch cell proportion (Prop, the number of cells in each branch divided by the total number of cells in a sample). The barplot shows the mean (pink bar) ± SD (blue bars) of the branch cell proportion across $n = 8$ samples for each lineage. Cell proportions are displayed as dots and heatmap. **e** Tree topology and detection rates after randomly removing $x = 80\%$ cells on the myeloid lineage (branch $5 \to 1$) as an illustration of simulation. **f** Simulations are conducted by either removing certain percentage of cells (x-axis) along the myeloid lineage across all eight samples (simulation 1: left) or removing cells in only half of the samples (simulation 2: right). **g** Lamian-reported the detection rate of the myeloid lineage (y-axis) after removing different percentages of cells (x-axis) in simulations. **h** In simulation 2, the difference between the two sample groups increases as the reduced cell percentage $x$ increases. For each $x$, 10,000 simulations were run. The proportion of Lamian differential topology test p-values (two-sided) smaller than the significance cutoff 0.05 in the simulations (y-axis) is shown as a function of the reduced cell percentage (x-axis). Source data are provided as a Source Data file.

Hence, the detection rate provides a reasonable measure for quantifying the certainty (or uncertainty) conveyed by the data about the presence of a branch.

In Simulation 2, we reduced the number of cells in the myeloid lineage in four out of the eight samples while retaining all cells in the other two lineages (Fig. 2f). As the number of cells decreased, the detection rate of the myeloid branch again decreased, but at a much slower rate compared to Simulation 1 (Fig. 2g). We found that conditional on the branch being detected, our differential topology tests (Module 2) were able to detect differences in the branch cell proportion between the two groups of samples in this simulation scenario. Most importantly, they controlled the probability of false positives (type I error rate) when there were no differences (i.e. removing no cells or 0% of cells) and also had increasing statistical power to detect true positives as we increased the percent of cells removed in half of the samples (Fig. 2h).

## Lamian comprehensively detects differential pseudotemporal gene expression and cell density

We next illustrate how Lamian adjusts for sample-to-sample variation to identify differential gene expression (Module 3: TDE and XDE tests)

and differential cell density (Module 4: TCD and XCD tests) along pseudotime using the eight samples in the HCA-BM dataset.

First, we ask which genes are varying along pseudotime (Module 3: TDE test). We reasoned that a proper TDE analysis should be able to identify transcriptional programs associated with lineage specification. Applying the TDE test with a 5% FDR cutoff, Lamian identified 8475, 7454 and 8953 TDE genes for the myeloid, erythroid, and lymphoid lineage, respectively (Fig. 3a–c). Among the TDEs, we found known lineage markers corresponding to each lineage, such as *CD14* for myeloid, *HBB* and *GATA1* for erythroid, and *CD3D*, *CD19*, *CD27* for lymphoid. Hence, TDE genes can be used to identify branch lineages in the tree topology. Unsupervised clustering of TDE genes and gene ontology (GO) analysis revealed the dynamic transcriptional programs associated with each lineage (Fig. 3a–c, Fig. S4). For example, as HSCs differentiate to the erythroid lineage, the TDE genes with increasing expression along pseudotime are enriched in red blood cell-related functions such as oxygen transport, whereas genes with functions in other lineages (e.g. CD8-positive, alpha-beta T cell activation, regulation of B cell receptor signaling pathway) show decreasing expression suggesting that they are increasingly suppressed (Fig. S4c, d). Meanwhile, for the lymphoid lineage, the TDE genes with increasing

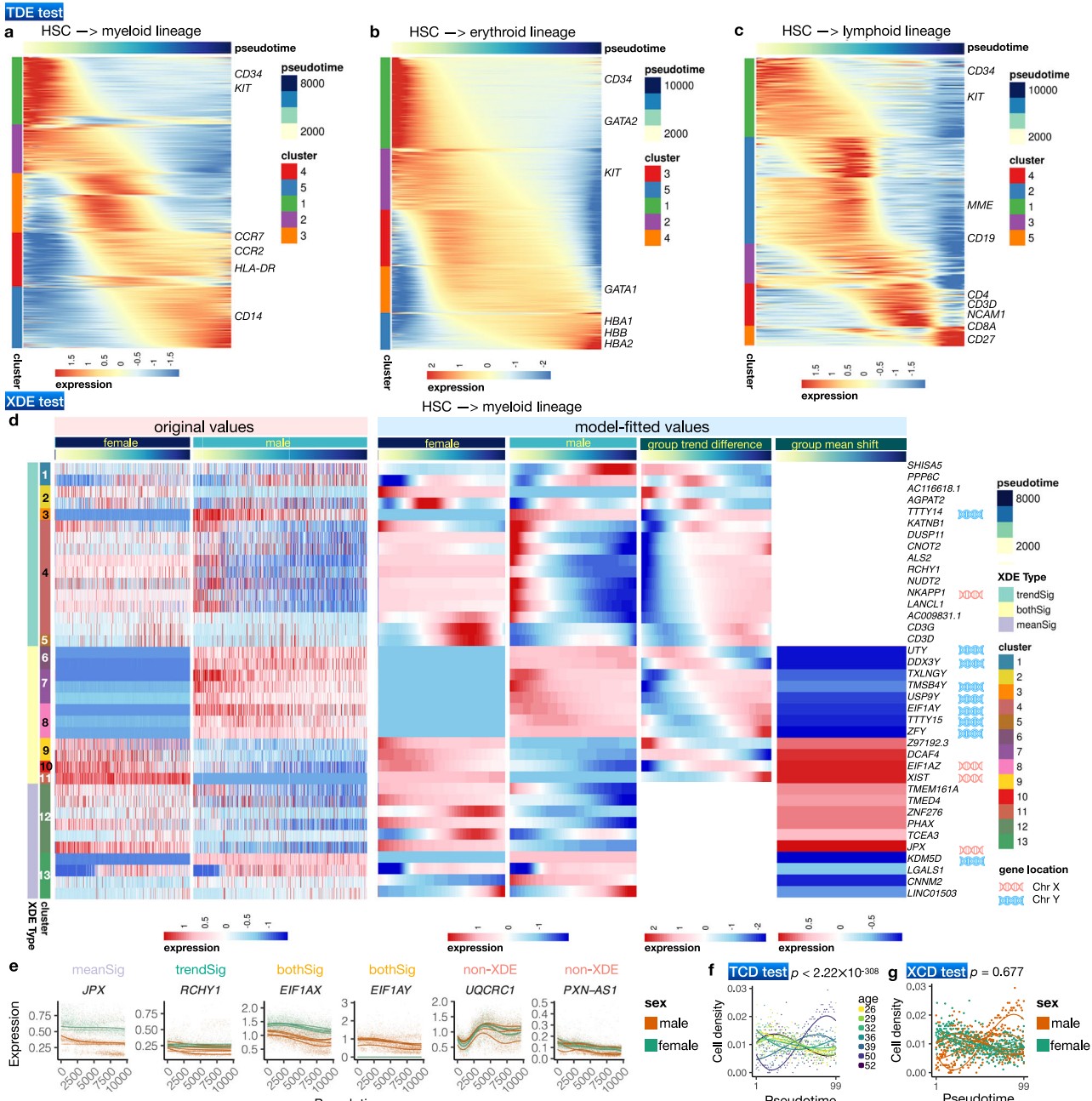

**Fig. 3 | Lamian supports comprehensive analysis of differential expression (Module 3: TDE and XDE tests) and cell density (Module 4: TCD and XCD tests) along pseudotime in the HCA bone marrow data. a–c** Heatmap of model-fitted expression values for TDE genes (rows, FDR < 0.05) along pseudotime for HSC to myeloid (**a**), erythroid (**b**), and lymphoid (**c**) differentiation lineages. Rows are *k*-means clustered. **d** Heatmaps of Lamian-detected XDE genes between male and female along the myeloid lineage (rows, FDR < 0.05). Only 38/43 XDE genes with either significant mean shift or trend difference (10 *meanSig*, 16 *trendSig*, 12 *both-Sig*) are shown. The other XDE genes (*otherSig*: both $FDR_{trend}$ and $FDR_{mean} \geq 0.05$) are not included. The six heatmaps from the left to right correspond to raw normalized gene expression along pseudotime for each sex (left two heatmaps), model fitted gene expression along pseudotime in each sex (middle two heatmaps), trend

difference and mean shift between female and male along pseudotime (right two heatmaps). Genes from chromosomes X and Y are labeled. **e** Example XDE (and non-XDE) gene expression along the myeloid lineage with significant mean shift (*meanSig*), trend difference (*trendSig*), and both (*bothSig*). The fitted curve for each sample is also shown. **f** The model-fitted cell density pseudotemporal patterns in myeloid lineage (one curve per sample) along with Lamian TCD test *p*-value (one-sided $p < 2.22 \times 10^{-308}$, $n = 8$ samples, log-likelihood ratio (LLR) = 207.18). **g** Similar to (**f**) but curves are colored by sex and the Lamian XCD test *p*-value (one-sided $p = 0.677$, $n = 8$, LLR = 2.86) is shown. Results in the erythroid and lymphoid lineages can be found in Fig. S5 (Module 3: XDE tests) and Fig. S6 (Module 4: TCD and XCD tests). Source data are provided as a Source Data file.

expression along pseudotime are enriched in T lineage commitment, whereas genes with decreasing expression lack enrichment of lymphocyte-specific functions (Fig. S4e, f).

Next, we tested whether there are differential gene expression patterns along pseudotime associated with sex as a covariate (Module 3: XDE test). Currently, which genes are sex-associated XDE genes in

this system is not completely known. However, we reasoned that if there is no sex-associated XDE gene, then any XDE gene reported by the algorithm would be noise, and a priori one would not expect genes that are random noise to be associated with sex chromosomes. On the other hand, if XDE genes reported by Lamian in a genome-wide analysis are found to be enriched in sex chromosomes, it would

suggest that sex-associated XDE genes exist and the algorithm is able to detect true XDE signals. For each gene, Lamian reports three FDRs: (1) $FDR_{overall}$ corresponds to testing if a gene is XDE (*overall test*), (2) $FDR_{trend}$ corresponds to testing if an XDE gene has significant trend difference associated with the sample covariate (*trend test*), and (3) $FDR_{mean}$ corresponds to testing if an XDE gene has significant mean shift associated with the covariate (*mean test*). In addition, there are two other categories: both mean and trend differences (*bothSig*), or neither mean or trend differences (*otherSig*). Using the XDE test, Lamian identified 43, 32 and 29 genes (*overall test*) with significant differences (at the 5% $FDR_{overall}$ cutoff) between male and female along the myeloid (Fig. 3d), erythroid (Fig. S5a), and lymphoid (Fig. S5b) lineages, respectively. Next, Lamian further annotated the XDE genes into the gene patterns described above. For the myeloid lineage, this results in 10 genes with mean shift only, 16 genes with trend difference only, and 12 genes with significant changes both in mean and trend (Fig. 3d,e). Among the XDE genes, 33% (*N*=14) are from chromosome X and Y, representing a significant enrichment in sex chromosomes (Fig. 3d, permutation test *p*-value = 0.0036 for chromosome X and *p* = 0.000 for chromosome Y, see Methods). Notably, among the genes that show significant mean shift (with or without trend difference), 12 genes have higher mean expression in males and they consist of 8 genes on Y chromosome and 4 genes on autosomes. Likewise, 10 genes have higher mean in females and they consist of 3 genes on X chromosome and 7 genes on autosomes (Fig. 3d). Unsupervised clustering of XDE genes revealed cascades of their dynamic transcriptional programs. For example, among genes with trend difference only, the difference in *SHISA5* expression between female and male was positive at the beginning and negative at the end of the pseudotime, whereas the difference in *DUSP11* was negative at the beginning and positive at the end (Fig. 3d). Analyses of the erythroid and lymphoid lineages yielded similar results (Fig. S5). Among the XDE genes, a number of them have been reported to have functions related to hematopoietic stem and progenitor cells (e.g. *ALS2*[39], *DDX3Y*[40], *ZFX*[41]). Our analysis here suggests that their functional activities may be sex-dependent.

Finally, we tested for changes in cell density both along the pseudotime (Module 4: TCD test) and whether these patterns were associated with sex as a sample covariate (Module 4: XCD test). The TCD test shows that cell density changed significantly along all three lineages (myeloid: Fig. 3f; erythroid: Fig. S6a; and lymphoid: Fig. S6c) (all *p*-values after adjusting for multiple testing are < $2.22 \times 10^{-308}$), although it is unclear whether the cell density change was due to technical sampling bias (e.g. certain cell types are easier to sample) or real biology. We asked whether the cell density changes were correlated with changes of cell cycle along pseudotime but did not find clear correlation (Fig. S7). In the XCD test, we did not find significant differences in cell density along pseudotime between male and female (myeloid: Fig. 3g; erythroid: Fig. S6b; and lymphoid: Fig. S6d).

## Lamian is more powerful than existing methods to detect differences while controlling the FDR by accounting for sample-level variation

In this section, we demonstrate that Lamian is more powerful than existing methods to detect gene expression differences that are associated with a covariate (Module 3: XDE test). Robustly comparing methods requires datasets with a sufficiently large number of known differential and non-differential genes to serve as the ground truth. Unfortunately, such datasets are not widely available. To address this, we combine simulations with the real HCA-BM data for method evaluation. The HCA-BM dataset is unique in that its male and female samples allow a between-sex comparison. Since there are many sex chromosome genes, the enrichment of sex-associated XDE genes in sex chromosomes can provide an objective and relatively robust benchmark to compare different methods. Thus, the HCA-BM data is used in this article for both method demonstration and evaluation. In

supplementary data, we also demonstrate how incorporating the the sample-to-sample variation into the differential gene expression test along pseudotime (Module 3: TDE test) leads to less false discoveries compared to existing methods that also perform TDE detection.

For XDE analysis, we compared Lamian with limma[42], Monocle2[21], tradeSeq[27], Phenopath[28], and condiments[29]. For Lamian, we also compared two ways to compute *p*-values and FDR: Lamian.pm (default) and Lamian.chisq. As limma is designed to detect differential mean gene expression, we pooled all cells on a pseudotemporal path or branch to create a pseudobulk expression profile (i.e. the average expression across cells for a gene) for each sample. In this way, limma uses the pseudobulk data to detect mean differences between two sample groups. tradeSeq (which is used by Slingshot) is a method originally developed for comparing different branches of a pseudotemporal trajectory within a single sample. Here, we tailored the function to compare the same branch in a pseudotemporal trajectory between two samples. Since tradeSeq and condiments do not consider cross-sample variability, cells from replicate samples were pooled and treated as if they came from a single sample for both methods. Phenopath was run by specifying each cell's sample origin and sample group label and exporting sample group-associated genes. Monocle2 does not directly handle XDE analysis, but we tailored its model and created a new function "monocle2TrajTestCorr" to allow XDE detection in our data (see Methods).

First, we created a null data set based on the HCA-BM data (Methods). Briefly, we first randomly partitioned the eight HCA-BM samples into two groups and removed the group differences to create a dataset where we do not expect any XDE genes between the two groups (Fig. 4a). When Lamian (Lamian.pm) was applied to detect group differences, no XDE genes were reported at 5% FDR cutoff. Using the same cutoff, Lamian.chisq reported 62 XDE genes. By contrast, other methods reported 7846 (monocle2TrajTestCorr), 8783 (tradeSeqPatternTest), 7259 (tradeSeqEarlyDETest), 5822 (tradeSeqDiffEndTest), 7400 (PhenoPath, max.iter = 500, without setting max.iter the program cannot provide results), and 8753 (condiments) differential genes, which are all false positives. Similar to Lamian, limma reported no XDE genes. However, as will be shown below, limma can only detect differences in mean expression and cannot detect trend differences in pseudotemporal patterns.

Building upon the null data set above, we then introduced in silico spike-in differential signals with varying strengths and pseudotemporal patterns between the two sample groups to a random set of genes (details in Methods). In this way, we know which genes are XDE genes and whether they have mean shift, trend difference, or both (Fig. 4b). Next, we applied Lamian to identify XDE genes and clustered genes based on their differential patterns using the default *k*-means clustering (Fig. 4c, d). Using Gaussian mixture model and Louvain clustering yielded similar results (Figs. S8, S9). We compared Lamian XDE genes with XDE analysis from other methods. For all three tests (*overall test*, *trend test*, *mean test*), and across all signal strength levels, the real FDR was smaller than the FDR reported by Lamian(Lamian.pm), demonstrating that Lamian was able to conservatively estimate FDR (Fig. 4e, g). Lamian.chisq also provided reasonable FDR estimates but it slightly underestimated the real FDR. The other methods do not report separate FDRs for mean and trend differences. TradeSeq can be run to detect different types of DE: earlyDETest identifies genes that show expression difference in early pseudotime; patternTest identifies genes that show expression difference along all pseudotime that are equally-spaced; diffEndTest compare the average expression at the end stage of pseudotime. It assigns an FDR for each test. Each of the other methods reports an overall FDR for each gene (see Methods). Unlike Lamian, all existing methods underestimated the real FDR: the difference between the real FDR and their reported FDR was positive in most cases (Fig. 4e). We also stratified XDE genes into three groups - mean shift only, trend difference only, and both

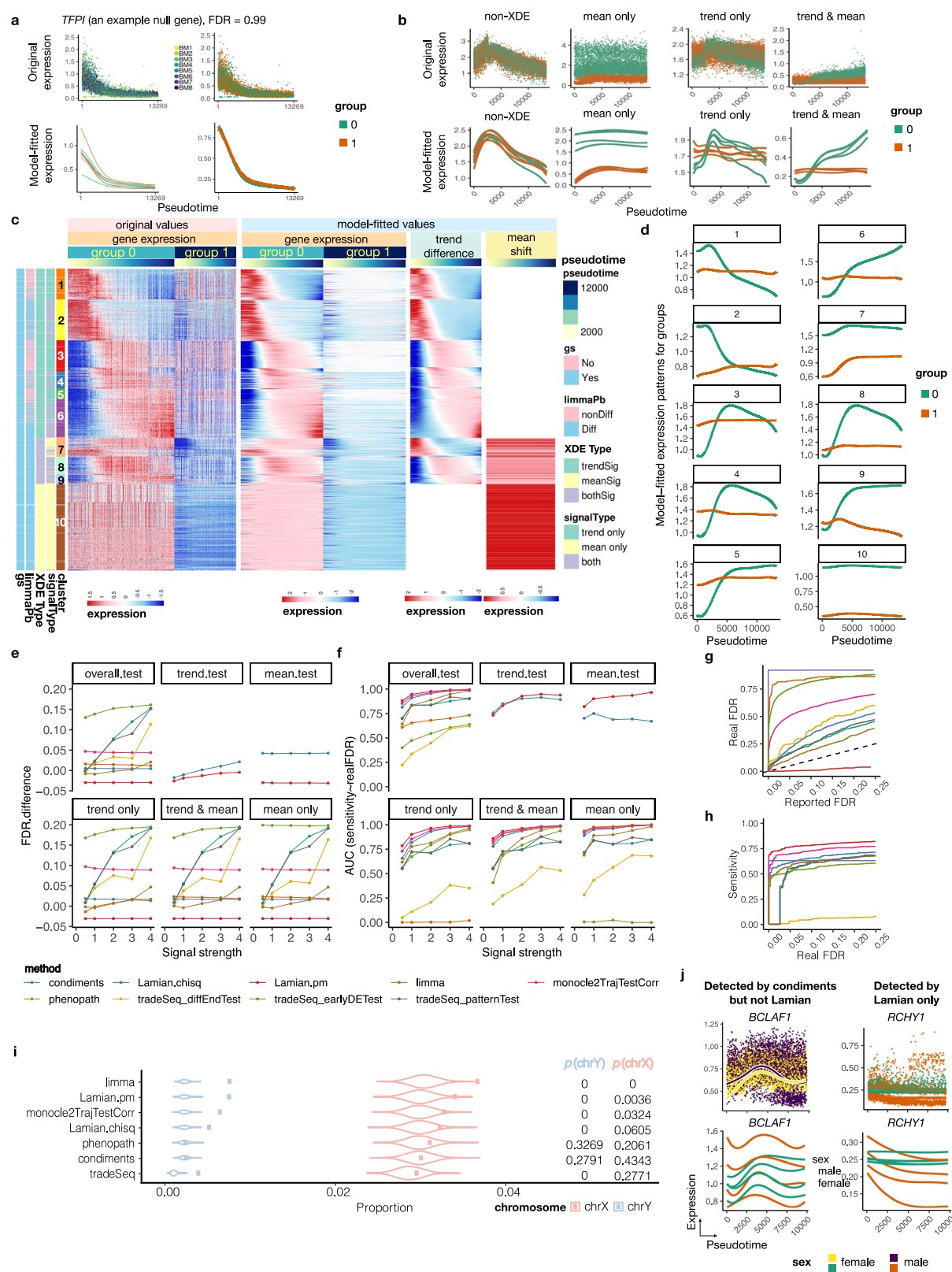

mean and trend differences - based on their true states. Within each stratum, the *FDR_overall* reported by `Lamian` conservatively estimated the real FDR, whereas the other methods underestimated the real FDR (Fig. 4e).

We further compared the statistical power of detecting differences in temporal gene expression associated with sample-level covariates via the sensitivity-realFDR curve and the area under the curve

(AUC) (Fig. 4f, h). The power of detecting XDE genes by `Lamian` increased with increasing signal strength, both for detecting XDE genes overall or for detecting a specific class of XDE genes (Fig. 4f). For detecting all XDE genes (*overall test*), all competing methods had lower power compared to `Lamian` (Fig. 4f). Within `Lamian`, Lamian.pm slightly outperformed Lamian.chisq. In the remaining methods, monocle2TrajTestCorr was among the top but it failed to control FDR.

**Fig. 4 | Evaluation of the FDR control and statistical power for detecting differential genes associated with sample covariate (XDE). a** An example null gene where no differential signals are expected along the trajectory between two sample groups (n = 4 for each group). Curves are model-fitted gene expression patterns for each sample (left) or sample group (right). **b** Examples of non-differential (non-XDE) genes and differential (XDE) genes. Each dot represents a cell. Curves represent sample-level model-fitted patterns. **c** Heatmaps in four white-bar-separated panels to show the expression patterns of significant genes (rows) by cells (columns) ordered by pseudotime. gs:"gold standard". limmaPb: `limma` pseudobulk method. **d** Model-fitted temporal patterns of group 1 and 0 averaged across each gene cluster. **e, f** Performance evaluation of all methods in five spike-in signal strengths settings (x-axis: 0.5, 1, 2, 3, and 4). **e** FDR control performance comparison among Lamian.pm, Lamian.chisq, and other existing methods. FDR

difference is the difference between the area under the realFDR - reportedFDR curve and the diagonal line as illustrated in (**g**). Plots in the 2nd row shows the comparison when gold-standard genes are stratified into trend, mean, and both trend and mean differences. **f** Similar to (**e**) but compares the power using the area under sensitivity - realFDR curve as illustrated in (**h**). **i, j** are about HCA bone marrow data. **i** Overlap (dot) between XDE genes reported by different methods and sex chromosome genes as a gold standard, along with permutation test null distribution (violin plot) of the overlap and p-values (one-sided, $10^4$ permutations, see Fig. S12a). **j** Purple-yellow plots display the patterns output by pooling all cells from all samples in each group by `condiments` while the green-brown plots display those output by `Lamian` which fits sample-specific patterns. Source data are provided as a Source Data file.

When XDE genes are stratified, `limma` had comparable power to `Lamian` for detecting XDE genes with mean shift (i.e. mean shift only or both mean and trend differences) but had zero power to detect genes with trend difference only. `TradeSeq` and `condiments` both had lower power than `Lamian` in all XDE gene categories (Fig. 4f).

In addition to our simulation studies, we compared different methods using the real HCA-BM dataset to detect sex differences (Fig. S10). For the myeloid lineage, `limma` detected 5 XDE genes and all of them were found by `Lamian`. `Lamian` reported an additional 38 genes not found by `limma` (25 with trend difference, 9 with mean shift only) (Fig. S10a). XDE genes found by `Lamian` but not `limma` showed significant enrichment in sex chromosomes (Fisher's exact test p−values: chrX 0.023, chrY $1.05 \times 10^{-12}$), suggesting that these genes are indeed sex related. `TradeSeq`, `condiments`, `PhenoPath` and monocle2TrajTestCorr reported 3677, 4226, 10661 and 10502 XDE genes, respectively. However, a closer examination of their results indicates that a subset of these genes are false positives (Figs. 4j, S11). For example, *BCLAF1* was reported as XDE by `condiments`. For this gene, when cells from replicate samples were treated as if they were from one sample, the fitted gene expression curve along pseudotime are different between male and female, which explains why `condiments` reported the gene as XDE. However, when the gene expression curve is fitted within each sample, the variation among replicate samples is much bigger than the difference between male and female and hence there is no real statistically significant sex difference (Fig. 4j). In contrast, *RCHY1* was an XDE gene reported by `Lamian` but not condiments, the sex difference is clear even after accounting for sample variability (Fig. 4j). Overall, XDE genes reported by `Lamian` and `limma` showed the most significant overlap with both chromosome X and chromosome Y (Fig. 4i). The performance of `Lamian` on the other two lineages was similar (Fig. S12). Indeed, only `Lamian` and `limma` showed significant overlap with both sex chromosomes in all three lineages. Additionally, `Lamian` also showed the largest overlap with genes escaping X-chromosome inactivation (XCI), further demonstrating its top performance in detecting sex-associated XDE (Fig. S13). Collectively, our analyses demonstrate that `Lamian` is better able to detect XDE genes compared to the other existing methods.

In addition to detecting differentially expressed genes along pseudotime that are associated with a sample covariate, `Lamian` can also detect differentially expressed genes along pseudotime without any covariate information (Module 3: TDE test). In this case, there are existing methods, such as `Monocle`, `Slingshot`, `tradeSeq`, `TSCAN` and `PseudotimeDE` that perform a similar test. However, unlike these existing methods, `Lamian` incorporates sample-to-sample variability into the statistical estimation framework. Using simulated data with multiple samples, we found that `Lamian`, compared to existing methods, controls the FDR, while also maintaining strong statistical power for TDE detection (Supplementary Notes, Fig. S14).

Finally, similar to DE analysis, our evaluation also shows that `Lamian` can accurately detect TCD and XCD with a well-controlled type I error rate and high statistical power (Supplementary Notes, Fig. S15).

## Lamian analysis of COVID-19 scRNA-seq data identifies differential CD8 T cell transcriptional programs during a critical stage of disease severity transition

To further demonstrate and evaluate `Lamian`'s ability to detect differences associated with sample covariates along a continuous process, we applied `Lamian` to a COVID-19 peripheral blood mononuclear cell (PBMC) 10x Genomics scRNA-seq dataset obtained from a recent study[43]. The COVID-19 disease severity of a patient may progress from mild to moderate to severe. It was reported that the mild to moderate transition is a critical stage with rapid immune landscape changes that may determine the trajectory of disease progression[43]. CD8+ T cell activation is an important component of COVID-19 patients' immune response to the infection. By analyzing scRNA-seq data from 66 mild and 48 moderate COVID-19 patients, we examined the CD8+ T cell activation program in these patients and asked how it changes during the mild-to-moderate disease severity transition. The relatively large sample size of this dataset also allowed us to partition samples into non-overlapping subsets and systematically benchmark different methods' ability to detect XDE genes by evaluating the detection consistency between different sample subsets.

First, we constructed a pseudotemporal trajectory using a total of 55,953 naive and CD8+ T cells identified from the harmonized PBMC scRNA-seq data (Fig. 5a, Methods). The trajectory contains only one path without branch, thus we skip evaluating the tree branch uncertainty and differential topology. TCD analysis shows statistically significant changes in cell density along the trajectory (Fig. 5d, $p < 2.22 \times 10^{-308}$). It is unclear whether the cell density change here was due to technical sampling bias or has any biological meaning, but the density change was not correlated with cell cycle (Fig. S7). Applying TDE test, `Lamian` identified 2195 TDE genes which were grouped into five clusters (Fig. 5b). Examination of these genes' dynamic expression patterns show that the inferred pseudotemporal trajectory reflects the CD8+ T cell activation process. For example, known naive/memory T cell associated genes including *TCF7*, *SELL* and *IL7R* were found in cluster 1 (Fig. 5b, c). Genes in this cluster showed decreasing expression along pseudotime, consistent with the loss of quiescent characteristics over the activation process. Genes such as *JUNB* and *CD7* are responsible in the induction of differentiation into effectors and thus catch up expression shortly in cluster 2. Genes in cluster 2 also include early activation marker *CD69*, *GZMK* and *AP-1* family members (e.g. *JUNB*, *JUN*), suggesting that this cluster plays a role in the cell fate switch from effector memory T cells to terminal effector T cell phase. By contrast, genes in clusters 4 and 5 both show increasing expression along pseudotime, with cluster 5 reaching its peak expression later than cluster 4. We found that genes encoding functional effector molecules such as *CCL5* and *IFNG* are enriched in cluster 4, and cluster 5 is enriched in both functional activation features such as *GZMB*, *TBX21* and *CX3CR1* and terminal differentiation gene features such as *GNLY*, *CD244* and *CD38* (Fig. 5c).

We next investigated differences in the CD8+ T cell activation program between mild and moderate patients. The analysis of cell

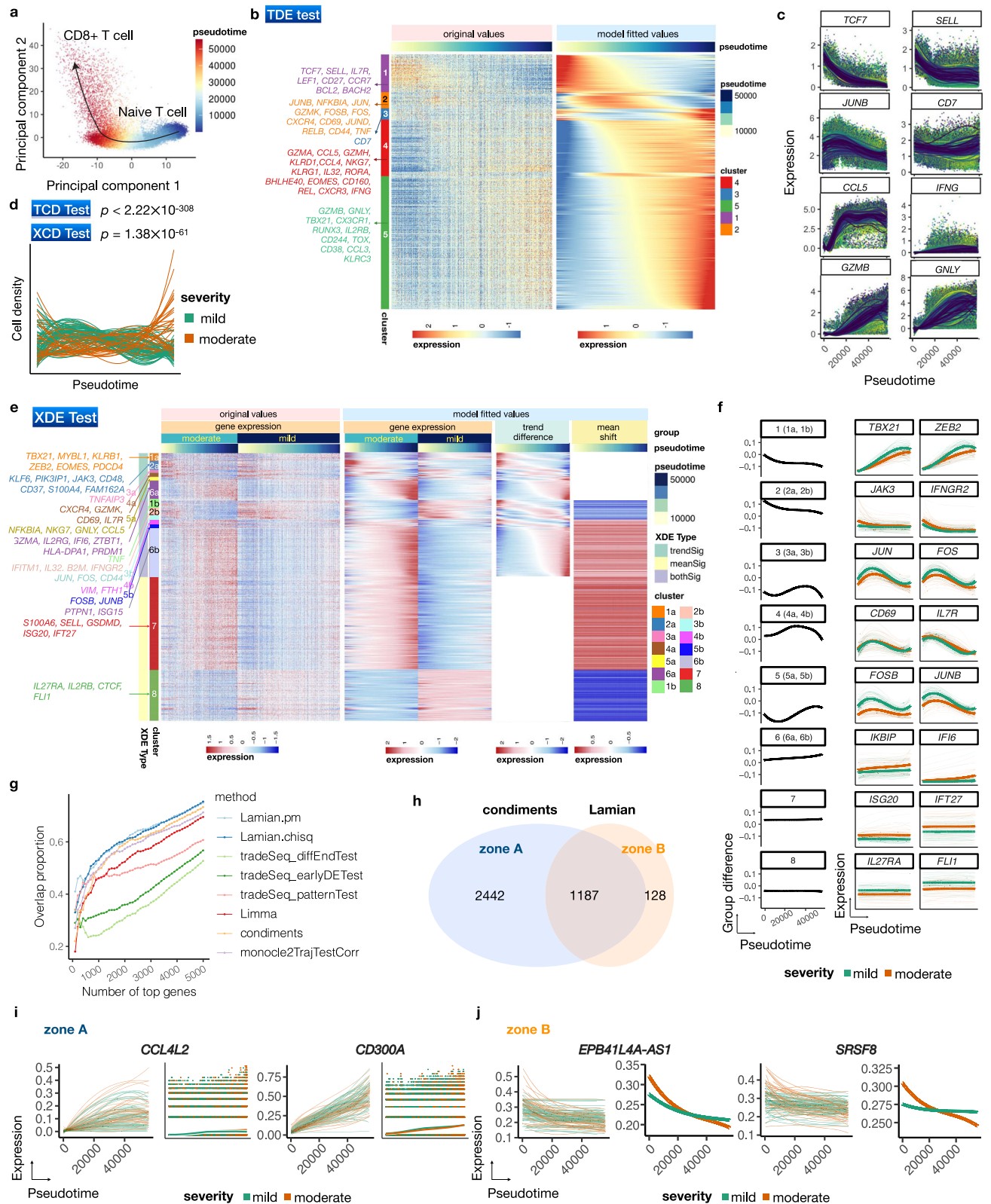

density using XCD test shows that the abundance of activated effector T cells is significantly increased in moderate compared to mild disease ($p = 1.38 \times 10^{-61}$, Fig. 5d). The analysis of gene expression using XDE test identifies 1315 XDE genes, which were grouped into 14 clusters (Fig. 5e). The first 12 clusters contain genes with pseudotemporal trend differences (including bothSig and trendSig), and their trend differences follow 6 major patterns (Fig. 5f, e.g. cluster 2a and 2b have the same

trend difference pattern, but cluster 2a has no significant mean shift whereas cluster 2b has significant mean shift). The last 2 clusters contain genes with mean shift only. In cluster 1, *TBET* (*TBX21*) and *ZEB2* are major transcription factors (TF) for CD8 T cell effector responses[44–47] and drive *IFNG* production. Genes in this cluster tend to have lower expression in moderate patients compared to mild patients and the magnitude of difference increases along the pseudotime

**Fig. 5 | Lamian analysis of COVID-19 samples identifies differential genes related to T-cell activation and inflammation between mild and moderate patients.** **a** Principal component plot of *CD8*+ T cells colored by pseudotime which reflects the T cell activation process. **b** Heatmap showing the original (left) and model-fitted (right) expression of TDE genes along the T cell activation pseudotime. **c** Example TDE genes from different TDE clusters shown in (**b**). Dots are cells. Curves are samples' pseudotemporal patterns fitted by Lamian. **d** Pseudotemporal pattern of the cell density for each sample represented by a curve. *P*-values (one-sided) of TCD and XCD tests indicate significant change along pseudotime and significant difference between mild and moderate samples (*n* = 114). **e** Heatmaps showing the pseudotemporal expression patterns of XDE genes (rows). Cells (columns) from moderate and mild patients are plotted separately and ordered by pseudotime.

White bars are used to separate original temporal gene expression, model-fitted temporal gene expression, group trend difference (moderate group minus mild group), and group mean shift. Example genes related to immune and inflammation processes are marked. **f** Each XDE gene cluster in (**e**) is shown with the averaged group difference (left: black) and two example genes. For each example gene, the curves represent its pseudotemporal pattern for each sample. **g** Samples are randomly partitioned into two datasets. For each method, the proportion of top *N* XDE genes that overlap between the two datasets (y-axis) is shown as a function of *N* (x-axis). **h** A Venn diagram showing the number of XDE genes reported by condiments and Lamian. **i**–**j** Example genes that are reported only by condiments (**i**) or Lamian (**j**). Each gene has two plots: sample-level patterns, and group-level patterns fitted by the method. Source data are provided as a Source Data file.

(Fig. 5e, f), suggesting that mild patients have a more robust functional effector CD8 T cell response. In cluster 6 (incl. 6a and 6b), several interferon stimulated genes such as *IFI6* and *ISG15* as well as terminal differentiation transcription factor *BLIMP-1* (encoded by *PRDM1*)[48] become increasingly more upregulated in moderate patients compared to mild patients along pseudotime, suggesting that a stronger inflammation in moderate patients drives CD8 T cell termination. Together, these data indicate that compared to mild disease, CD8 T cells in moderate COVID-19 patients are programmed to be less functional effector-like and more terminally differentiated. This is consistent with previous observation that comparing to the COVID-19-recovered donors, ongoing disease patients show a more *TEMRA* differentiation with less T-bet+ functional effector CD8 T cells[49].

We further compared Lamian with the other XDE detection methods. We first randomly partitioned the COVID samples into two sets and detected XDE genes between mild and moderate samples within each set. We then examined the proportion of overlap between the two XDE gene lists. By applying Lamian.pm and Lamian.chisq, we achieved the highest overlap proportion between the two partitioned data sets. Phenopath failed to run on this data within one week and with 400GB memory. Among the remaining methods, condiments performed slightly better than the other methods, followed by monocle2TrajTestCorr, limma, and tradeSeq, but all methods performed worse than Lamian (Fig. 5g). This suggests that XDE genes identified by Lamian are most reproducible when analyzing different sets of samples. A closer examination of the sample-level pseudotemporal curves shows that XDE genes detected by the other methods contain a large number of false positives. Take condiments, the top performer in the remaining methods, as an example. Condiments reported 3809 genes, including 2622 that were not detected by Lamian. The sample-level curves show that many of such genes did not show clear group differences after accounting for sample-level variability (Fig. 5i). Lamian reported 1315 XDE genes, including 128 that were solely detected by Lamian. For these genes, group differences cannot be explained only by the sample-level variation (Fig. 5j).

Collectively, our analyses demonstrate that Lamian provides a powerful tool for identifying differences associated with covariates that the other methods do not offer. The COVID analysis also demonstrates how one can use multi-sample differential pseudotime analysis to understand dynamic gene expression programs in a disease.

**Lamian analysis of tuberculosis data demonstrates efficiency in handling large datasets while adjusting for batch effects**

To demonstrate and evaluate Lamian 's ability to analyze large datasets and detect differences associated with sample covariates while adjusting for potential confounders such as batch effects, we analyzed an atlas-size dataset consisting of 337,191 memory T cells from 184 donors (100 females and 84 males) in a tuberculosis (TB) progression cohort[50] (Fig. 6a, b). This dataset has recently been used for demonstrating co-varying neighborhood analysis and biologically meaningful cell abundance differences between males and females were reported

along the second principal component of the co-varying neighborhood abundance matrix (NAM-PC2)[51]. Consistent with that study, we provided NAM-PC2 as cells' pseudotime and conducted differential analysis (Fig. 6a). Samples in this dataset are profiled in multiple batches (Fig. 6b). We added batch indicators to the Lamian regression model to account for batch effects.

TCD analysis shows that the cell density changed significantly along the trajectory (Fig. 6c), but the density change was not correlated with cell cycle (Fig. S7). Like previous examples, while the cell density change here could reflect real biology, we cannot rule out the possibility that it is due to technical sampling bias. TDE analysis shows that genes with expression elevated in the middle range of pseudotime (cluster 2) are enriched in regulation of T cell activation, and the genes with strong upregulation in the later stage of pseudotime (clusters 5) are enriched in gene ontology terms such as "immune response - activating cell surface receptor signaling pathway", suggesting that the pseudotime reflects a T cell activation process (Fig. S16a, b). Consistent with this, typical effector transcription factors, such as *ZEB2*, *TBX21*, as well as other effector genes such as *GZM* family members and *PRF1*, all show a clear increasing pattern over the pseudotime (Fig. S16a).

Consistent with previous report[51], XCD test revealed significant cell abundance changes between males and females along the pseudotime. T cells from females were more enriched towards naive status (early pseudotime) compared to T cells from males. By contrast, male cells were more enriched towards terminal activation status (late pseudotime) (Fig. 6c).

XDE analysis between male and female identified 1120 sex-associated differential genes grouped into 14 clusters (Fig. 6d, S16c). Among them, 12 clusters had trend differences or both trend differences and mean shifts. The trend differences of these 12 clusters can be further grouped into 6 patterns. For example, pattern 4 (clusters 4a, 4b) were more highly expressed in males than females and their difference has an overall increasing and then decreasing trend (Fig. S16c). These genes were enriched in proteins targeting ER and membrane (Fig. S16d). By contrast, genes in cluster 1 were more highly expressed in females compared to males along pseudotime, and the absolute difference between female and male first increased and then decreased (Fig. S16c). These genes were enriched in gene ontology terms including lymphocyte activation, leukocyte activation and other immune-activation-related features (Fig. S16d), suggesting that T cells from the female group have a stronger T cell response to the disease. The data also suggest that the induction of these genes happens earlier in females along the T cell activation pseudotime. Furthermore, key T cell activation transcription factors, such as *ID2* and *STAT5B*, were involved in this activation process, along with other functional effector molecules such as *GZMA*, *CCL5* and *GZMK* (Fig. 6d). These molecular-level discoveries are consistent with the phenotype of female patients infected with TB having a higher *TH1* response feature compared to male patients[52]. On the other hand, the increased abundance of naive T cells in females compared to males (Fig. 6c) could potentially provide a compensating mechanism to control the total

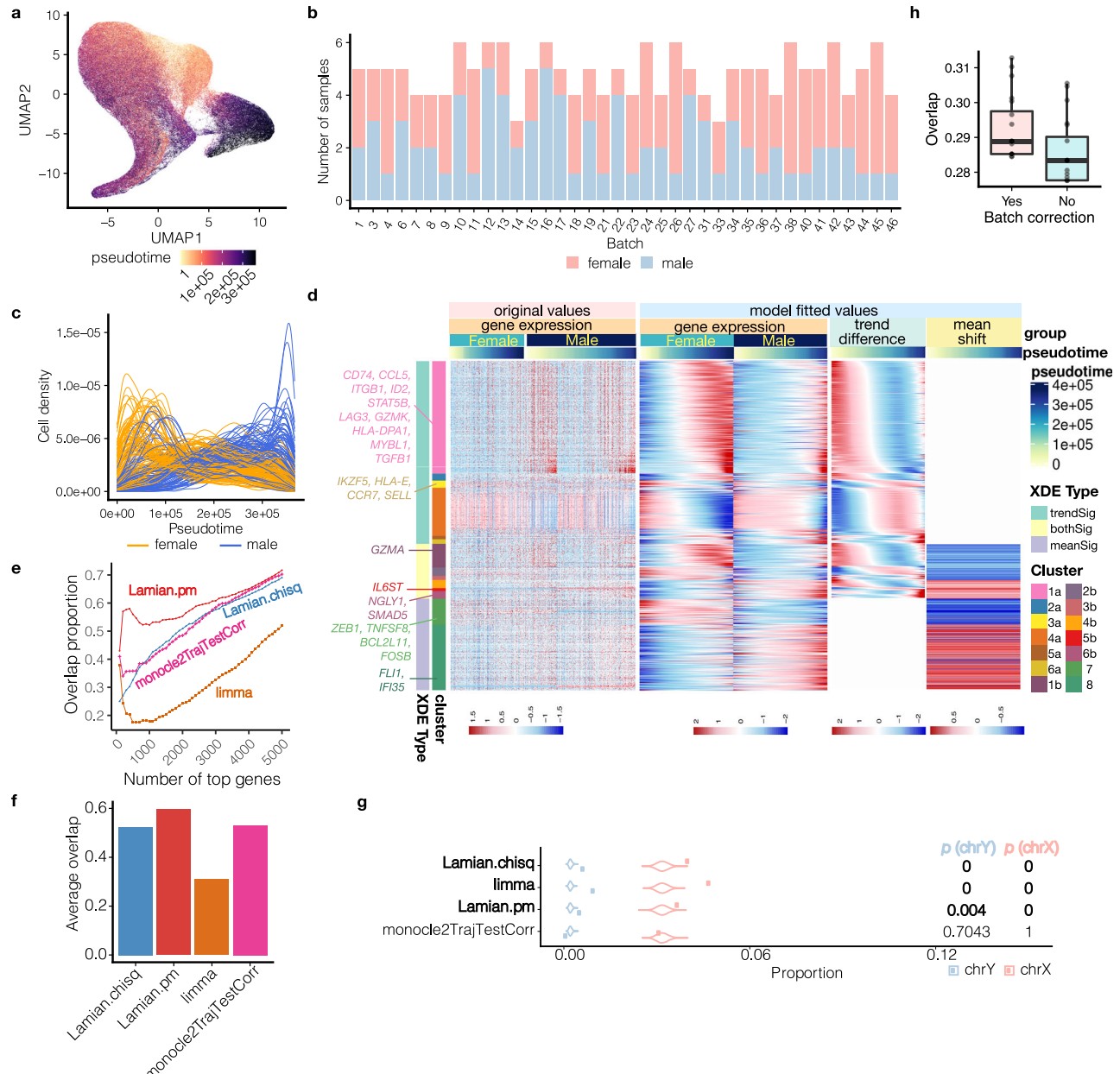

**Fig. 6 | Lamian analysis of 184 tuberculosis (TB) samples identifies differential pseudotemporal genes and cell density between male and female while controlling for batch effects. a** UMAP[51] showing cells color-coded by pseudotime constructed using NAM PC2. **b** Study design showing sample batches. **c** Cell density distribution along pseudotime for each sample (curve) (TCD test $p < 2.22 \times 10^{-308}$, XCD test $p = 8.9 \times 10^{-6}$, both are one-sided). **d** Heatmaps showing the expression patterns of sex-associated XDE genes (rows) in cells (columns) ordered by pseudotime. White bars are used to separate original expression, model-fitted expression, group trend difference (male minus female), and group mean shift. **e** Samples are randomly partitioned into two datasets. For each method, the proportion of top $N$ XDE genes that overlap between the two datasets (y-axis) is shown as a function of $N$ (x-axis). **f** Average overlap in (**e**) (i.e. area under the curve divided by the x-axis

length). **g** Overlap between XDE genes reported by different methods and sex chromosome genes, along with $p$-values in permutation test (one-sided, $10^4$ permutations). Methods without outputs within 1 week and 400GB are not shown in (**e**–**g**). **h** Proportion of XDE genes that overlap between two partitions of data, with versus without batch correction. The TB dataset was split into two: one contains one batch, and the other one contains all remaining batches (only use the batches with ≥2 samples in both sexes). For the one with multiple batches, Lamian is run with and without batch correction. The data split was repeated 12 times. Each boxplot shows the distribution (center: median; bounds of box: 1st and 3rd quartiles; bounds of whiskers: data points within 1.5 IQR from the box; minima; maxima) of overlap in the $n = 12$ splits. Source data are provided as a Source Data file.

amount of functional immune activation in vivo, via reducing the number of responding cells in females when the per cell effector function is high.

To compare with the other XDE methods, We randomly partitioned the samples into two sets, detected XDE genes between female and male samples within each set, and examined the proportion of overlap between the two XDE gene lists. `Phenopath`, `tradeSeq` and `condiments` failed to run on this atlas-size data. Among the remaining

methods, XDE gene rankings produced by Lamian.pm were most reproducible, followed by Lamian.chisq and monocle2TrajTestCorr (Fig. 6e, f). XDE genes reported by `Lamian` and `limma` showed significant overlap with the sex chromosome (X and Y) genes and largest overlap with the genes escaping X-chromosome inactivation, whereas monocle2TrajTestCorr did not (Figs. 6g, S17).

Finally, we compared `Lamian` results with and without adjusting for batch effects by examining the consistency of XDE genes between

separate partitions of samples. Adjusting for batch effects improved the analysis, yielding more reproducible XDE genes (Fig. 6h).

## Computational efficiency

Lamian is computationally tractable. For analyzing the HCA bone marrow dataset with 32,819 cells and 8 samples, Lamian.pm took 4.2 hours to run the whole pipeline (0.1h for trajectory variability, 2.7 h for XDE detection and 1.4h for TDE detection, 0.01h for cell density test) on a computer cluster with 25 CPUs (2.5 GHz CPU and at most 163 GB RAM combining all CPUs). Lamian.chisq is more efficient and only took 0.5 h and 6.2 GB RAM. For analyzing 39,512 CD8 T cells in the COVID dataset with 114 samples, Lamian.pm and Lamian.chisq took 37 and 2.9 hours and 285 and 5 GB RAM, respectively, to run the whole analysis pipeline. For atlas-size data (more than $10^5$ cells), Lamian uses HDF5 file format to store and analyze the data to increase the computational efficiency. For the TB dataset with 337,191 cells from 184 samples, Lamian.pm can finish the analysis with 114.1 h and 243 GB memory, and Lamian.chisq took 15.8 h and 5.1 GB memory. As the TB analysis only involved Lamian modules 3 and 4, we additionally benchmarked modules 1 and 2 using synthetic data and found that they are also scalable to atlas level data (Fig. 18a, b). XDE analysis is the most time-consuming and memory-intensive component of the whole analysis. For this component, Fig. S18c, d and Table S2 further compare the computation time and memory of different methods on different datasets. Lamian.chisq and monocle2TrajTestCorr are the fastest and Lamian.chisq requires least memory. Lamian.pm is slower but it is capable of handling atlas-level data. Unlike Lamian, PhenoPath, condiments and tradeSeq are not scalable to large datasets. condiments and tradeSeq failed to handle the TB data, and PhenoPath failed to handle both COVID and TB data within one week and with 400GB memory.

## Discussion

In summary, Lamian provides a systematic solution to multi-sample pseudotime analysis capable of detecting topology, gene expression and cell density differences between different conditions. In biomedical research, while making new discoveries is exciting, ensuring that the discoveries are real and replicable is equally important. One challenge the scientific community faces today is that many findings cannot be replicated or validated in independent studies[53]. One important contributor to this problem is the flawed statistical analyses which can produce a large number of false discoveries. Such irreplicable false discoveries can be detrimental by distracting investigators from real signals and misleading subsequent research efforts, resulting in substantial waste of precious human and financial resources. In the context of pseudotime analysis, our results demonstrate that, due to lack of appropriate consideration of cross-sample variability, existing pseudotime methods can report thousands of false differential genes in null simulations where the data do not contain any true differential signals. This highlights a critical gap in the pseudotime literature and an open challenge that needs to be addressed. Lamian fills in this gap by introducing a comprehensive statistical framework, including a functional mixed effects model, to account for cross-sample variability in the multi-sample differential pseudotime analysis. In order to benchmark this method, we applied it to both simulated and real data. We note that the analyses of three real datasets (HCA-BM, COVID, TB) mainly serve the purpose of illustrating and evaluating Lamian, and that making new biological discoveries per se is not the focus of this study. The orthogonal information (e.g. sex chromosome genes) and large sample size (e.g. COVID and TB data) available in these data make it possible to objectively and robustly compare different methods by quantifying the overlap with orthogonal information or between independent data partitions. Our results in these simulated and real data show

that the solution provided by Lamian substantially outperforms other existing methods to prioritize true discoveries and filter out false discoveries that are not generalizable to new samples.

Lamian is a free and open source R package with a modular structure. While we demonstrated its default pipeline in this article, users can replace certain analysis modules by their own data or algorithm. For sample harmonization, we used Seurat[32] to embed cells into a common low-dimensional space. One could also use other methods such as Harmony[33] and scVI[34]. For example, in our HCA bone marrow analysis, using Seurat, Harmony and scVI produced similar branching structure and differential genes (Figs. S19, S20). In real applications, different harmonization methods may perform differently. We recommend users to compare different harmonization methods and choose the one most consistent with the existing knowledge. A systematic comparison of harmonization methods is beyond the scope of this study. Readers are referred to a recent benchmark study[35] for discussions on which harmonization methods to use under different conditions.

In Lamian, TSCAN is used as the default method to construct pseudotime due to its flexibility and scalability. TSCAN uses the cluster-based MST approach to reduce the number of tree nodes (e.g. clustering 1 million cells into 1000 clusters will result in only 1000 tree nodes instead of 1 million tree nodes) and hence can handle a large number of cells. In terms of flexibility, while TSCAN by default determines the number of cell clusters automatically via an elbow method, users have the option to specify their own cluster number if they are not satisfied with the default cluster number. Increasing the cluster number may create a more complex tree with a more detailed view of the biological process. However, the increased complexity could also introduce noise and false branches (Fig. S3). In real applications, even though one may construct a more complex tree, a key question is whether one can trust that the tree structure is real rather than random noise. Answering this question is challenging when there is little or no prior knowledge about the underlying biological process. Lamian addresses this issue via bootstrap and detection rate. A low confidence tree branch can be reflected by its low detection rate. Based on our experience, applying this criterion often leads to relatively simple tree structure. This does not imply that the underlying tree structure is necessarily simple. Instead, it only reflects the fact that the available data can only provide enough information to support robust conclusion on a relatively simple tree and there is not enough information to draw conclusions on a more complex tree structure. If users have prior knowledge that supports a more complex tree structure, they can use the manual option provided by TSCAN to choose a larger cluster number to define more detailed tree structure. In addition to cluster number, TSCAN also allows users to manually specify the order of cell clusters in the trajectory, providing another way to adjust the trajectory based on users' prior knowledge. These options in TSCAN allow users to conveniently perform analysis on more complex tree structures. Once the tree topology is given, all the remaining analyses including those in modules 2 to 4 (differential topology, XDE, TDE, XCD, TCD) can be carried out as usual.

Besides TSCAN, one also has an option to use user-provided pseudotemporal trajectories as illustrated in the TB analysis. In fact, one may use Lamian modules 3 and 4 as downstream analysis tools for other pseudotime methods such as Monocle2, Monocle3 and slingshot. However, Lamian modules 1 and 2 which construct trajectory and quantify its uncertainty and variation currently do not support other pseudotime methods due to various issues including scalability and implementation challenges (Supplementary Notes). For example, slingshot, a popular MST method similar to TSCAN, does not scale well to bootstrap due to its time-consuming principal curve fitting. For Monocle2 and Monocle3, modifying Lamian's trajectory topology uncertainty module to support them is non-trivial due to lack of interoperability between the data structures used by different

methods to represent trajectory topology. A future direction is to tailor these methods to improve their scalability and/or interoperability to allow their seamless connection to `Lamian` modules 1 and 2.

Uncertainties in the pseudotime analysis include both the uncertainty of the inferred pseudotemporal trajectory and the uncertainty of gene expression or cell abundance conditional on pseudotime. In `Lamian`, the trajectory inference uncertainties are characterized by bootstraping cells to compute detection rates. Conceptually, one could also account for the pseudotime reconstruction uncertainty in the downstream differential gene expression and cell abundance analysis by fitting the temporal gene expression and cell abundance curves for each bootstrapped tree. However, practically, it will make the differential analysis difficult to implement and make the results difficult to summarize and report. This is because trees reconstructed from different bootstrap samples can have different topologies due to the randomness. A branch that appears in one tree may not exist in another tree, and often it is unclear how one should align branches of different trees. It is unrealistic to enumerate all branches that occurred in bootstrapped trees, and the meaning of differential expression along a branch can be unclear if the branch does not always exist. For this reason, `Lamian` separated the evaluation of uncertainties of the inferred pseudotemporal trajectory (i.e. the construction of minimum spanning tree) and the evaluation of uncertainties of gene expression using a sequential "conditional" procedure. In other words, our module 1 evaluates the uncertainty of pseudotime (MST) construction. Next, conditional on a tree lineage and conditional on the corresponding inferred pseudotime, modules 3 and 4 perform differential analyses using bootstrap sampling to account for the cell-level uncertainty, followed by modelling sample- and cell-level variability to account for gene expression variability and uncertainty. This sequential procedure avoids the complication of comparing different trees, making it easier for summarizing the analysis results to end users. Thus, while it may be imperfect, it provides a practical solution to this complicated problem. Developing better methods that can simultaneously account for all sources of uncertainties including pseudotime inference uncertainty, gene expression and cell abundance inference uncertainty, and cross-sample and cross-cell variability remains a future research topic that warrants further investigation.

Currently, the statistical model in `Lamian` is formulated for scRNA-seq data. However, its general principle and statistical framework may be applicable to other data types such as single-cell ATAC-seq data as well, although the other data types may have different data characteristics that require one to tailor the model accordingly. These extensions will be a topic for future research.

## Methods
### Data
**Human Cell Atlas bone marrow dataset (HCA-BM)**. The raw count matrix of bone marrow scRNA-seq data sequenced in 10x Genomics platform from 8 healthy donors were downloaded from the Human Cell Atlas (HCA) data portal[37,38] (immune cell atlas of human hematopoietic system). The raw data consist of 42,925 genes and 290,861 cells. Cells with fewer than 5000 reads, fewer than 1,000 expressed genes (i.e. genes with nonzero read count), or more than 10% of reads mapped to the mitochondrial genome were deemed as low quality and filtered out. We also filtered out genes that were expressed in less than 0.1% of all cells. This results in a data matrix of 22,401 genes × 32,819 cells used for subsequent analyses. See Supplementary Notes and Fig. S21 for a more detailed discussion of filtering parameters and additional quality control (QC) plots.

**COVID19 dataset (COVID-Su)**. The raw count matrices of 256 PBMC 10x Genomics scRNA-seq samples from 139 COVID-19 patients were downloaded from E-MTAB-9357[43]. We filtered out cells with fewer than 2,000 reads or 500 expressed genes or more than 10% mitochondrial

reads. We also filtered out samples with fewer than 500 cells. `Seurat`(v.3.2.1)[32] was applied to process, integrate data across samples and perform the cellular clustering with default settings. Cell types were annotated based on known marker genes. CD8+ T cells were identified using *CD3D* expression > 1 log-scaled library-size-normalized `SAVER`-imputed read counts and *CD8A* expression > 1 criterion. Samples with fewer than 100 CD8+ T cells were filtered out. Among the total of 161 samples that passed the filters, we focused on analyzing samples from 66 mild and 48 moderate patients subsequently. This results in a data matrix of 26, 701 genes × 55, 953 cells used for subsequent analyses.

**Tuberculosis (TB) dataset**. We obtained the pre-processed gene expression file GSE158769_exprs_raw.tsv.gz of the TBRU dataset directly from GSE158769[50]. This dataset consists of 500,089 memory T cells from 259 donors that were profiled with CITE-seq. NAM-PC1 and NAM-PC2 coordinates of 393,998 cells were obtained from the authors of CNA[51] (also see its supplementary file of Fig. 5A[51]). Batches with at least one male sample and at least one female sample were retained. Samples with at least 1000 cells and genes with expression values > 0.1 in at least 1% of cells were retained. The NAM-PC2 values were used to order cells to produce pseudotime. These processing steps result in a data matrix of 9317 genes × 337,191 cells (184 samples from 38 batches, with 100 female and 84 male samples) which was used for subsequent analyses.

### Data harmonization and preprocessing
Before `Lamian` analysis, one needs to first harmonize data from different samples. The purpose of harmonization is to match cells of the same type across samples so that the same type of cells can be compared across samples. As such, it removes both biological differences of interest (e.g. the same cell type can have differential expression between two sample conditions which is removed by harmonization) and unwanted technical differences (e.g. batch effects) among samples. In downstream analyses, since one is interested in biological variation across samples and conditions, `Lamian` will use the original normalized gene expression values instead of the harmonization-corrected expression values, and it will use a regression framework to remove unwanted technical variations such as batch effects but retain biological differences across samples. In this study, we used `Seurat`(v.3.2.1)[32] to integrate (or harmonize) multiple samples in each dataset. For differential expression (DE) analysis, `SAVER` was used to impute gene expression values to address the drop-outs in the data. All DE methods used imputed values except `tradeSeq` and `condiments` since they require count values as inputs. Principal Component Analysis (PCA) and Uniform Manifold Approximation and Projection (UMAP)[54] were used for visualization, and they were both run using default settings.

### Constructing pseudotemporal trajectory and evaluating its uncertainty
In the default mode of `Lamian`, after samples are integrated, the harmonized data are used to construct pseudotemporal trajectory using a cluster-based minimum spanning tree (cMST) approach. *K*-means clustering is applied to cluster cells based on the top principal components (PCs) of log2-transformed library-size-normalized gene expression profiles. Trajectories are then inferred as in `TSCAN` by constructing a minimum spanning tree that treats cluster centers as nodes. The number of PCs and the cell cluster number are both determined using an elbow method[23]. The origin of the pseudotime is specified by users based on marker gene expression (or the origin cell types if users input the cell types annotation). For example, in the bone marrow data, the cluster with the highest expression of hematopoietic stem cell (HSC) marker *CD34* was set as the origin. Once the origin of the trajectory is given, one can enumerate all paths and branches. Branches are identified based on nodes with degree > 2.

For each of the branch, we characterize its uncertainty using its detection rate in 1000 bootstrap samples. Each bootstrap sample is created by sampling cells from the original data with replacement. Cells in the bootstrap sample are used to reconstruct pseudotemporal trajectory using the same cMST approach as in the original data. The origin of the pseudotime in a bootstrap sample is determined using the cell cluster with the smallest mean of cells' pseudotime in the original data. We then ask whether each branch in the original data is also identified in the bootstrap sample by performing pairwise comparison of branches between the original and bootstrap data. For a pair of branches (one from original data and one from bootstrap sample), we use the Jaccard index to evaluate their overlap (i.e., what percentage of cells in these two branches are shared). If the Jaccard index exceeds a cutoff, then the branch in the original data is called detected in the bootstrap sample. To determine the cutoff, a null distribution of Jaccard index is constructed by evaluating the overlap between the cells in the branch and a randomly sampled set of cells with the cell number matching those in the branch for 1000 times. The 0.99 quantile of this null distribution is used as the cutoff. After comparing the original trajectory with all bootstrap samples, the detection rate of a branch is defined as the proportion of bootstrap samples in which the original branch can be detected.

## Tree variability across samples and differential topology analysis

For each sample, the proportion of cells in each branch is calculated and referred to as "branch cell proportion". For each branch, the variance of branch cell proportion across samples is reported to characterize its cross-sample variability.

To test differential topology, by default a binomial logistic regression model is fitted for each branch. Here the branch cell count is treated as the dependent variable and modeled using binomial distribution $Binomial(n, p)$ where $n$ is the total cell count in a sample and $p$ is the underlying true branch cell proportion. The regression models $\log(p/(1-p))$ as a function of the sample covariates which are specified by users as the independent variables. Statistical significance of the association between a sample covariate and the branch cell proportion is determined by testing whether the corresponding regression coefficient is zero using Wald test. The $p$-values are adjusted for multiple testing using the Benjamini-Hochberg procedure to obtain false discovery rates (FDRs)[31]. By default, FDR≤0.05 is used as the significance cutoff. As an example, if two conditions have different topologies and each has a condition-specific branch, then after data integration and trajectory construction, one will have a branch (branch A) that only contains cells from condition 1, and another branch (branch B) that only contains cells from condition 2. The differential topology test will test the cell proportion differences between the two conditions for each branch. For branch A, it will report that there is a significant difference in cell abundance between condition 1 and condition 2, and it will also report the mean cell proportion in that branch for each condition. Users will be able to see that the proportion of cells in branch A in each sample from condition 2 is almost zero, but the cell proportion in branch A for condition 1 is above zero. Therefore, based on this information one will know that branch A is likely condition-1-specific. Similarly, one can tell that branch B is condition-2-specific since the cell proportion on that branch is almost zero for condition 1 and is positive for condition 2, and the difference between the two conditions is significant.

Optionally, users can also fit a multinomial logistic regression by considering all branches jointly. Assume there are $L$ branches, and let $p_1, ..., p_L$ be the underlying true branch cell proportions for these branches in a given sample ($\sum_{l=1}^{L} p_l = 1$). In the multinomial regression, one chooses a branch as the reference branch. By default,

Lamian chooses the most abundant branch (i.e. the branch with the largest number of cells) as the reference branch. Without loss of generality, let $L$ denote the reference branch. The model assumes that the branch cell counts in a sample follow a multinomial distribution $Multinomial(n, (p_1, ..., p_L))$ where $n$ is the total cell count of the sample. It models $\log(p_l/p_L)$ ($l = 1, ..., L-1$) as functions of sample covariates. Statistical significance of the association between a sample covariate and log odds is determined by testing whether the corresponding regression coefficients are zero, similar to binomial logisitic regression. Compared to fitting a binomial logistic regression for each branch, multinomial logistic regression allows one to account for the fact that cell abundance in different branches are not independent. The binomial logistic regression, on the other hand, may allow one to conveniently explore whether branch cell proportion of a given branch increases or decreases (Supplementary Notes).

## Modeling gene expression along pseudotime

Given a pseudotemporal path or branch, Lamian will describe how gene expression $Y$ varies along pseudotime $t$ and characterize the relationship between each gene's pseudotemporal expression pattern $Y(t)$ and $V$ sample covariates $X_1, ..., X_V$ (e.g. disease status, age, etc.) using a functional mixed effects model.

Without loss of generality, below we presents the statistical model for one gene. All other genes can be analyzed in the same way. We use lowercase letters $s$ and $c$ to denote sample and cell, respectively, and we use capital letter $S$ to denote the total number of samples. Assume that sample $s$ consists of $C_s$ cells. Let $t_{sc}$ be the pseudotime of cell $c$ in sample $s$. Given a gene, let $y_{sc}$ denote its expression level in cell $c$ of sample $s$. Let $\mathbf{x}_s = (1, x_{s1}, ..., x_{sV})^T$ be the realized values of covariates in sample $s$. Here, we introduced an additional term $x_{s0} \equiv 1$ as an intercept term for the subsequent regression model.

We model each gene's expression pattern along pseudotime as functional curves and represent the function using a total of $K+1$ B-spline basis functions $\phi_0(t), \phi_1(t), ..., \phi_K(t)$. Here $K$ is the number of equidistant knots used to define B-spline bases. The gene's functional curve in sample $s$ is $Y_s(t) = \sum_{k=0}^{K} \phi_k(t)b_{sk}$. For each gene, the optimal $K$ is automatically chosen by comparing values ranging from 0 to a predefined maximum (20 by default) and selecting the one that minimizes the Bayesian Information Criterion (BIC). The BIC for a given $K$ is calculated as $BIC_K = KS \ln(\sum_s C_s) - 2\sum_s l_{K,s} + const$. Here $const$ is a constant term that does not depend on $K$ (hence irrelevant for finding optimal $K$), and $l_{K,s}$ is the log-likelihood of the B-spline regression for sample $s$ (i.e. we fit a linear regression where the response variable is the gene expression in cells and the independent variables are the $K+1$ B-spline bases).

The observed data of the gene are assumed to be generated from this unobserved function after adding cell-level random noise $\epsilon_{sc}$ as follows:

$$
\begin{aligned}
y_{sc} &= Y_s(t_{sc}) + \epsilon_{sc} \\
&= \sum_{k=0}^{K} \phi_k(t_{sc})b_{sk} + \epsilon_{sc} \\
&= \boldsymbol{\phi}(t_{sc})^T \mathbf{b}_s + \epsilon_{sc}
\end{aligned}
\tag{1}
$$

where

$$
\begin{aligned}
\boldsymbol{\phi}(t) &= \left[\phi_0(t), \phi_1(t), ..., \phi_K(t)\right]^T \\
\mathbf{b}_s &= \left[b_{s0}, b_{s1}, ..., b_{sK}\right]^T \\
\epsilon_{sc} &\sim N(0, \sigma_s^2)
\end{aligned}
\tag{2}
$$

Since all samples share the same B-spline bases $\phi(t)$, the sample-specific temporal pattern is described via the sample-specific regression coefficients $\mathbf{b}_s$. To model the relationship between a gene's

pseudotemporal pattern $Y_s(t)$ and sample covariates $\mathbf{x}_s$ while accounting for sample-to-sample variability that cannot be explained by the covariates, we further assume

$$\mathbf{b}_s = \begin{bmatrix} b_{s0} \\ b_{s1} \\ \vdots \\ b_{sK} \end{bmatrix} = \begin{bmatrix} \beta_{00} & \beta_{01} & \cdots & \beta_{0V} \\ \beta_{10} & \beta_{11} & \cdots & \beta_{1V} \\ \vdots & \vdots & \vdots & \vdots \\ \beta_{K0} & \beta_{K1} & \cdots & \beta_{KV} \end{bmatrix} \begin{bmatrix} 1 \\ x_{s1} \\ \vdots \\ x_{sV} \end{bmatrix} + \begin{bmatrix} u_{s0} \\ u_{s1} \\ \vdots \\ u_{sK} \end{bmatrix} = \mathbf{B}\mathbf{x}_s + \mathbf{u}_s \tag{3}$$

where $\mathbf{B}$ is a $(K+1) \times (V+1)$ matrix representing unknown fixed effects of covariates, and $\mathbf{u}_s$ is a $(K+1) \times 1$ vector representing unobserved sample-level random effects (i.e. random variations among samples with the same covariate values):

$$\mathbf{u}_s \sim N(\mathbf{0}, \sigma_s^2 \boldsymbol{\Omega}) \tag{4}$$

Here $\boldsymbol{\Omega}$ is a $(K+1) \times (K+1)$ positive definite matrix. Note that the degrees of freedom for estimating sample-level covariance matrix $\boldsymbol{\Omega}$ after accounting for $V+1$ covariates are $S-(V+1)$ and one needs at least $K+1$ degrees of freedom to estimate a full rank covariance matrix with dimension $K+1$. Therefore, if the sample size $S$ does not exceed $V+K+2$, we do not have enough information to estimate an unconstrained $\boldsymbol{\Omega}$. In that scenario, we add a constraint by assuming $\boldsymbol{\Omega} = \omega^2 \mathbf{I}_{(K+1)\times(K+1)}$ where $\mathbf{I}$ represents an identity matrix. This constraint reduces the number of parameters in $\boldsymbol{\Omega}$ to 1. Define

$$\boldsymbol{\beta}_{k.} = [\beta_{k0}, \beta_{k1}, \ldots, \beta_{kV}]^T$$
$$\boldsymbol{\beta}_{.v} = [\beta_{0v}, \beta_{1v}, \ldots, \beta_{Kv}]^T$$

$\boldsymbol{\beta}_{k.}^T$ is the $k$th row of $\mathbf{B}$, corresponding to regression coefficients for basis $\phi_k(t)$. $\boldsymbol{\beta}_{.v}$ is the $v^{th}$ column of $\mathbf{B}$, corresponding to regression coefficients for the $v$th covariate $X_v$. If gene $g$'s expression pattern does not depend on $X_v$, then $\boldsymbol{\beta}_{.v} = 0$.

To facilitate developing the model fitting algorithm, Eq. (3) can also be rewritten in a vectorized form. Let $\mathbf{I}_K$ be a $K \times K$ identity matrix, and

$$\mathbf{X}_s = \mathbf{I}_{K+1} \otimes \mathbf{x}_s^T = \begin{bmatrix} \mathbf{x}_s^T & \mathbf{0} & \cdots & \mathbf{0} \\ \mathbf{0} & \mathbf{x}_s^T & \cdots & \mathbf{0} \\ \vdots & \vdots & \vdots & \vdots \\ \mathbf{0} & \mathbf{0} & \cdots & \mathbf{x}_s^T \end{bmatrix}_{(K+1)\times[(K+1)(V+1)]}$$
$$\boldsymbol{\beta} = \left[\boldsymbol{\beta}_{0.}^T, \boldsymbol{\beta}_{1.}^T, \ldots, \boldsymbol{\beta}_{K.}^T\right]^T = [\beta_{00}, \ldots, \beta_{0V}, \beta_{10}, \ldots, \beta_{1V}, \ldots, \beta_{K0}, \ldots, \beta_{KV}]^T \tag{5}$$

Then Eq. (3) can also be written as:

$$\mathbf{b}_s = \mathbf{X}_s \boldsymbol{\beta} + \mathbf{u}_s \tag{6}$$

Thus, the observed data model in Equation (1) is equal to

$$\begin{aligned} y_{sc} &= \boldsymbol{\phi}(t_{sc})^T \mathbf{b}_s + \epsilon_{sc} \\ &= \boldsymbol{\phi}(t_{sc})^T (\mathbf{B}\mathbf{x}_s + \mathbf{u}_s) + \epsilon_{sc} \\ &= \boldsymbol{\phi}(t_{sc})^T (\mathbf{X}_s \boldsymbol{\beta} + \mathbf{u}_s) + \epsilon_{sc} \end{aligned} \tag{7}$$

where $\epsilon_{sc} \sim N(0, \sigma_s^2)$ and $\mathbf{u}_s \sim N(\mathbf{0}, \sigma_s^2 \boldsymbol{\Omega})$. We further assume that $\sigma_s^2$ follows an inverse-Gamma distribution:

$$\sigma_s^2 \sim IG(\alpha, \eta) \tag{8}$$

For the given gene, let $\mathbf{y}_s = [y_{s1}, \ldots, y_{sC_s}]^T$ denote its expression in all cells in sample $s$, $\boldsymbol{\epsilon}_s = [\epsilon_{s1}, \ldots, \epsilon_{sC_s}]^T$, and $\boldsymbol{\Phi}_s = [\boldsymbol{\phi}(t_{s1}), \ldots, \boldsymbol{\phi}(t_{sC_s})]^T$, then Eq. (7) can also be written in a matrix form as:

$$\begin{aligned} \mathbf{y}_s &= \boldsymbol{\Phi}_s (\mathbf{B}\mathbf{x}_s + \mathbf{u}_s) + \boldsymbol{\epsilon}_s \\ &= \boldsymbol{\Phi}_s (\mathbf{X}_s \boldsymbol{\beta} + \mathbf{u}_s) + \boldsymbol{\epsilon}_s \end{aligned} \tag{9}$$

The above model can be fit using an Expectation-Maximization (EM) algorithm (see details in the Supplementary Notes). The algorithm can estimate the unknown parameters $\boldsymbol{\Theta} = \{\boldsymbol{\beta}, \boldsymbol{\Omega}, \alpha, \eta\}$ and infer $\sigma_s^2$ based on the observed data. Here $\sigma_s^2, \alpha, \eta \in \mathbb{R}$, $\boldsymbol{\Omega} \in \mathbb{R}^{(K+1)\times(K+1)}$, $\boldsymbol{\beta} \in \mathbb{R}^{(K+1)(V+1)}$.

## Detecting differential expression associated with sample covariate (XDE)

Under the Lamian model, detecting differential expression associated with a sample covariate $X_v$ amounts to testing whether $\boldsymbol{\beta}_{.v} = [\beta_{0v}, \beta_{1v}, \ldots, \beta_{Kv}]^T = \mathbf{0}$. An XDE gene is a gene with $\boldsymbol{\beta}_{.v} \neq \mathbf{0}$. For an XDE gene, if $\beta_{0v} = \beta_{1v} = \ldots = \beta_{Kv} = c$ (i.e. all $\beta_{kv}$s are equal), then the effect of the covariate is to shift the gene's pseudotemporal curve up or down by a constant $c$ for every unit change in $X_v$ (because the B-spline bases satisfy $\sum_{k=0}^{K} \phi_k(t) = 1$). Such a gene is called XDE with mean shift only. If $\beta_{kv}$s are not all equal for an XDE gene, then the covariate also changes the trend of the gene's pseudotemproal curve. To systematically detect and classify XDE genes, we consider the following nested models:

- $M_0$: $\boldsymbol{\beta}_{.v} = [\beta_{0v}, \beta_{1v}, \ldots, \beta_{Kv}]^T = \mathbf{0}$.
- $M_1$: $\boldsymbol{\beta}_{.v} \neq \mathbf{0}$ and $\beta_{0v} = \beta_{1v} = \ldots = \beta_{Kv} = c$.
- $M_2$: $\boldsymbol{\beta}_{.v} \neq \mathbf{0}$.

We conduct the following hypothesis tests:
- *Overall XDE test*: the null model $M_0$ is compared with the alternative model $M_2$. Rejecting $M_0$ implies XDE.
- *Mean test*: $M_0$ and $M_1$ are compared. Rejecting $M_0$ implies mean shift.
- *Trend test*: $M_1$ and $M_2$ are compared. Rejecting $M_1$ implies trend difference.

A gene is called XDE if the XDE test is significant. For an XDE gene, if the mean test is significant but the trend test is not significant, the gene is called XDE with mean shift only. If the trend test is significant but the mean test is not, then the XDE gene is called XDE with trend difference only. If both the mean test and the trend tests are significant, then the XDE gene is called XDE with both mean shift and trend difference.

To conduct a hypothesis test comparing two models, we use a permutation-based likelihood ratio test. Without loss of generality, consider comparing null model $M_0$ versus alternative model $M_1$ as an example (other model comparisons are handled similarly). The test statistic is the log-likelihood ratio (LLR) between $M_1$ and $M_0$ computed using the observed data. To construct the null distribution of the test statistics, we use a permutation approach. In each permutation, we first bootstrap the cells (keeping cell number the same as the observed data) to account for the pseudotime variability, and then we permute the values of the covariate $X_v$ among the samples. Using the permuted data, the models are refit and the LLR statistic is recomputed. Using the LLR obtained from all permutations (by default, 100 times), an empirical distribution is fitted using kernel density estimate (base::density()) to serve as the null distribution. The $p$-value is calculated as the tail probability of the null distribution (i.e. probability that a LLR drawn from the null distribution is equal or larger than the observed LLR). The $p$-values from all genes are adjusted for multiple testing using the Benjamini-Hochberg procedure to obtain false

discovery rates (FDRs)[31]. By default, $FDR \leq 0.05$ is used as the significance cutoff.

Besides permutation test, we also provide an option to compute $p$-values and FDR based on the asymptotic null distribution, that is, chi-squared distribution, of the likelihood-ratio test (`stat::pchisq()`). The degree of freedom is the difference in the number of parameters between the full and null model. This option can be used if users need computational efficiency and are willing to sacrifice some accuracy to control FDR.

## Adjusting for confounding variables such as batch effects
Since `Lamian` uses a general regression framework, one can adjust for confounding variables such as batch effects by properly specifying the design matrix. The design matrix $\mathbf{x}$ can contain multiple columns corresponding to multiple sample covariates. For example, given eight samples (4 males and 4 females) sequenced in three batches, the design matrix for XDE can be specified as

$$\mathbf{x} = \begin{bmatrix} 1 \\ \mathbf{x}_1 \\ \vdots \\ \mathbf{x}_s \\ \vdots \\ \mathbf{x}_S \end{bmatrix} = \begin{bmatrix} 1 & 1 & 0 & 0 \\ 1 & 1 & 0 & 0 \\ 1 & 1 & 1 & 0 \\ 1 & 1 & 0 & 1 \\ 1 & 0 & 0 & 0 \\ 1 & 0 & 1 & 0 \\ 1 & 0 & 0 & 1 \\ 1 & 0 & 0 & 1 \end{bmatrix}$$

Here each row corresponds a sample. The first column is the intercept. The second column represents samples' sex (1 for female, 0 for male). The third and fourth columns are dummy variables to indicate batches. Suppose one is interested in detecting XDE genes associated with sex, one will only use the regression coefficients for the sex variable to identify differential genes. The batch effects are accounted for by columns 3 and 4.

## Detecting differential expression along pseudotime (TDE)
Unlike `Lamian`, most existing pseudotime methods do not detect differential expression associated with covariates (XDE). Instead, they detect differential expression along pseudotime (TDE). While our main focus is to detect XDE genes, `Lamian` also provides a function to detect TDE genes.

When all samples are from one group without covariate, the Equation (3) becomes

$$\mathbf{b}_s = \begin{bmatrix} \beta_{00} \\ \beta_{10} \\ \vdots \\ \beta_{K0} \end{bmatrix} + \mathbf{u}_s \tag{10}$$

Note that $\sum_{k=0}^{K} \phi_k(t) = 1$. Thus, if $\beta_{00} = \beta_{10} = ... = \beta_{K0} = c$ (i.e. all $\beta_{k0}$s are equal), then the pseudotemporal pattern shared by samples is $\phi(t)^T \boldsymbol{\beta}_{\cdot 0} = c$, which is a constant that does not change along pseudotime. Therefore, TDE detection can be formulated as comparing the following two models:

- $H_0$: $\beta_{k0}$ $(k = 0, 1, ..., K)$ are all equal
- $H_1$: $\beta_{k0}$ $(k = 0, 1, ..., K)$ are not necessarily all equal

This yields the following hypothesis test:
- *TDE test*: $H_0$ and $H_1$ are compared. Rejecting $H_0$ implies differential expression along pseudotime (TDE).

The TDE test can also be generalized to account for sample covariates. With covariates, the compared models become:

- $H_0$: $\beta_{kv}$ $(k = 0, 1, ..., K)$ within each column of $\mathbf{B}$ in Equation (3) are equal (i.e. $\boldsymbol{\beta}_{\cdot v} = c_v \mathbf{1}$ where $v = 0, 1, ..., V$ and $\mathbf{1}$ represents a $K + 1$ vector with all elements equal to 1)
- $H_1$: No constraint on $\mathbf{B}$

The hypothesis test is conducted using a permutation-based likelihood ratio test. We first compute the log-likelihood ratio (LLR) between $H_1$ and $H_0$ as the test statistic using observed data. We then construct the null distribution of LLR using permutations. In each permutation, we first bootstrap the cells to account for pseudotime variability, and we then permute the pseudotime of the cells within each sample. Using the permuted data, the models are refit and the LLR statistic is recomputed. The null distribution is derived by applying the kernal density estimate (`base::density()`) to the empirical LLR statistics obtained from all permutations (by default, for 100 times). $P$-value is calculated as the tail probability of the empirical distribution. The $p$-values from all genes are adjusted for multiple testing using the Benjamini-Hochberg procedure to obtain FDR[31]. By default, $FDR \leq 0.05$ is used as the significance cutoff.

## EM algorithm for fitting the Lamian model
The algorithm used to fit the `Lamian` model is provided in Supplementary Notes in detail.

## Analysis of cell density changes
Given a pseudotemporal path or branch, we divide the pseudotime from 0 to its maximum into 100 consecutive intervals of equal lengths. The number of cells in each interval $t$ and sample $s$ is counted and denoted as $r_{st}$. One approach to modeling cell density changes is to model $r_{st}$ using a count distribution (e.g. Poisson or Negative binomial) with mean $L_s \lambda_{st}$ where $L_s$ is a sample-specific normalizing constant corresponding to the total cell number on the pseudotemporal path. One can then model $\log \lambda_{st}$ as functional curves using B-spline bases similar to the gene expression model. Fitting such a model, however, requires algorithms such as Markov Chain Monte Carlo which makes this approach less appealing computationally. We therefore use an alternative and simpler approach in which $r_{st}/L_s$ is modeled in the same way as the gene expression model in equation (1) (i.e. treating time interval $t$ as cell and treating $r_{st}/L_s$ in the same way as $y_{sc}$). In this way, testing if the cell density changes along pseudotime (*TCD test*) or if a sample covariate changes the pseudotemporal cell density curves (*XCD test*) can be handled following the same procedure for TDE and XDE tests. This approach is more computationally efficient and yields reasonable results empirically in our benchmark data.

## Comparisons with existing methods
**XDE detection.** For detecting differential expression associated with covariates, we compared `Lamian` with `tradeSeq`[27] (v.1.1.23), `limma`[42] (v.3.40.6), `monocle2`[21] (v2.14.0), `PhenoPath`[28] (v1.8.0), and `condiments`[29] (v.0.99.14). We applied `tradeSeq` by considering the cells belonging to two groups as those belonging to two lineages. The cell weights on each group were set as 0.99 and 0.01 respectively. We then fit the models by running the fitGAM() function with the default setting. All three types of tests for between-lineage comparisons were included. Specifically, earlyDETest(), diffEndTest() and patternTest() were applied to identify early drivers of differentiation, differentiated markers and expression patterns over pseudotime, respectively. `limma` was applied by pooling each sample as a pseudobulk. Its functions lmFit(), eBayes(), and topTable() were used to perform the test. `Monocle2` provides a trajectory-conditioned test 'monocle2TrajTest' which compares a full model $g(E(Y)) \sim \beta_0 + f_1(G) + f_2(\phi)$ with a null model $g(E(Y)) \sim \beta_0 + f_1(G)$ (see page 23 of the Monocle2 supplemental material[21]). Here $E(Y)$ is the expected values of the transcript counts $Y$, $g(\cdot)$ is the log function, $G$ indicates sample group (note: in the original `Monocle2` paper $G$ refers to genotype), $\phi$ is pseudotime, $f_1(G)$ models

the different intercepts for different groups (note: $f_1(G)$ does not involve pseudotime $\phi$), and $f_2(\phi)$ is a non-parametric function that models gene expression as a function of pseudotime $\phi$. The 'monocle2TrajTest' evaluates whether $f_2(\phi)$ is zero and it assumes that $f_2(\phi)$ is the same for different sample groups $G$. Therefore, the trajectory-conditioned test 'monocle2TrajTest' is essentially a TDE test and does not detect XDE. To detect XDE, we modified the build-in functions in `Monocle2`, resulting in a revised test which we call 'monocle2TrajTestCorr'. In our revised test, the null model is $g(E(Y)) \sim \beta_0 + f_2(\phi)$, and the full model is $g(E(Y)) \sim \beta_0 + f_1(G) + f_2(\phi) + f_2(\phi)f_1(G)$. Comparing these two models will test whether $f_1(G) + f_2(\phi)f_1(G)$ (i.e., the difference between different sample groups including both intercept and pseudotemporal trend differences) is zero. To run `PhenoPath`, we provided sample covariate information as a design matrix where the first column is the phenotype (1 or 0, indicating the two sample groups), and the second to $S$-th columns are $S-1$ dummy variables to indicate the $S$ samples. $P$-values are $1$ – the tail probability of the test statistics' $z$-scores (m_beta divided by s_beta) which are assumed to be standard normal. `condiments` was run based on its user manual. The XDE in `condiments` was implemented using the conditionTest() in the `tradeSeq` package.

**TDE detection.** For detecting differential expression along pseudotime, we compared `Lamian` with `Monocle2` (v.2.14.0), `Monocle3`[22] (v.3.0.2.1), `tradeSeq`[27] (v.1.1.23) and `TSCAN`[23] (v.1.7.0). All methods other than `Lamian` treat cells from all samples as if they were from one sample. `Monocle2` performs the testing with an approximate $\chi^2$ likelihood ratio test. In this test, generalized additive models (GAMs) are applied to fit the gene expression against pseudotime as a full model, while the null model considers the gene expression as a constant along pseudotime. `Monocle3` performs trajectory inference on the coordinates from UMAP and then implements the Moran's I test to identify genes whose expression is associated with pseudotime with statistical significance. `TSCAN` applies the same fitting and testing method as `Monocle2` except that `TSCAN` uses `MGCV` package and `Monocle2` applies `VGAM` package. `tradeSeq` is used by `Slingshot`[24] to identify dynamic genes along pseudotime. Both tests designed for within-lineage comparisons in `tradeSeq` were included (startVsEndTest() and associationTest()). We also tried `pseudotimeDE`, but it did not output results within one week and with 400GB and 20 CPU cores.

**Significance cutoff**
All $p$-values are provided as exact values except for situations where the $p$-value computation reaches the computer's precision lower bound (i.e. the smallest value allowed by the precision). In that case, $p$-values are reported as smaller than the precision lower bound. All $p$-values reported by each method were adjusted for multiple-testing using the Benjamini-Hochberg procedure to obtain false discovery rates (FDRs)[31]. By default, FDR ≤ 0.05 is used as the significance cutoff.

**Simulations**
**XDE detection.** We first created null simulation data where we do not expect any XDE genes. The simulation was based on the 13,269 cells on the erythroid branch in the real HCA-BM data described above. For the null simulation in Fig. 4a, the eight bone marrow scRNA-seqsamples were randomly partitioned into two groups (group 0 and 1). Next, to remove any group differences for a given gene, we divided the pseudotime into 100 non-overlapping intervals of equal lengths. Within each interval and within each sample group, we calculated the median of the gene's normalized expression. For cells in the sample group with lower median value, we added their expression with the difference of median expression between the two groups so that the two groups have similar expression values.

Building upon the null dataset above, we then introduced in silico spike-in differential signals with varying strengths and pseudotemporal

patterns between the two sample groups to a random set of genes. This spike-in simulation data set was used in Fig. 4b–h. We randomly selected 20% (1814) genes as the gold standard XDE genes (gs genes) and randomly assigned them to 3 groups: trend difference only, mean shift only, and both trend & mean differences. We then spiked in differential trend, mean, or both trend & mean signals into these gold standard genes based on their differential type. To generate the spike-in signals, we selected highly variable genes from the remaining 80% non-gold-standard (non-gs) genes using cells in sample group 0 and using their original unpermuted data. To select highly variable genes, we applied B-splines to fit the relationship between the standard deviation (SD) and the mean of gene expression of the non-gold-standard (non-gs) genes across cells in group 0. Genes with positive residuals (i.e. SD is larger than its expected value estimated from the mean expression) are selected as highly variable. We applied $k$-means clustering to cluster these genes into 5 clusters using their standardized $\log_2$-transformed SAVER-imputed expression. Louvain and Gaussian mixture model clustering have also been separately applied to examine the sensitivity to clustering methods. Here the cluster number 5 was determined using an elbow method. For each gene that was clustered, we fit a B-spline on the $\log_2$-transformed SAVER-imputed expression against pseudotime. We evaluated the magnitude of change of the gene along pseudotime by calculating a $F$-statistic that compares a full model (which assumes gene expression along pseudotime is modeled using the B-spline curve plus additive noise) and a null model (which assumes gene expression along pseudotime is a constant plus additive noise). We used highly variable genes (i.e. those with positive residuals) as "source genes". We ordered source genes in increasing $F$-statistics. We categorized the tail 1814 source genes into 4 groups from the smallest to the largest $F$-statistics to represent signal strengths from weakest (1) to highest (4). In each signal-strength simulation, we added the gene expression profiles in each sample from the source genes in the same strength group onto those gold standard genes. The signal-spike-in procedures were performed in SAVER-imputed gene expression matrix and original count matrix in parallel. For gold standard genes with trend difference, we added signals to both group 0 and 1, except that the signals were permuted before adding to group 1. For gold standard genes with mean shift, we permuted the source gene expression profiles within each sample before adding signals to group 0. For gold standard genes with both trend and mean differences, we added source signals directly to group 0 cells without centering the data.

**TDE, TCD, and XCD detection.** Simulations for evaluating TDE, TCD and XCD detection are presented in Supplementary Notes.

**Evaluation**
**Evaluation in simulation.** Performance of `Lamian` (Lamian.pm), Lamian.chisq and other existing methods is compared based on FDR difference and AUC. FDR difference is the difference between the area under the realFDR vs. reportedFDR curve and the diagonal line. The differences between the true and reported FDRs were calculated for overall XDE test, trend test and mean test. Within each set of gold-standard genes (trend, mean, and trend & mean), the area under sensitivity-realFDR curve (AUC) was also calculated.

**Evaluation on sex chromosomes.** To evaluate overlap between XDE genes reported by different methods and sex chromosome genes as gold standard (see Fig. S12a), we counted the overlap (i.e. the number of sex chromosome genes) among the top $N$ XDE genes for different $N$s. The mean of the overlap across all $N$s was used as the observed overlap statistic. Null distribution was constructed by permuting the order of the genes which originally were ordered by increasing FDR. Violin plots show the permutation null distribution used to determine the statistical significance

of the observed overlap statistics (dots), and *p*-values are shown on the right of each plot.

## Visualization

Each heatmap to visualize XDE test results is organized in four white-bar-separated panels to show the expression patterns of XDE genes (rows) by cells (columns) ordered by pseudotime. The 1st and 2nd panels show original values and model-fitted values of gene expression. Cells from the samples in group 0 and 1 are separated. The 3rd and 4th panels show the standardized model-fitted group difference (trend difference) and the mean shift between groups, where white space denotes no significant difference.

## Data availability

The data used in this manuscript are all downloaded from publicly available data sources. Specifically, HCA-BM data were downloaded from HCA data portal (immune cell atlas of human hematopoietic system)[37,38] [https://data.humancellatlas.org/explore/projects/cc95ff89-2e68-4a08-a234-480eca21ce79]. COVID-19 data were downloaded from the ArrayExpress database under accession code E-MTAB-9357 [https://www.ebi.ac.uk/arrayexpress/experiments/E-MTAB-9357][43], and TB data were downloaded from the Gene Expression Omnibus (GEO) database under accession code GSE158769 [https://www.ncbi.nlm.nih.gov/geo/query/acc.cgi?acc=GSE158769][50]. All relevant information about data is described in the Methods section. All processed data generated in this study are provided in the Supplementary Information/Source Data file. Source data are provided with this paper and in Zenodo under accession code https://doi.org/10.5281/zenodo.8274409[https://zenodo.org/record/8274409][55]. Source data are provided with this paper.

## Code availability

The `Lamian` package (v.0.99.1) is provided as an open-source software package with a detailed user manual available at https://github.com/Winnie09/Lamian. All codes to reproduce the presented analyses are publicly available in Github repository https://github.com/Winnie09/trajectory_variability and also in Zenodo under the accession code DOI: 10.5281/zenodo.8197779 [https://zenodo.org/record/8197779][56]. R version 4.0.2, `topGO` (v.2.42.0)[57], and `ComplexHeatmap` (v.2.6.2)[58] were used to perform the analyses in the manuscript. The R package `ggplot2` (v.3.3.0)[59] for data visualization was used. All competing methods are described in Table S1. BioRender (BioRender.com) was used for generating part of Fig. 1a under a paid subscription, and the publication agreement number is FS25TOPP8E.

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

## Acknowledgements

This work is supported by the National Institutes of Health grants R01HG010889 and R01HG009518 to HJ, R00HG009007 to SCH, and K99HG011468 and R00HG011468 to WH. SCH is also supported by CZF2019-002443 from the Chan Zuckerberg Initiative DAF, an advised fund of Silicon Valley Community Foundation. We would like to thank the Maryland Advanced Research Computing Center (MARCC) and Rockfish systems, and The Joint High Performance Computing Exchange (JHPCE) for providing computing resources.

## Author contributions

W.H., Z.J., S.C.H. and H.J. conceived the study. H.J., S.C.H. and W.H. conceptualized the Lamian framework. W.H. and H.J. developed the statistical model and algorithm with feedback from Z.J. and S.C.H. W.H. implemented the model and software. Z.J., W.H., and Z.C. prepared the data. W.H. and Z.J. analyzed the data. W.H., Z.C., E.J.W., S.C.H. and H.J. interpreted the results. W.H., S.C.H. and H.J. drafted the manuscript. All authors edited and approved the final manuscript.

## Competing interests

E.J.W. is a member of the Parker Institute for Cancer Immunotherapy that supports research in the Wherry lab. E.J.W. is an advisor for Danger Bio, Marengo, Janssen, NewLimit, Pluto Immunotherapeutics Related Sciences, Santa Ana Bio, and Synthekine. E.J.W. is a founder of and holds stock in Arsenal Biosciences and holds stock in Coherus. Other authors declare no competing interests.
