## [Transparent Peer Review File · Nature Communications]

A statistical framework for differential pseudotime analysis with multiple single-cell RNA-seq samplesEditorial Note: Parts have been redacted where manuscript number has been mentioned

Reviewer Comments, First Version:

Reviewer #1 (Remarks to the Author: Overall significance):

In this paper, the authors propose a statistical framework, Lamian, for multi-sample pseudotime differential analysis. Lamian includes the following major modules:

1. quantifying the uncertainty of the tree branches by bootstrap;
2. differential tree topology between two conditions by comparing branch proportions (defined by numbers of cells assigned to branches);
3. a gene's differential expression (DE) analysis along a pseudotemporal trajectory / a branch, including TDE (if a gene has expression changes along the trajectory) or XDE (if a gene has the same expression dynamics between two conditions);
4. differential cell density (CD) analysis along a pseudotemporal trajectory / a branch (TCD and XCD).

The authors applied Lamian to two real scRNA-seq datasets to demonstrate its effectiveness in applications.

Overall, I think this article is well-written and the Lamian framework is easy to understand. One key novelty is that the authors used the mixed-effect like models to address multi-sample variation. However, I have the following questions and suggestions about the statistical methods used in Lamian.

1. The Lamian relies on TSCAN for pseudotime inference. This leads to two questions. First, can Lamian be used as a downstream pipeline for other popular pseudotime inference methods, e.g., Slingshot and Monocle3? Second, for the comparison with two existing DE methods (limma and tradeSeq), do all DE methods use the same input pseudotime? If not, is it possible that the quality of pseudotime inference affects the DE performance so that the comparison is not completely fair?
2. A drawback of Lamian is that it does not account for the uncertainty in pseudotime inference because it only applies the bootstrap to the pseudotime values of all cells. In other words, the bootstrap only captures the randomness in sampling cells, not the randomness in the inference pseudotime values of all cells.
3. Because of 2, the permutations used in the likelihood-ratio tests (LRTs) for XDE and TDE are not necessary because the null distribution only captures the randomness in sampling cells. Hence, the authors should be able to use the theoretical, asymptotic null distribution of the LRT statistic and greatly reduce the computational time required by permutations.
4. A computational issue related to 2 is that the authors used the kernel density estimate (KDE) of the permuted LRT statistic values as the null distribution. However, KDE relies on a pre-specified kernel bandwidth, which is automatically chosen by the R function for every gene but not necessarily appropriate for some genes. This computational instability should be avoided by replacing the permutation approach by the theoretical null distribution as I suggested in 2.
5. Module 2 uses the two-sample t test to compare branch proportions between two conditions. The t test is inappropriate for two reasons: (1) branch proportions are not normally distributed; (2) proportions of different branches are not i.i.d.
6. A limitation of Module 2 is: what if two conditions have different topologies, e.g., the two conditions have difference single branches. In this case, how can Lamian tell that the two single branches are not the same?
7. Figure 4h suggests that the simulation for power comparison and FDR control was not realistic. The ROC curve is too perfect. I suggest that the authors add a more realistic simulation study for benchmarking Lamian against limma and tradeSeq.

8. In Methods, why do equations (2) and (4) share the same variance σ_s^2 ? Equation (4) does not seem to need σ_s^2 .

9. Table S1 listed several existing methods, but these methods were not cited in the main text. The authors should mention these methods in the main text and add their citations.

Minor:

1. A right parenthesis in line 27 on page 7 should be removed.

Reviewer #1 (Remarks to the Author: Impact):

Reviewer #1 (Remarks to the Author: Strength of the claims):

Please see my comments under the overall significance.

Reviewer #1 (Remarks to the Author: Reproducibility):

I didn't run the code, but the writing was overall clear.

Reviewer #2 (Remarks to the Author: Overall significance):

In this work, Wenpin Hou, et. al. reported a statistical framework, Lamian, that builds on top of other approaches to perform comprehensive differential pseudotime analyses of scRNA-seq datasets with consideration of multiple samples. In particular, Lamina deal with multi-sample datasets and is able to identify changes in differential genes along pseudotemporal trajectory that are related to various sample covariates, as well as to detect the associated changes in gene expression, cell density, and topology.

Reviewer #2 (Remarks to the Author: Impact):

scientific reports.

I don't think this will influence thinking in the field

Reviewer #2 (Remarks to the Author: Strength of the claims):

Major concern:

The authors argue that comparing pseudotemporal dynamics across multiple samples is lacking and claim lamina is one of the first model to do so. This claim, however, is rather weak and should be tuned down or backed up much better. The author should do a more inclusive review or summary of existing tools for multi-sample pseudotime analysis to support this claim (for example, providing a supplementary table to summarize existing pseudotime analyses tools and how lamina fills this gap). For example, Kieran R Campbell (<https://www.nature.com/articles/s41467-018-04696-6>) has published papers on differential pseudotime analyses with covariates. In addition, the trajectory-conditioned test from Monocle 2 (Fig 2 of <https://www.ncbi.nlm.nih.gov/pmc/articles/PMC5764547/>) performs differential analyses to identify genes that change under different genetic backgrounds (samples) along pseudotime trajectory. Fundamentally, inclusion of the sample covariate when performing a generalized linear regression is a rather conventional practice in biostatistics and thus the novelty of this approach is limited.

Furthermore, different samples may lead to strong batch effects that are not biologically relevant. Thus, it is important to discern the real biological difference and the technical differences between different samples. The author relies on existing approaches to remove batches before downstream differential analyses but it is worth discussing this issue. Another possibility is to avoid the batch effect issue by focusing on analyzing datasets that are generated from the same batch but still involve different samples (for example, including drug treatment, genetic perturbation, etc.).

Minor comments:

P1: Line 20, the author mentioned spatial dynamics, however the entire paper doesn't really touch analyses of this type of data and I would suggest removing this to avoid spatial transcriptomics bandwagon.

P2. The first step of Lamian uses TSCAN and bootstrapping for pseudotime and batch certainty detection. There are obviously many superb alternatives, including principal graph, diffusion pseudotime, etc., than TSCAN for pseudotime analysis and it is important to explain why this choice was used (even if this is something that is related to your lab's previous research). In addition, Ruslan Soldatov, et.al previously reported using bootstrapping of the developmental tree construction with simplePPT to estimate the certainty of the tree structure. It may be interesting to compare your approach with this method (https://www.science.org/doi/10.1126/science.aas9536?url_ver=Z39.88-2003&rfr_id=ori:rid:crossref.org&rfr_dat=cr_pub 0pubmed)

P3. Some information about the schematic is not very clear. For example, what is the meaning of 0.3 (0.2) for example, in the section related to cell proportion in a branch of the panel b (mean and s.d.?) Similarly, simple explanations of the meaning of the equations in panel C needs to be spelled out in the schematic.

P4. Overall, the author needs to justify why the HCA-BM dataset was used for the analyses and why comparing the male and female is meaningful and interesting at all. For example, the analysis in 2.2.2 seems quite strange because you won't ever expect the hematopoiesis to differ across men and women.

In 2.2.3, the author shows that decreasing the number of cells decreased the detection rate for myeloid branch.

It will be interesting to check whether this low detection rate is biologically relevant or whether it is purely driven by uneven capture of cell number for the myeloid lineage.

P5. Following gene short name convention, the author may need to go through the paper carefully to italicize all gene names (for example CD14 to *CD14*, etc.). Importantly, the example genes shown here are less informative as most of them are CD markers which are reported previously based on FACS data (protein level). It will be more relevant to showcase known transcription factors or other markers which have a known high RNA expression level for each particular lineage.

Lastly, some of the enrichment analysis doesn't make the perfect sense, for example, why "as HSCs differentiate to the erythroid lineage, the TDE genes with initially 4 high expression but low expression at the end are enriched in CD8-positive, alpha-beta T cell activation"? What we expect is that those genes should be enriched in HSC stemness maintenance related pathways. Similarly the "platelet degranulation" doesn't make a lot of sense.

P7. The number of the XDE genes detected are very few and mostly related to the X/Y-chromosome, can you find any meaningful gene that are related to hematopoiesis while also have sex-difference. Basically I wonder whether what you identified are simply genes that differ between male and female but have nothing to do with hematopoiesis. This boils down again to whether it is meaningful to even perform the differential analyses between different sex for the hematopoiesis.

The author didn't obtain any biological insights with the cell density analyses. Can the author find a dataset to demonstrate the relevance of this approach?

P9. The authors try to provide some biological insights in the final section on SARS-Covid2 infection but their conclusions are too tentative which provide little advance the Covid-19 biology and it is not clear whether their work on the subject is to drive the Covid bandwagon to their method.

Reviewer #2 (Remarks to the Author: Reproducibility):

great as it provided an associated package

Reviewer #3 (Remarks to the Author: Overall significance):

Hou et al presented a comprehensive statistical framework Lamian for pseudo-time analysis of single cell RNA-seq data from multiple samples with various conditions. The main functions of Lamian include preprocessing, tree structure inference, evaluating topology difference, identifying differential gene along pseudotime or between different conditions. They used both simulations and real scRNAseq data, including COVID-19 patients with different disease severity levels, to demonstrate the advantages of Lamian over competing software such as tradeSeq. The paper is well written and the statistical model is rigorous. It adds novel tools to the crowded trajectory analysis field, which mainly focuses on single sample analysis. I have the following comments to strengthen the paper.

Major comments

1. The authors described the input for Lamian includes a low-dimensional space representation of scRNA-seq data from multiple samples after batch effect correction (page 2, line 18-20), and listed candidate methods such as Seurat and Harmony. However, in the paper the authors conducted all analyses using Seurat approach (Section 4.2, page14, line 20), but didn't try to illustrate Lamian's performance with input from alternative methods, for example Harmony. It is important to provide some practical guidance to users about which batch effect methods to use under different conditions (e.g., different tissue types), and how the result differs. Another popular batch effect method that should fit Lamian is scVI (<https://www.nature.com/articles/s41592-018-0229-2>). I suggest the authors try both Harmony and scVI in addition to Seurat for this data processing step to see if the result differ much, and discuss more about it.
2. For HCA-BM data, the authors mentioned they identified 6 cell clusters after applying TSCAN to the harmonized bone marrow data. However, as a complex tissue type, people can usually identify 10-20 or even more cell subpopulations. Is it an over-simplification of the problem by setting the number of cell clusters as 6 here? In addition, can the authors further instruct if Lamian can accommodate a large number of cell clusters, which form a tree with a large number of branches?
3. As for computational burden, the authors mentioned Lamian is computationally tractable. However, with the example of HCA bone marrow dataset (32,819 cells and 8 samples), it requires 4.1 hours with 25 CPUs and 163 GB RAM (page 11, line 19-21), which doesn't seem much friendly to users, and cannot be handled with a laptop. Since Lamian is designed for multi-batch data, which could scale up to 100K or 1M cells coming from a larger number of samples, will Lamian still work? Some discussion on computational complexity is needed.
4. This paper lacks a comparison with a recent competing method called condiments (<https://www.biorxiv.org/content/10.1101/2021.03.09.433671v1>).
5. The authors showed "decreasing the number of cells decreased the detection rate". This might be a limitation: as different datasets may vary in cell numbers, how can we believe the detection rate would be high enough to be different from random testing, especially when the dataset has limited number of cells?
6. In the validation of XDE test (P7), the author mentioned the identified XDE genes are enriched in sex chromosomes, as the argument to show their method can accurately captured the DEGs between

male and female. Did the author check whether the identified XDE genes on ChrX are the known genes escaping X-inactivation? If yes, this can further support the authors' results.

7. XCD test is not very attractive in my opinion, just as the author mentioned "the change can be due to technical sampling bias or real biology". In addition, the changes in cell cycle status or proliferation rate may be a major reason leading to the changes in cell density along pseudotime. Can the authors check whether there are any relationships between their detected changes in cell density and the distribution of cell cycle status along pseudotime?

Minor comments

1. In Section 2.1.3, the authors mentioned unsupervised k-means clustering is applied to DE genes (page 4, line 1). Does the result differ much if other clustering methods are used, such as Louvain or GMM?

2. In Section 4.1 Data (page 14, line 5-9), the authors stated the raw HCA-BM data consist of 290,861 cells, but after screening, the final data for analysis consist of only 32,819 cells, which means nearly 90% of cells are screened out. The authors should justify why such a strict screening is necessary and provide descriptive plots to illustrate the choice of 5,000 reads, 1,000 expressed genes and 10% mitochondrial gene expression are reasonable thresholds.

3. It is unclear what Lamian stands for as an abbreviation. Some explanation is needed. Both Lamian (majority) and Lamain were used in the text.

Reviewer #3 (Remarks to the Author: Impact):

Nature Methods is appropriate for the manuscript given the novelty of the statistical approach, the broad application of the paper, and convincing biological examples.

Reviewer #3 (Remarks to the Author: Reproducibility):

All data are publicly available. Code has been deposited to GitHub.

Reviewer reports:

Reviewer #1:

In this paper, the authors propose a statistical framework, Lamian, for multi-sample pseudotime differential analysis. Lamian includes the following major modules:

- 1. Quantifying the uncertainty of the tree branches by bootstrap;*
- 2. Differential tree topology between two conditions by comparing branch proportions (defined by numbers of cells assigned to branches);*
- 3. A gene's differential expression (DE) analysis along a pseudotemporal trajectory / a branch, including TDE (if a gene has expression changes along the trajectory) or XDE (if a gene has the same expression dynamics between two conditions);*
- 4. Differential cell density (CD) analysis along a pseudotemporal trajectory / a branch (TCD and XCD).*

The authors applied Lamian to two real scRNA-seq datasets to demonstrate its effectiveness in applications. Overall, I think this article is well-written and the Lamian framework is easy to understand. One key novelty is that the authors used the mixed-effect like models to address multi-sample variation. However, I have the following questions and suggestions about the statistical methods used in Lamian.

We are grateful for the reviewer's time and overall positive feedback. We have now addressed your questions and comments in this revision, which has greatly improved our manuscript.

1. The Lamian relies on TSCAN for pseudotime inference. This leads to two questions. First, can Lamian be used as a downstream pipeline for other popular pseudotime inference methods, e.g., Slingshot and Monocle3? Second, for the comparison with two existing DE methods (limma and tradeSeq), do all DE methods use the same input pseudotime? If not, is it possible that the quality of pseudotime inference affects the DE performance so that the comparison is not completely fair? [Editor: It is important to clarify these features for further consideration at Communications Biology.]

Regarding the first question, Lamian can be used as a downstream pipeline for other pseudotime inference methods to allow multi-sample differential gene expression and differential cell abundance analysis (i.e. modules 3 and 4). In order to do so, users only need to provide the pseudotime inferred from other methods as an input for Lamian, and they can then run modules 3 and 4. **For evaluating the uncertainty of tree topology and differential topology analysis (i.e. modules 1 and 2), Lamian currently is built upon TSCAN's cluster-based minimum spanning tree approach, and it does not support other pseudotime methods for two reasons.** First, the data structure for describing trajectory and the pseudotime reconstruction algorithm vary from method to method. Technically it is non-trivial to implement each algorithm from scratch. Second, it is also difficult to directly call the existing methods within Lamian or modify their code to integrate with Lamian since we do not have complete information about the data structure implemented in each method and meanings of each line of code. The modules 1 and 2 in Lamian are currently implemented using TSCAN since (1) TSCAN has competitive performance according to the recent benchmark studies, (2) the scalability of its cluster-based minimum spanning tree approach to large number of cells and its flexibility to support both automatic and manual trajectory construction, and (3) we have full control of its code and hence can modify its code to allow its built-in functions to be used by Lamian. In summary, **for the differential expression and cell abundance tests in modules 3 and 4, users can input any pseudotime inferred by their preferred pseudotime inference method**, and this is independent of modules 1 and 2. We clarify these in Section 2.1.1 of the manuscript, as follows:

"The advantages of using TSCAN to construct pseudotime include (i) the scalability of its cMST approach to large number of cells (since the number of tree nodes in the spanning tree is determined by cell cluster number instead of cell number) and repeated bootstrap resamplings, (ii) the flexibility it provides to support both automatic and manual trajectory construction (Ji, Z. & Ji, H., 2016), and (iii) its overall competitive performance in multiple previous benchmark evaluations (Saelens et al., 2019;

Tian, L. et al., 2019). Lamian also allows one to use pseudotemporal trajectories generated by other methods to replace TSCAN as input for downstream analysis. However, the uncertainty quantification of trajectory topology for the other methods are currently not supported in Lamian due to technical complications in implementation.”

Regarding the second question, we used the same input pseudotime for all existing DE methods involved in the comparisons. Therefore, the comparison is fair and not influenced by the quality of pseudotime inference.

2. A drawback of Lamian is that it does not account for the uncertainty in pseudotime inference because it only applies the bootstrap to the pseudotime values of all cells. In other words, the bootstrap only captures the randomness in sampling cells, not the randomness in the inference pseudotime values of all cells.

[Editor: This and other limitations of the method should be clearly discussed in the revision.]

We thank the reviewer for raising this point. Uncertainties in the pseudotime analysis include both the uncertainty of the inferred pseudotime and the uncertainty of gene expression conditional on pseudotime. In order to account for the uncertainty in constructing the pseudotime, our module 1 bootstraps cells to reconstruct the minimum spanning tree, repeatedly, and hence incorporating the uncertainty from the inferred pseudotime itself. Conceptually, one could account for the pseudotime reconstruction uncertainty in the downstream differential gene expression and cell abundance analysis by fitting the temporal gene expression and cell abundance curves for each bootstrapped tree. However, practically, it will make the differential analysis difficult to implement and make the results difficult to summarize and report. This is because trees reconstructed from different bootstrap samples can have different topologies due to the randomness. A tree branch that appears in one tree may not exist in another tree, and often it is unclear how one should align branches of different trees. Consequently, it is not always clear what it means if one says that “a gene is differential along a trajectory lineage” since the lineage does not always exist in all trees. It is also unrealistic to enumerate and report all branches that occurred in bootstrapped trees and differential genes for each bootstrapped branch.

For this reason, we separated the evaluation of uncertainties of the inferred pseudotime (i.e. the construction of minimum spanning tree) and the evaluation of uncertainties of gene expression using a sequential “conditional” procedure. In other words, our module 1 evaluates the uncertainty of pseudotime (MST) construction. Next, **conditional** on a tree lineage and conditional on the corresponding inferred pseudotime, modules 3 and 4 perform differential analyses using bootstrap sampling to account for the cell-level uncertainty, followed by modelling sample- and cell-level variability to account for gene expression variability and uncertainty. This logic is analogous to the use of conditional probabilities in statistics, that is, in order to study the probability that a gene is differential along a cell lineage, one can first study the probability that the cell lineage exists and then study the conditional probability that a gene is differential given the lineage. We use this sequential procedure as it avoids the complication in the tree alignment and it is also easier to summarize and report its results to the end users. We acknowledge that this approach may not be perfect, but it is practical to incorporate both types of uncertainty.

We also note that a recent method pseudotimeDE also tries to account for pseudotime reconstruction uncertainties in pseudotime analysis via subsampling cells and permuting pseudotime which aim to capture randomness both in sampling cells and inferred pseudotime values. However, when we tested pseudotimeDE on our benchmark data, we found that its subsampling and permutation procedure was computationally slow and does not scale well to relatively large datasets. For the TDE analysis of our HCA bone marrow data, for instance, pseudotimeDE failed to generate results after running one week on a computing cluster with 20 CPU cores and 200GB. Furthermore, pseudotimeDE was developed for single sample analysis and does not consider multiple samples and therefore does not characterize variability and differences across samples. Unlike pseudotimeDE, Lamian considers multiple samples, is capable of handling large datasets, and performs reasonably well in our benchmark datasets. For these reasons, we adopted the current procedure in Lamian which, although imperfect, was able to balance our needs for computational feasibility while accounting for pseudotime inference uncertainty.

In the Discussion, we have now added the following:

“Uncertainties in the pseudotime analysis include both the uncertainty of the inferred pseudotemporal trajectory and the uncertainty of gene expression or cell abundance conditional on pseudotime. In Lamian, the trajectory inference uncertainties are characterized by bootstrapping cells to compute detection rates. Conceptually, one could also account for the pseudotime reconstruction uncertainty in the downstream differential gene expression and cell abundance analysis by fitting the temporal gene expression and cell abundance curves for each bootstrapped tree. However, practically, it will make the differential analysis difficult to implement and make the results difficult to summarize and report. This is because trees reconstructed from different bootstrap samples can have different topologies due to the randomness. A tree branch that appears in one tree may not exist in another tree, and often it is unclear how one should align branches of different trees. Consequently, it is not always clear what it means if one says that “a gene is differential along a trajectory lineage” since the lineage does not always exist in all trees. It is also unrealistic to enumerate and report all branches that occurred in bootstrapped trees and differential genes for each bootstrapped branch. For this reason, Lamian separated the evaluation of uncertainties of the inferred pseudotime (i.e. the construction of minimum spanning tree) and the evaluation of uncertainties of gene expression using a sequential “conditional” procedure. In other words, our module 1 evaluates the uncertainty of pseudotime (MST) construction. Next, conditional on a tree lineage and conditional on the corresponding inferred pseudotime, modules 3 and 4 perform differential analyses using bootstrap sampling to account for the cell-level uncertainty, followed by modelling sample- and cell-level variability to account for gene expression variability and uncertainty. This sequential procedure avoids the complication in comparing different trees and makes it easier to summarize and report analysis results to the end users. Thus, while it is imperfect, it provides a practical solution to this complicated problem. Developing better methods that can simultaneously account for all sources of uncertainties including pseudotime inference uncertainty, gene expression and cell abundance inference uncertainty, and cross-sample and cross-cell variability remains a future research topic that warrants further investigation.”

3. Because of Point #2, the permutations used in the likelihood-ratio tests (LRTs) for XDE and TDE are not necessary because the null distribution only captures the randomness in sampling cells. Hence, the authors should be able to use the theoretical, asymptotic null distribution of the LRT statistic and greatly reduce the computational time required by permutations.

We thank the reviewer for this suggestion. We have now added an implementation of the XDE and TDE test based on the asymptotic null distribution, that is, chi-squared distribution, of the likelihood-ratio tests (LRTs). We have provided both permutation-based null and chi-squared null distributions as options for users. We denote Lamian based on chi-squared null distribution as “Lamian.chisq”, and the Lamian based on permutation null distribution as “Lamian.pm”. We evaluated and compared Lamian.chisq and Lamian.pm in the new **Figures 4,5,6,S13**.

In our analyses, we found Lamian.chisq to be more computationally efficient. For example, for the TB dataset, which consists of 337,191 cells from 184 samples, our model can finish the most complicated test, i.e. XDE test, in 9 hours within 5 GB memory using Lamian.chisq, and 63 hours within 250 GB memory using Lamian.pm. However, we observed that Lamian.chisq was less conservative than Lamian.pm and failed to control FDR in a number of simulations, even though Lamian.chisq’s overall performance remains reasonably good and outperforms most other methods. This is not unexpected since the chi-squared null for LRT is based on asymptotic results and there is no guarantee that the asymptotic results will hold true in finite sample scenarios. Compared to Lamian.chisq, Lamian.pm was better able to control FDR because it does not rely on asymptotic results and therefore is more robust to deviations from model assumptions, and it also accounted for the randomness of cell sampling (via bootstrapping cells) which the asymptotic chi-squared null distribution does not consider.

Therefore, there are pros and cons of using the asymptotic null distribution. The pro is computational efficiency, and the con is the possibility of underestimating FDR in some datasets. We provide both Lamian.pm and Lamian.chisq to users. We recommend that users apply Lamian.chisq when they would like to quickly get an initial result and can tolerate, to some extent, false positives, especially when dealing with a large dataset. Users can apply Lamian.pm to better control FDR when they have enough computational time or resources to implement parallelizations.

This is now discussed in the **Results section 2.1.3**:

“By default, Lamian uses a permutation approach to determine statistical significance of the DE tests (Lamian.pm). This approach is more reliable but can be computationally slow. For fast computation, Lamian also provides an option to determine significance using the chi-squared distribution as the asymptotic null for the likelihood ratio statistics (Lamian.chisq). This option is fast but less accurate. It can be used when users want to run a quick initial analysis while waiting for more rigorous results from Lamian.pm, especially when dealing with a large dataset. Below Lamian refers to Lamian.pm unless otherwise specified.”

4. A computational issue related to Point #2 is that the authors used the kernel density estimate (KDE) of the permuted LRT statistic values as the null distribution. However, KDE relies on a pre-specified kernel bandwidth, which is automatically chosen by the R function for every gene but not necessarily appropriate for some genes. This computational instability should be avoided by replacing the permutation approach by the theoretical null distribution as I suggested in Point #2.

We thank the reviewer for this suggestion. It is true that KDE relies on a pre-specified kernel bandwidth. As the reviewer correctly pointed out, we used the default in the base R function `stats::density()`, which automatically chooses bandwidth based on the data. It may have its limitations just as all methods have their own limitations. In response to this suggestion and to the previous comment, we have implemented the theoretical asymptotic null distribution (Lamian.chisq). We have included Lamian.chisq in all benchmarks. As discussed above, our comparison between the asymptotic null (Lamian.chisq) and the permutation null (Lamian.pm) shows that the permutation null performs slightly better than the asymptotic null with respect to controlling false discovery rates. In the revised manuscript and corresponding software implementation, both Lamian.pm and Lamian.chisq are provided to users. While Lamian.pm was set as the default method in Lamian, users can choose which method to use based on the needs of their own application.

5. Module 2 uses the two-sample t test to compare branch proportions between two conditions. The t test is inappropriate for two reasons: (1) branch proportions are not normally distributed; (2) proportions of different branches are not i.i.d.

We thank the reviewer for raising this point. We have now implemented multinomial logistic regression as an alternative option to the two-sample *t*-test. Multinomial logistic regression is a standard approach for modelling proportions for multi-category data considering the constraint that all proportions in a sample sum up to 1. This approach models the log ratio of cell proportions between each branch and a baseline branch as a function of sample covariates. By default, we choose the branch with the largest number of cells as the baseline branch. The advantage of multinomial logistic regression approach is that it appropriately models the proportions. The disadvantage is that interpretation of its results is less straightforward for a user not familiar with this statistical model. Since the ratio involves two branches, one always needs to involve two branches when reporting the results. For example, a typical report of the result would look like “changing sample covariate X1 by one unit will result in an increase of log ratio of cell proportions between the myeloid lineage (branch 1) and lymphoid lineage (branch 2) by β_1 ”. It is not straightforward to know whether the cell proportion of the myeloid lineage (branch 1) is increased or decreased, since the answer will also depend on

how branch 3 or other branches' cell proportion change relative to branch 1 and branch 2 (i.e. the ratios of branch3/branch2, branch4/branch2, etc.).

Although the *t*-test has the limitation that it ignores some properties of proportion, it has the advantage of an easy interpretation. In the *t*-test approach, each branch is analysed separately from other branches. The results are typically reported as, for example, "changing sample covariate X1 by one unit will result in an increase of cell proportion by 30% in the myeloid lineage (branch 1)". This is easier to communicate and also easier to understand for users not familiar with statistical models. Based on our experience with our biologist collaborators, the *t*-test approach is preferred by most of our collaborators as it better facilitates scientific communication.

Nonetheless, we now provide both options in Lamian. The default output uses *t*-test, but users can choose to report the multinomial logistic regression results as well. The multinomial logistic regression option is now mentioned in the Methods section 4.4:

"Statistical significance of the association between a sample covariate and the branch cell proportion is determined by testing whether the corresponding regression coefficient is zero using either two-sided t-test (default) or multinomial logistic regression (optional)."

6. A limitation of Module 2 is: what if two conditions have different topologies, e.g., the two conditions have difference single branches. In this case, how can Lamian tell that the two single branches are not the same?

If two conditions have different topologies (e.g. each has a condition-specific branch), then after data integration (e.g. Seurat or Harmony) and trajectory construction, one will have a branch (branch A) that only contains cells from condition 1, and another branch (branch B) that only contains cells from condition 2. The differential topology test in our module 2 will test the cell proportion differences between the two conditions for each branch. For branch A, it will report that there is a significant difference in cell abundance between condition 1 and condition 2, and it will also report the mean cell proportion in that branch for each condition. Users will be able to see that the proportion of cells in branch A in each sample from condition 2 is almost zero, but the cell proportion in branch A for condition 1 is above zero. Therefore, based on this information one will know that branch A is likely condition 1 specific. Similarly, one can tell that branch B is condition 2 specific since the cell proportion on that branch is almost zero for condition 1 and is positive for condition 2, and the difference between the two conditions is significant.

We have now added this explanation in **Methods section 4.4**.

7. Figure 4h suggests that the simulation for power comparison and FDR control was not realistic. The ROC curve is too perfect. I suggest that the authors add a more realistic simulation study for benchmarking Lamian against limma and tradeSeq. [Editor: This point would be necessary for further consideration at Communications Biology.]

We apologize for the confusion. Our simulation indeed covered a wide range of signal-to-noise levels. The ROC curve presented in the initial submission appeared to be "too perfect" simply because we showed the ROC analysis for the strongest signal-to-noise ratio scenario ("signal strength = 4"). In this strongest signal case, the AUC of Lamian is over 0.95. However, that figure did not capture the full spectrum of our simulations. Our simulations also covered weaker signal strengths settings including 0.5, 1, 2, and 3. Similar to the sensitivity-realFDR curves of signal strength 4 in the original Figure 4h, in the new Figure 4h (also shown below) we show the curves of "signal strength 1". As one can see, in this setting, the AUC of Lamian is far from perfect (~0.85), and many competing methods have very low AUC (e.g. below 0.25 by

tradeSeq_diffEndTest). To avoid this confusion, now we have replaced the original Figure 4h (signal strength = 4) with this figure (signal strength = 1).

— condiments
 — Lamian.chisq
 — Lamian.pm
 — limma
 — monocle2TrajTestCorr
— phenopath
 — tradeSeq_diffEndTest
 — tradeSeq_earlyDETest
 — tradeSeq_patternTest

8. In Methods, why do equations (2) and (4) share the same variance σ_s^2 ? Equation (4) does not seem to need σ_s^2 .

In principle, assuming $u_s \sim N(0, \sigma_s^2 * \Omega)$ or $u_s \sim N(0, \Omega)$ will both work. We chose to use the form $u_s \sim N(0, \sigma_s^2 * \Omega)$ coupled with the assumption $\sigma_s^2 \sim \text{InverseGamma}(\alpha, \eta)$ simply because it provides a conjugate prior for u_s and σ_s^2 . This conjugate prior leads to a cleaner form of the posterior (the posterior is given in Supplementary Notes 1) which can be more easily and efficiently implemented and computed.

9. Table S1 listed several existing methods, but these methods were not cited in the main text. The authors should mention these methods in the main text and add their citations. **[Editor: For the sake of reproducibility, please cite these Methods and list any relevant version information.]**

We thank the reviewer for this excellent point. We have now cited all methods in Table S1 in the manuscript. We also included two new methods in Table S1 (condiments and PhenoPath) and cited them. Limma was cited in Section 2.4.1 and all other methods were cited in the Introduction.

10. A right parenthesis in line 27 on page 7 should be removed.

Thank you for pointing this out. We have removed the right parenthesis in line 27 on page 7.

11. Regarding impact: Nature Communications.

12. Regarding strength of claims: Please see my comments under the overall significance.

13. Regarding reproducibility: I didn't run the code, but the writing was overall clear.

We thank the reviewer for the above comments.

Reviewer #2:

1. Regarding impact: Scientific Reports; I don't think this will influence thinking in the field. [Editor: As before, all decisions regarding publication are made by editors.]

We sincerely thank the reviewer's time and constructive feedback. We have carefully addressed the concrete questions raised by the reviewer. For the evaluation of impact, it represents an opinion. While we respect the reviewer's personal opinion, we do feel strongly that the evidence we saw from the data does not support this reviewer's judgement and therefore have a different opinion. We would like to point out that a crucial limitation of the existing pseudotime analysis methods is that most methods do not consider sample-to-sample variability. Without accounting for such variability, statistical inference will be invalid in multi-sample studies, which can lead to hundreds to thousands of false discoveries. For example, In the HCA bone marrow null simulation where there is no differential genes for the XDE analysis, Lamian.pm (with permutation test) reported zero false positives at 5% FDR cutoff, whereas the other methods reported 7846 (monocle2TrajTestCorr), 8783 (tradeSeqPatternTest), 7259 (tradeSeqEarlyDETest), 5822 (tradeSeqDiffEndTest), 7400 (PhenoPath), and 8753 (condiments) differential genes which are all false positives. Similar null simulation analysis results for TDE analysis is shown in **Figure S13a-b**. Given that pseudotime analysis is now used widely and almost all such analyses rely on these existing tools, such false discoveries are common and will have a substantial negative impact on scientific research.

The importance of accounting for sample level variability has been appreciated in many other areas of genomics (e.g. limma, DESeq, edgeR etc for conventional differential gene expression analysis), but so far it has not been widely recognized and systematically addressed in the field of pseudotime analysis. While **PhenoPath** (Campbell and Yau, 2018) is a method that can compare samples from different conditions, it has restrictive assumptions that limits its generality, and it lacks scalability to large datasets. Moreover, it performs substantially worse than Lamian (see details in the responses to question #2 below). This is why we believe that it is so important to make people aware of this issue and to provide a practically viable solution to address it in order to avoid false discoveries and better detect true signals. Indeed, both Reviewers #1 and #3 agree with our overall assessment of the importance of Lamian. Reviewer 3 recommended Nature Methods and Reviewer 1 recommended Nature Communications. We note that their assessments of impact are very different from Reviewer #2.

2. The authors argue that comparing pseudotemporal dynamics across multiple samples is lacking and claim lamina is one of the first model to do so. This claim, however, is rather weak and should be tuned down or backed up much better. The author should do a more inclusive review or summary of existing tools for multi-sample pseudotime analysis to support this claim (for example, providing a supplementary table to summarize existing pseudotime analyses tools and how lamina fills this gap).

For example, Kieran R Campbell (<https://www.nature.com/articles/s41467-018-04696-6>) has published papers on differential pseudotime analyses with covariates. In addition, the trajectory-conditioned test from Monocle 2 (Fig 2 of <https://www.ncbi.nlm.nih.gov/pmc/articles/PMC5764547/>) performs differential analyses to identify genes that change under different genetic backgrounds (samples) along pseudotime trajectory. Fundamentally, inclusion of the sample covariate when performing a generalized linear regression is a rather conventional practice in biostatistics and thus the novelty of this approach is limited. [Editor: Further benchmarking and reporting of Lamian's advantages over existing methods would be necessary for consideration at Communications Biology.]

We thank the reviewer for raising this point. We have now compared PhenoPath (Campbell and Yau, 2018) and Monocle 2 in this revised manuscript. Below we would like to first explain the differences between these methods and Lamian and then use data to show why Lamian represents a substantial improvement compared to these methods.

PhenoPath (Campbell and Yau, 2018)

PhenoPath (Campbell and Yau, 2018) is **significantly different from Lamian**. Their main differences can be summarised below.

- (1) Phenopath assumes gene expression changes linearly along pseudotime and cannot deal with arbitrary differences between conditions which may be non-linear functions of pseudotime. By contrast, our Lamian model uses a B-Spline regression for the temporal patterns and hence can deal with non-linear pseudotemporal gene expression patterns.
- (2) PhenoPath is not scalable. For example, it failed to generate results for the COVID dataset with ~56,000 cells and the TB dataset with ~337,000 cells. By contrast, Lamian is scalable and can be successfully run on both datasets.
- (3) Phenopath does not consider complex pseudotemporal trajectories. It considers a very specific scenario where data can be represented as a one-dimensional linear embedding which represents a biological progression or pseudotime (see Campbell and Yau, 2018 Methods section). By contrast, Lamian and many other competing methods such as condiments, tradeSeq, etc., considers more general scenarios and can infer more complex pseudotemporal trajectories including those with branches.
- (4) PhenoPath considers comparison between two groups of cells. Unlike Lamian, it does not consider the variation of multiple samples within each group. It does not estimate sample-level variance separately from cell-level variance. Thus, similar to condiments and tradeSeq, PhenoPath cannot assess whether the observed difference between conditions is real or expected by chance based on the random sample-level variability within each condition. By contrast, Lamian can do this.

As suggested by the reviewer, we have now benchmarked Phenopath in our revised manuscript. Lamian outperformed Phenopath substantially both in terms of accuracy and scalability (HCA bone marrow: **Figure 4e-i**; COVID, TB: phenoPath failed to run on data at this scale; computational efficiency and scalability: **Figure S19**). We have also added a brief review of Phenopath in the **Introduction section**:

“Phenopath assumes gene expression changes linearly along pseudotime and cannot deal with arbitrary differences between conditions which may be non-linear functions of pseudotime. Moreover, it does not estimate sample-level variance separately from cell-level variance. Thus, similar to condiments, one cannot assess whether the observed difference between conditions is real or expected by chance based on the random sample-level variability within each condition. Although properly accounting for the variation across samples is important in multi-sample single-cell data, neither PhenoPath nor condiments can meet this need.”

Monocle 2

We have carefully read the Monocle 2 paper and its software website. We found that the trajectory-conditioned test mentioned by the reviewer is not a test for comparing genes' pseudotemporal trajectories between two groups of samples. Instead, it is a test that evaluates whether a gene changes its expression along pseudotime, after accounting for potential covariates. In other words, it is essentially a TDE test rather than a XDE test. Thus, it does not provide the key function Lamian provides for XDE detection. Because of this, Lamian is fundamentally different from Monocle 2. Below we elaborate on this.

First of all, we did not find the trajectory-conditioned test in Fig 2 of the Monocle 2 paper which was suggested by the reviewer (<https://www.ncbi.nlm.nih.gov/pmc/articles/PMC5764547>) or the software website (<http://cole-trapnell-lab.github.io/monocle-release/docs/#differential-expression-analysis>). However, we were

able to find the trajectory-condition test on page 23 of the supplemental materials. The screenshot below shows the original description of this test in the Monocle 2 paper's supplemental material:

In principle, trajectory-conditioned test can run by considering both of the pseudotime and the grouping, similarly to the previously defined BEAM test, where the full model is

$$g(E(Y)) = \beta_0 + f_1(G) + f_2(\varphi)$$

The alternative model is

$$g(E(Y)) = \beta_0 + f_1(G)$$

In each of the model, $E(Y)$ represents the expected value for the transcript counts data Y where Y is negative binomial distributed or $Y \sim NB(r, p)$. g is a link function; for negative binomial distribution, it is \log . $f_1(G)$ represents the indicator function for the genotype G while $f_2(\varphi)$ represents the non-parameteric function, such as the natural spline implemented in VGAM [17] (*sm.ns* function), of pseudotime φ .

Based on this description, φ is pseudotime. A gene's pseudotemporal expression pattern is modelled using a temporal function $f_2(\varphi)$ which can be shifted up or down by an intercept term $\beta_0 + f_1(G)$. The intercept depends on sample phenotype G (in Monocle 2, G represents sample's genetic background). The trajectory-conditioned test compares two models to determine whether the $f_2(\varphi)$ term is zero or not. A significant p -value will suggest that $f_2(\varphi)$ is non-zero, and therefore gene expression changes along pseudotime φ . Note that under the model assumption, all samples have the same $f_2(\varphi)$. This means their pseudotemporal patterns are assumed to be the same, and the only difference between samples with different genetic background G is their different intercept (or offset) $f_1(G)$. An insignificant p -value will suggest that $f_2(\varphi)$ is zero, and therefore gene expression does not change along pseudotime φ . Then for each sample, the gene expression is a constant $\beta_0 + f_1(G)$ which depends on sample's genetic background G . Clearly, the trajectory-conditioned test does not test whether the pseudotemporal pattern $f_2(\varphi)$ is different between different sample groups. Thus, it is a TDE test rather than a XDE test.

Based on the original description, the trajectory-conditioned test also does not test whether $f_1(G)$ is zero or not. In other words, it does not test whether the gene expression is associated with sample's genetic background G . The term $f_1(G)$ is only included in this test as a covariate adjustment to account for potential differences in genetic background. Even though in principle one can modify the code to test $f_1(G)$, this requires users to study and modify the code of Monocle 2 and is not generally accessible to an average Monocle 2 user.

With this said, we gave Monocle 2 the benefit by modifying its model and code by ourselves to allow both TDE and XDE test.

For TDE test, the code below implements the model presented above. This is referred to as "monocle2_trajTest":

```
full_model_fit <- VGAM::vglm(s~x+sm.ns(pt, df=3), epsilon=1e-1,
family='uninormal')
reduced_model_fit <- VGAM::vglm(s~x, epsilon=1e-1, family='uninormal')
lrt <- VGAM::lrtest(full_model_fit, reduced_model_fit)
lrt@Body["Pr(>Chisq)"][2,]
```

For XDE test, we first modified the model and test. The revised null model is $\beta_0 + f_2(\varphi)$, and the revised full model is $\beta_0 + f_1(G) + f_2(\varphi) + f_2(\varphi) f_1(G)$. Comparing these two models will test whether $f_1(G) + f_2(\varphi) f_1(G)$ is zero. This is equivalent to testing both mean shift ($f_1(G)$) and trend difference ($f_2(\varphi) f_1(G)$). We refer to this test as "monocle2TrajTestCorr" (Corr = 'corrected'). Below is the code:

```
full_model_fit <- VGAM::vglm(s~x+sm.ns(pt, df=3)*x, epsilon=1e-1,
family='uninormal')
```

```

reduced_model_fit <- VGAM::vglm(s~sm.ns(pt, df=3), epsilon=1e-1,
family='uninormal')
lrt <- VGAM::lrtest(full_model_fit,reduced_model_fit)
lrt@Body["Pr(>Chisq)"][2,]

```

We have now benchmarked this modified Monocle2 in our revised manuscript, and our results show that it also performs worse than Lamian (**Figure 4, 5, 6, S11, S13**).

Below are some major results.

In the HCA bone marrow null simulation for XDE analysis: Lamian.pm (with permutation test) reported zero false positives at 5% FDR cutoff. Using the same cutoff, Lamian.chisq reported 62 XDE genes. By contrast, other methods reported 7846 (monocle2TrajTestCorr), 8783 (tradeSeqPatternTest), 7259 (tradeSeqEarlyDETest), 5822 (tradeSeqDiffEndTest), 7400 (PhenoPath500, max.iter= 500 , without setting max.iter the program cannot provide results), and 8753 (condiments) differential genes, which are false positives. Similar to Lamian, limma reported no XDE genes. However, as will be shown in later Results sections, limma can only detect differences in mean expression and cannot detect trend differences in pseudotemporal patterns.

In the HCA bone marrow spike-in simulation (new Figure 4e,f): Lamian.pm properly controlled the FDR (left plot) and provided the best sensitivity vs. FDR curve (right plot). The other methods either failed to control FDR or performed much worse in sensitivity vs. FDR curve or both.

In HCA real data example (new Figure 4i, S11): For detecting sex-associated genes, XDE genes reported by Lamian and limma were significantly enriched in sex chromosome genes in all three major hematopoietic differentiation lineages, whereas the other methods did not.

- Myeloid lineage

• **Erythroid lineage:**

• **Lymphocyte lineage:**

In COVID-19 real data example (new Figure 5g): XDE genes reported by Lamian showed the best consistency between two independent partitions of data. The other methods had worse performance. PhenoPath failed to run with one week of time and 400G memory.

In TB real data example (new Figure 6e): XDE genes reported by Lamian again showed the best consistency between independent partitions of data. PhenoPath, tradeSeq, and condiments failed to run with one week of time and max 400G memory.

As shown in the above figures, Lamian has an overall superior performance compared to existing methods. This is because the methodological advances, which we summarize below:

- In terms of single-cell pseudotime analysis, we are the first to systematically illustrate, benchmark and report the serious impact of ignoring sample-level variability on downstream differential expression (including it may create substantial number of false discoveries) that so far received little attention in the community and published literature.
- We are also the first to develop a statistically-rigorous framework that systematically addresses different sources of variability, particularly sample-level variability, in the multi-sample pseudotime analysis. It offers a comprehensive package of solutions to analyze trajectory topology uncertainty, differential topology, differential gene expression and differential cell abundance. Its model offers generality to handle non-linear pseudotemporal patterns, and our software showed good scalability to large datasets that many other methods do not have.
- Besides the major methodological advances above, we also designed and implemented a set of visualization functions in Lamian such as the heatmap that systematically organizes and presents the results to biologists. These visualisation functionalities in the software provide useful and important tools to facilitate data exploration, communication and interpretation by the broad scientific community.

3. Furthermore, different samples may lead to strong batch effects that are not biologically relevant. Thus, it is important to discern the real biological difference and the technical differences between different samples. The author relies on existing approaches to remove batches before downstream differential analyses but it is worth discussing this issue. Another possibility is to avoid the batch effect issue by focusing on analyzing datasets that are generated from the same batch but still involve different samples (for example, including drug treatment, genetic perturbation, etc.). [Editor: Discussion of this point would be necessary for further consideration at Communications Biology.]

The reviewer raised an excellent point on the importance of batch effects. We are sorry that we may not have made it clear, but Lamian is able to correct batch effects. Also, the term “batch effect correction” is used in the field in different ways which can cause confusion. Below we will clarify what we mean by batch correction and how we did it. To make it clear, first we know that the difference between any two samples can be either due to true biological difference or due to unwanted technical differences such as systematic batch effects or random measurement noises. The goal of differential expression analysis is to identify true biological differences. In order to achieve this, there are two main steps in the multiple-sample scRNA-seq differential gene expression analysis.

STEP 1 (sample integration):

In this step, one integrates cells from different samples together so that cells of the same type are aligned across samples. This is to make sure that when we compare different samples, we are comparing oranges with oranges (e.g. T cells from patient 1 vs. T cells from patient 2) and apples with apples (e.g. B cells vs. B cells), rather than comparing oranges with apples (e.g. T cells from patient 1 vs. B cells from patient 2). This step is accomplished using existing approaches such as Seurat(CCA) and Harmony. These approaches aim to retain biological differences across cell types, but they remove both biological differences across samples

and systematic technical differences such as batch effects. Even though some people call these methods “batch correction” methods, they are actually methods that remove both batch effects and true biological differences across samples to facilitate sample integration. For this reason, we prefer to call them “sample integration” methods rather than “batch correction” methods. These methods may report corrected gene expression values after sample integration (i.e. after removing sample-level differences), but these corrected expression values cannot be directly used to study differential expression across samples (e.g. healthy vs. COVID difference) because all true biological differences between samples are removed together with batch effects.

STEP 2 (differential expression analysis with batch effect correction):

After matching cells of the same type across samples, STEP 2 then analyses differential expression using the matched cells. Since one cannot use the corrected expression values from STEP 1 for the reasons explained above, one has to use the original uncorrected gene expression values to perform this analysis. This is also what the authors of Seurat recommended (see the “Identify conserved cell type markers” section: https://satijalab.org/seurat/articles/integration_introduction.html, first code chunk: “*For performing differential expression after integration, we switch back to the original data*”). This is not saying that STEP 1 is not useful. That step is indeed crucial since after STEP 1 we now know which cells are of the same type that we can meaningfully compare across samples. Now for a given type of cell (e.g. T cell), differences in the original gene expression values between different samples contain both true biological difference and technical difference, and one has to separate the true biological difference from the technical difference such as batch effect. For that, one needs to do batch correction. **Note that here “batch correction” refers to a procedure that removes batch effects while retaining the true biological differences across samples, which is different from the “sample integration” step in single-cell analysis where both differences are removed.** When the reviewer says that we used the existing batch correction method, the reviewer seems to be confused between sample integration and batch correction. The methods referred by the reviewer are sample integration methods rather than batch correction methods.

Lamian indeed supports batch correction (STEP 2) in the differential expression analysis via its regression framework. In its main function `lamian.test()`, users can specify multiple covariates for the samples in the design matrix. This allows one to remove and account for confounding effects across samples such as batch effect, sample’s baseline characteristics (e.g. age), genetic background, etc. In the design matrix, each row represents a sample, each column represents a sample covariate, and the matrix entries are values of the sample covariates (except for the first column where all values are equal to 1 to represent the intercept). Users can pass the matrix to the argument *design*, and specify the covariate they are interested in running differential tests in the argument *test.for* as well as the covariate(s) they would like to adjust for as confounding factors in the argument *adjust.for*. For example, if the user wants to test sex differences while adjusting for the samples’ batch effect, he/she can run the model as

```
Res <- lamian.test(expr = expdata$expr,
                  cellanno = expdata$cellanno,
                  pseudotime = expdata$pseudotime,
                  design = expdata$design,
                  test.type = 'variable',
                  test.for = 'sex',
                  adjust.for = 'batch',
                  permuiter = 5)
```

where the design has multiple columns for indicating intercept, sex, and batches (as dummy variables).

In the revised version, we **added a new Methods Section 4.7 “Adjusting for confounding variables such as batch effects” and a discussion in Results Section 2.1** to explain this. Furthermore, we added a **new large TB dataset** with 339,191 cells from 184 donors in 38 batches to illustrate the ability of Lamian to

correct for batch effects. We adjust for batch effects in this TB dataset by including the samples' batch information in the design matrix (**Figure 6, particularly Fig 6b,h**).

4. Page 1, Line 20: the author mentioned spatial dynamics, however the entire paper doesn't really touch analyses of this type of data and I would suggest removing this to avoid spatial transcriptomics bandwagon.

Thank you for this suggestion. In response, we have removed the "spatial" from the manuscript.

5. Page 2: The first step of Lamian uses TSCAN and bootstrapping for pseudotime and batch certainty detection. There are obviously many superb alternatives, including principal graph, diffusion pseudotime, etc., than TSCAN for pseudotime analysis and it is important to explain why this choice was used (even if this is something that is related to your lab's previous research). In addition, Ruslan Soldatov, et.al previously reported using bootstrapping of the developmental tree construction with simplePPT to estimate the certainty of the tree structure. It may be interesting to compare your approach with this method (https://www.science.org/doi/10.1126/science.aas9536?u ri_ver=Z39.88-2003&rfr_id=ori:rid:crossref.org&rfr_dat=cr_pub 0pubme d)

This is a very good question. Here, we chose TSCAN to construct the trajectory for several reasons. First, it is one of the top pseudotime inference methods (Saelens et al. 2019; Tian, L. et al., 2019). Second, its cluster-based minimum spanning tree (cMST) approach has been shown to provide robust trajectory inference compared to cell-level MST. Moreover, the cluster-based MST is highly scalable and can be efficiently applied to large atlas-level datasets since the number of tree nodes is determined by the number of cell clusters instead of cell number. Third, TSCAN also provides options for users to manually adjust the tree topology using their prior expert knowledge. If the tree automatically created by the program does not fully capture the known biology, users have the option to adjust the topology and orders of the tree nodes (i.e. cell clusters). This is feasible only because the tree nodes are clusters rather than cells. Thus, users do not need to manually order thousands of cells. Instead, they only need to work with a relatively small number of tree nodes (clusters). Although the data examples in this manuscript are all based on automatically generated trees, the flexibility for users to adjust the tree structure is practically useful in many applications as well since no pseudotime methods currently can achieve 100% accuracy in all applications. Fourth, also because we operate on cell clusters, the computation for tree inference is fast, which is important if one needs to repeatedly run tree inference tens of thousands of times in the bootstrap.

We do recognize that there are many other excellent pseudotime reconstruction methods. Therefore, we designed Lamian in a modular fashion to allow users to provide their own pseudotime as input for Lamian for the downstream differential expression and cell abundance analysis. This gives users flexibility to use their preferred methods to construct pseudotime (e.g. Monocle, diffusion pseudotime, etc.) and then use Lamian to run downstream analyses. We were also trying to incorporate some of these methods into Lamian framework, but implementing the bootstrap to evaluate uncertainties of trajectory topology is technically complicated and nontrivial (e.g., one has to address the issues of aligning tree branches, which is not easy for a complex tree structure and especially if we cannot easily access the data structure and code used to store and operate the trees in each pseudotime inference algorithm). Thus, technically it is difficult to directly incorporate these other methods into our module 1 and 2. We will continue to try these extensions in the future, but it will be beyond the scope of the current paper.

The following has been added to the manuscript Section 2.1.1.

"The advantages of using TSCAN to construct pseudotime include (i) the scalability of its cMST approach to large number of cells (since the number of tree nodes in the spanning tree is determined by cell cluster number instead of cell number) and repeated bootstrap resamplings, (ii) the flexibility it provides to support both automatic and manual trajectory construction (Ji, Z. & Ji, H., 2016), and (iii)

its overall competitive performance in multiple previous benchmark evaluations (Saelens et al., 2019; Tian, L. et al., 2019). Lamian also allows one to use pseudotemporal trajectories generated by other methods to replace TSCAN as input for downstream analysis. However, the uncertainty quantification of trajectory topology for the other methods is currently not supported in Lamian due to technical complications in implementation.”

Regarding simplePPT, we appreciate the reviewer for pointing us towards it. We read the paper noted by the reviewer (Soldatov et al. *Science* 364, eaas9536 (2019)), and found that on page 12 of the Supplementary Methods section, the authors wrote “We fit a principal tree to the observed data in high-dimensional space based on the SimplePPT approach (Qiu et al. *Nat Methods* 14, 979–982 (2017))”. We would like to clarify some misunderstandings about (1) what is simplePPT, (2) how simplePPT is different from the bootstrap framework presented in the *Science* paper, and (3) the availability of code/software for these functions.

First, simplePPT is a method for dimensionality reduction, which is implemented in the Monocle2 software package (Qiu et al. *Nat Methods* 14, 979–982 (2017)). Specifically, the original method called simplePPT (published in Mao et al. *SIAM International Conference on Data Mining* (2015)) is a principal tree algorithm that learns principal points and tree structure. This method was then implemented in the `reduceDimension()` function in Monocle2 (Qiu et al. *Nat Methods* 14, 979–982 (2017), function manual: <https://rdrr.io/bioc/monocle/man/reduceDimension.html>). To be clear, simplePPT does not use bootstrap sampling to estimate the uncertainty of branches of trees.

Second, the authors of the *Science* paper wrote on page 12 of their Supplemental Methods that “branches of trees produced under such parameters are not driven by a small number of outlier cells as evident from reconstruction of the ensemble of trees using subsampling of 90% of cells”. The authors of the *Science* paper used subsampling to estimate the uncertainty of the branches of the trees learned from simplePPT. We agree with the reviewer that this subsampling idea is similar to our Module 1 where we also bootstrap the cells to estimate the uncertainty of the tree structure. Upon understanding the similarity, we set out to compare our module 1 to the subsampling framework by the authors in the *Science* paper. This brings us to the third point.

Third, after some explorations, we realize that we were not able to compare our Module 1 to their subsampling framework as the authors of the *Science* paper did not provide code to implement their subsampling framework. In this sense, the *Science* paper does not provide a general uncertainty evaluation tool that a third-party user can easily use. Since simplePPT was included in the Monocle2 package as a dimension reduction methods’ option, and this *Science* paper did not provide any publicly-available codes or detailed description about how they implemented the bootstrapping with monocle2’s simplePPT option, we think it is beyond a regular user’s ability to reproduce or apply their algorithm in practice. Therefore, we think it is beyond the scope of our current paper to try to guess and reimplement their method as a comparison to our method.

6. Page 3: Some information about the schematic is not very clear. For example, what is the meaning of 0.3 (0.2) for example, in the section related to cell proportion in a branch of the panel b (mean and s.d.?) Similarly, simple explanations of the meaning of the equations in panel C needs to be spelled out in the schematic.

Thank you for this great suggestion and we are sorry for not making it clear. We have made the following modifications.

- (1) In figure 1 panel b, we have removed the numbers in the parentheses (sd) to avoid confusions. The updated table is shown below. For each branch, it shows the branch detection rate based on bootstrapping cells, and the proportion of cells in that branch in each sample.

Cell proportion in branches

branch	sample 1	sample 2	...	sample N
1	0.3	0.4		0.2
2	0.4	0.4		0.1
3	0.1	0.1		0.3
4	0.2	0.1		0.4

- (2) We have modified Figure 1 schematic panel c to add explanations of the equation, as follows.

$$\begin{array}{ccccccc}
 \text{observed data} & & \text{group-level effect} & & \text{sample-level effect} & & \text{cell-level effect} \\
 \mathbf{y}_s = & & \mathbf{\Phi}_s \mathbf{X}_s \boldsymbol{\beta} & + & \mathbf{\Phi}_s \mathbf{u}_s & + & \epsilon_s \\
 \text{B-spline basis} & \text{sample design} & \text{coefficient of} & & & & \\
 \text{of pseudotime} & \text{(covariates)} & \text{interest} & & & &
 \end{array}$$

We've also added additional information in the legend providing the meaning of the equations. Specifically, In figure 1's legend, we added two sentences to explain the equations in panel C, as follows.

“Gene’s or cell abundance’s pseudotemporal patterns are modelled using combinations of B-spline bases (Φ_s) to allow non-linear patterns. The combination coefficients are decomposed into effects due to sample covariates ($X_s\beta$, where X_s is the design matrix) and variation among samples with common covariate values (u_s). Cell-level data y_s in sample s are generated from the sample-level curve by adding cell-level random noise ϵ_s . See Section 4.5 and Supplementary Notes for details.”

7. Page 4: Overall, the author needs to justify why the HCA- BM dataset was used for the analyses and why comparing the male and female is meaningful and interesting at all. For example, the analysis in 2.2.2 seems quite strange because you won't ever expect the hematopoiesis to differ across men and women.

Thank you for this question. Below, we want to explain the rationale and clarify what this paper is and is not. The main goal of this paper is to present a new computational method for multi-sample pseudotime analysis and benchmark the method by comparing with other existing solutions. Making new biological discoveries, while it could be interesting and may add values, is not the primary goal and focus of our current paper. Instead, benchmark is a crucial component for establishing a new method. For a benchmark, one will need datasets for which ground truth is known or partially known to the extent that different methods' performance can be robustly evaluated and compared. Unfortunately, such benchmark datasets for multiple sample differential trajectory analyses are difficult to find. The bone marrow dataset is unique in that it can largely recapture the known hematopoietic differentiation process (for demonstrating and evaluating the tree topology reconstruction), and it has samples from both males and females to allow a between-sex comparison. A priori, one would expect that differential genes associated with sex should be enriched in sex chromosomes. Since there are many sex chromosome genes, this will provide a relatively robust benchmark to compare different methods. In other words, here we mainly use the sex comparison as a way to benchmark the methods, rather than trying to make new biological discoveries. If the motivation and main novelty of a study is to claim that we have new biology about hematopoiesis, then it makes a lot of sense to investigate other aspects of hematopoiesis and delve deep into one or two findings with follow-up experiments. However, since our motivation is to benchmark a new method and compare it to other methods, it makes more sense to have a comparison where a robust and objective benchmark is feasible. That explains why we used the sex comparison here. Thus, whether our analysis looks strange or not depends on how you view the purpose and goal of this study. We hope that after we clarified the goal of the study, this example makes more sense now.

Also, we respectfully disagree with the reviewer's statement that "you won't ever expect the to differ across men and women". If the reviewer believes that this statement is true, we would be grateful if the reviewer can provide specific evidence and data to support this claim. Indeed, many groups have studied sex difference in hematopoiesis or its related process. Literature indicate that sex-associated difference exists in many hematopoiesis, bone marrow or related data, such as:

So, E.Y., Jeong, E.M., Wu, K.Q., Dubielecka, P.M., Reginato, A.M., Quesenberry, P.J. and Liang, O.D., 2020. Sexual dimorphism in aging hematopoiesis: an earlier decline of hematopoietic stem and progenitor cells in male than female mice. *Aging (Albany NY)*, 12(24), p.25939. <https://www.ncbi.nlm.nih.gov/pmc/articles/PMC7803521/>

Nakada, D., Oguro, H., Levi, B.P., Ryan, N., Kitano, A., Saitoh, Y., Takeichi, M., Wendt, G.R. and Morrison, S.J., 2014. Oestrogen increases haematopoietic stem-cell self-renewal in females and during pregnancy. *Nature*, 505(7484), pp.555-558. <https://pubmed.ncbi.nlm.nih.gov/24451543/>

Patterson, A., 2021. 3029—AGE AND SEX DIFFERENCES IN HEMATOPOIETIC STEM AND PROGENITOR CELLS CORRESPOND TO RADIORESISTANCE IN GERIATRIC MURINE MODELS OF THE HEMATOPOIETIC ACUTE RADIATION SYNDROME. *Experimental Hematology*, 100, p.S57. <https://www.sciencedirect.com/science/article/pii/S0301472X21007098>

Singer, K., Maley, N., Mergian, T., DelProposto, J., Cho, K.W., Zamarron, B.F., Martinez-Santibanez, G., Geletka, L., Muir, L., Wachowiak, P. and Demirjian, C., 2015. Differences in hematopoietic stem cells contribute to sexually dimorphic inflammatory responses to high fat diet-induced obesity. *Journal of Biological Chemistry*, 290(21), pp.13250-13262. [https://www.jbc.org/article/S0021-9258\(20\)33688-7/fulltext](https://www.jbc.org/article/S0021-9258(20)33688-7/fulltext)

Ray, R., Novotny, N.M., Crisostomo, P.R., Lahm, T., Abarbanell, A. and Meldrum, D.R., 2008. Sex steroids and stem cell function. *Molecular medicine*, 14(7), pp.493-501. <https://molmed.biomedcentral.com/articles/10.2119/2008-00004.Ray>

Faiola, B., Fuller, E.S., Wong, V.A., Pluta, L., Abernethy, D.J., Rose, J. and Recio, L., 2004. Exposure of hematopoietic stem cells to benzene or 1, 4-Benzoquinone induces Gender-Specific gene expression. *Stem Cells*, 22(5), pp.750-758. <https://academic.oup.com/stmcls/article/22/5/750/6395715>

Hörner, S., Pasternak, G. and Hehlmann, R., 1997. A statistically significant sex difference in the number of colony-forming cells from human peripheral blood. *Annals of hematology*, 74(6), pp.259-263. <https://link.springer.com/article/10.1007/s002770050296>

Topel, M.L., Hayek, S.S., Ko, Y.A., Sandesara, P.B., Samman Tahhan, A., Hesaroieh, I., Mahar, E., Martin, G.S., Waller, E.K. and Quyyumi, A.A., 2017. Sex differences in circulating progenitor cells. *Journal of the American Heart Association*, 6(10), p.e006245. <https://www.ahajournals.org/doi/10.1161/JAHA.117.006245>

According to our analysis, we also found genes that show differential temporal gene expression related to sex difference. Because sex difference was understudied in the past in many scientific investigations, a lot remains unknown about such differences, and this lack of knowledge should not be used to claim that there is no sex differences ever. Indeed, this situation is recognized by the US National Institutes of Health, and it motivates NIH to encourage investigations of sex as a biological variable in its funded research.

8. In 2.2.3, the author shows that decreasing the number of cells decreased the detection rate for myeloid branch. It will be interesting to check whether this low detection rate is biologically relevant or whether it is purely driven by uneven capture of cell number for the myeloid lineage.

This is a simulation analysis where we decreased the cell number in the myeloid lineage and asked how it will change detection rate of the myeloid branch. Therefore, the observed decrease in detection rate here is a result of the decreasing cell number rather than biology (the underlying true biology would remain the same since we did not change the biological system studied here).

Note that the detection rate is developed to help users judge to what extent they should trust the tree branch structure. Even if a branch does exist, if there are a limited number of cells, the data would not have enough information or provide enough evidence to support the existence of the branch, therefore the detection rate will be low. That is the purpose of the detection rate, and it is analogous to the concept of variance, which is developed to characterize the uncertainty of a parameter estimate (e.g. sample mean). If the sample size is small, one has larger variance, indicating that there is large uncertainty associated with the parameter estimate. Similarly, the fact that detection rate decreases when cell number decreases is also expected since detection rate is a measure developed to reflect the fact that one will have lower confidence about the tree branch when there are not many cells supporting the tree inference. Our observed results in the data are consistent with this expectation, supporting that the detection rate developed here is a good measure that can be used to characterize confidence or uncertainty of the inferred tree branch.

9. Page 5: Following gene short name convention, the author may need to go through the paper carefully to italicize all gene names (for example *CD14* to *CD14*, etc.). Importantly, the example genes shown here are less informative as most of them are CD markers which are reported previously based on FACS data (protein level). It will be more relevant to showcase known transcription factors or other markers which have a known high RNA expression level for each particular lineage.

Thank you for raising this point. We have italicized all gene names in the manuscript, for example,

We note that for many genes, protein level and RNA level are positively correlated (Li J, Biggin MD. *Science*. 2015 Mar 6;347(6226):1066-7. doi: 10.1126/science.aaa8332; Li J, Bickel PJ, Biggin MD. *PeerJ*. 2014 Feb 27;2:e270. doi: 10.7717/peerj.270), and therefore showing the RNA-level of cell surface marker proteins is not completely baseless. In our case, they clearly support the identities of the three lineages. In the revised manuscript, we have also added more plots of additional genes including lineage-specific transcription factors and surface receptors in **Supplementary Figure S1**, as follows.

Figure S1. Marker gene expression levels in the three main lineages of hematopoietic stem cell differentiation in HCA-BM data.

10. Lastly, some of the enrichment analysis doesn't make the perfect sense, for example, why "as HSCs differentiate to the erythroid lineage, the TDE genes with initially 4 high expression but low expression at the end are enriched in CD8-positive, alpha-beta T cell activation"? What we expect is that those genes should be enriched in HSC stemness maintenance related pathways. Similarly the "platelet degranulation" doesn't make a lot of sense.

Thank you for raising this point. Our wording "high expression" or "low expression" in the old manuscript is not accurate. We have now rephrased it to more accurately describe the results as follows:

"For example, as HSCs differentiate to the erythroid lineage, the TDE genes with increasing expression along pseudotime are enriched in red blood cell-related functions such as oxygen transport, whereas genes with functions in other lineages (e.g. CD8-positive, alpha-beta T cell activation, regulation of B cell receptor signaling pathway) show decreasing expression suggesting that they are increasingly suppressed (Fig. S3c,d). Meanwhile, for the lymphoid lineage, the TDE genes with increasing expression along pseudotime are enriched in T lineage commitment, whereas genes with decreasing expression lack enrichment of lymphocyte-specific functions (Fig. S3e,f)."

Note that up- or down-regulation of a gene along pseudotime does not necessarily require the gene to be highly expressed at one end of the pseudotime. For example, lymphoid genes could have low expression in HSCs, but they can be further down-regulated and therefore show differential expression along the erythroid lineage.

Also, enrichment analysis of gene sets and GO terms are not perfect. There is no guarantee that relevant gene sets will always show up (e.g. HSC stemness maintenance genes may not necessarily show statistically significant enrichment in the start of pseudotime) because gene set annotations may not be complete or clean, or they may not have enough number of genes to ensure enough statistical power. Here we simply try to objectively report the results obtained from the GO enrichment analysis by topGO (v.2.36.0). Even though certain GO terms such as HSC stemness was not reported by this analysis, the overall results still make sense since the analysis did identify relevant lineage-specific functions corresponding to the right lineages.

11. Page 7: The number of the XDE genes detected are very few and mostly related to the X/Y-chromosome, can you find any meaningful gene that are related to hematopoiesis while also have sex-difference. Basically I wonder whether what you identified are simply genes that differ between male and female but have nothing to do with hematopoiesis. This boils down again to whether it is meaningful to even perform the differential analyses between different sex for the hematopoiesis.

Among the XDE genes reported by Lamian, a number of them have functions related to hematopoiesis. For example:

ALS2 (lineage: myeloid)

<https://www.ncbi.nlm.nih.gov/pmc/articles/PMC1796843/>

“Here we report that ALS2 knockout (ALS2^{-/-}) mice developed peripheral lymphopenia but had higher proportions of hematopoietic stem and progenitor cells in which the stem cell factor-induced cell proliferation was up-regulated”

DDX3Y (lineage: myeloid + lymphoid)

<https://www.nature.com/articles/s41598-020-71447-3>

“In contrast, other studies have detected DDX3Y peptides in either neural progenitor or hematopoietic stem cells”

PDCD4 (lineage: erythroid)

<https://pubmed.ncbi.nlm.nih.gov/17259349/>

“Expression of PDCD4 was markedly up-regulated during all-trans retinoic acid (ATRA)-induced granulocytic differentiation in NB4 and HL60 AML cell lines and in primary human promyelocytic leukemia (AML-M3) and CD34(+) hematopoietic progenitor cells but not in differentiation-resistant NB4.R1 and HL60R cells.”

ZRSR2 (lineage: lymphoid)

<https://pubmed.ncbi.nlm.nih.gov/33691379/>

“ZRSR1 co-operates with ZRSR2 in regulating splicing of U12-type introns in murine hematopoietic cells”

XIST (lineage: myeloid, lymphoid)

<https://pubmed.ncbi.nlm.nih.gov/16980619/>

“We show that Xist has the ability to initiate silencing in immature hematopoietic precursor cells.”

ZFX (lineage: erythroid, lymphoid)

<https://www.ncbi.nlm.nih.gov/pmc/articles/PMC1899089/>

“Zfx controls the self-renewal of embryonic and hematopoietic stem cells”

Meanwhile, we would like to restate that the HCA-BM dataset is used to demonstrate the functions of our Lamian framework and benchmark its performance. We used chrX and chrY to validate the results in the XDE test since sex chromosome information is objective. Our primary goal is not to study the biology of the hematopoietic process. Also, In the other two immune-related real datasets, COVID and TB, many of the genes reported by our method Lamian are related to the T-cell activation process and have clear immune related functions. Our Lamian framework provides a useful tool for researchers to identify differential genes so that they can identify promising candidates to generate hypotheses and further study the underlying biology.

We have added the following sentence in the manuscript:

“Some of these XDE genes reported by Lamian have functions related to hematopoiesis (e.g. ALS2 (Erie et al. 2007), DDX3Y (Deschepper et al. 2020)). ”

12. The author didn't obtain any biological insights with the cell density analyses. Can the author find a dataset to demonstrate the relevance of this approach?

The cell density analysis consists of TCD analysis and XCD analysis.

For XCD analysis (differential cell density between conditions), we found significant differences between mild and moderate COVID patients in the COVID example. In the revised manuscript, we also added a new

tuberculosis (TB) dataset with 337,191 memory T cells from 184 samples. The XCD analysis also identified significant sex-differences, consistent with previous report in the literature. This new dataset is now presented and discussed in **Results Section 2.6**. Together, these results illustrate that XCD analysis can identify meaningful cell density differences in real data.

For TCD analysis (changes of cell density along pseudotime), we acknowledge that we did not find a dataset for which TCD results led to meaningful biological findings. In HCA bone marrow, COVID, and TB data, we all found significant cell density changes along the pseudotime. However, it is unclear whether the cell density change was due to technical sampling bias (e.g. certain cell types are easier to sample) or real biology, which makes the interpretation challenging. Despite this, we cannot rule out that some users will need this function in their own data analysis. Therefore, we decided to provide the TCD function in Lamian for completeness of the package so that users will have a tool to support such analysis if they need it in their own projects.

13. Page 9: The authors try to provide some biological insights in the final section on SARS-Covid2 infection but their conclusions are too tentative which provide little advance the Covid-19 biology and it is not clear whether their work on the subject is to drive the Covid bandwagon to their method.

We thank the reviewer for raising this point. We would like to clarify that the main innovation and contribution of this work are methodological instead of biological. As a result, the main goal of the COVID example is (1) to illustrate how the Lamian framework can be applied in a real dataset and (2) to benchmark Lamian with the other methods, rather than systematically studying COVID19 biology. In that example, the Lamian results are consistent with previously reported observations summarized from the literature. As we described in the manuscript,

“This is consistent with previous observation that comparing to the COVID-19-recovered donors, ongoing disease patients show a more TEMRA differentiation with less T-bet+ functional effector CD8 T cells (Mathew, D. et al., 2020).”

This provides evidence that Lamian analysis can generate meaningful biological results confirmed by other studies. Furthermore, the comparison with other methods in **Figure 5g** shows that the differential genes found by Lamian are most reproducible between two sets of samples. Since this is a paper primarily for introducing a computational method, advancing COVID19 biology is not our main goal and focus.

More generally, while we understand that generating new biological insights is one way to make impact, we would like to point out that having novel biological findings is not the only way to make impact. Compared to making new discoveries, avoiding and preventing false discoveries that cannot be replicated are equally important for scientific community. Indeed, the community including the Nature family journals is well aware of the replication crisis currently existing in the literature, that is, many published new findings cannot be replicated (e.g., Baker, M., Nature 533, 452–454 (2016)). Such irreplicable false discoveries have huge detrimental effects on scientific research as they can mislead other researchers, result in tremendous waste of resource, and significantly slow down the scientific progress.

One important origin of irreplicable discoveries is the incorrect statistical analysis. For multi-sample pseudotime analysis, this is indeed a problem – existing methods do not offer a good solution to analyze data from multiple samples and conditions. Because these methods fail to appropriately account for sample-level variation, using them can result in huge amounts of false discoveries reported incorrectly as “new biological findings” in the literature. For example, in our HCA bone marrow XDE null simulation analysis, we do not expect any true differential genes. However, existing methods reported thousands of “differential” genes (tradeSeq-PatternTest: 8783, tradeSeq-EarlyDETest: 7259, tradeSeq-DiffEndTest: 5822, PhenoPath: 7400, condiments: 8753, monocle2TrajTestCorr (a function in monocle2 modified by us to detect XDE):

7846,) and all of them are false positives. By contrast, Lamian with permutation test correctly reported zero false positives at 5% FDR cutoff. This result is also consistent with the analysis of the real bone marrow scRNA-seq data where existing methods reported thousands of XDE genes whereas Lamian only reported a handful of XDE genes (Fig. S9) that better captured the known biology (Figs. 4i-j, S11,S12). This is a serious problem because many of these existing methods are widely used in the literature, and not many people or reviewers are aware of their limitations in handling multiple samples. Even if they know, they do not have a handy tool to address it. If not fixed, this will continue to generate problematic publications using inappropriate data analysis which would amplify the replication crisis.

For this reason, we strongly believe that narrowly focusing on using “new biological insights” to evaluate impact of a statistical method paper misses the broader picture of the need for developing methods to address the replication crisis by ensuring statistical rigor. The impact of the Lamian method and hence this manuscript is not a specific biological finding but lies in its ability to influence how thousands of future biological studies would interpret their data and appropriately draw scientific conclusions.

14. Regarding reproducibility: Great, as it was provided as an associated package.

We thank the reviewers for the positive feedback on the reproducibility.

Reviewer #3:

1. *The authors described the input for Lamian includes a low- dimensional space representation of scRNA-seq data from multiple samples after batch effect correction (page 2, line 18-20), and listed candidate methods such as Seurat and Harmony. However, in the paper the authors conducted all analyses using Seurat approach (Section 4.2, page14, line 20), but didn't try to illustrate Lamian's performance with input from alternative methods, for example Harmony. It is important to provide some practical guidance to users about which batch effect methods to use under different conditions (e.g., different tissue types), and how the result differs. Another popular batch effect method that should fit Lamian is scVI (<https://www.nature.com/articles/s41592-018-0229-2>). I suggest the authors try both Harmony and scVI in addition to Seurat for this data processing step to see if the result differ much, and discuss more about it.*

We thank the reviewer for this comment. As the reviewer pointed out, there are a number of scRNA-seq sample integration methods and their performance may vary across different datasets. Lamian takes integrated scRNA-seq data as input. In other words, we assume that the integration is done by users and therefore users need to decide which integration method to use. The integration and Lamian pseudotime analysis on the integrated data are two separate analysis tasks. Our current study is focused on the latter. The quality of integration obviously will influence the downstream pseudotime analysis, as one would not expect that poorly integrated data will generate reliable pseudotime analysis results. In terms of which integration methods to use under different conditions, we refer readers to a recent benchmark study by Dr. M. Colomé-Tatché and Dr. Fabian J. Theis's groups (Benchmarking atlas-level data integration in single-cell genomics. Luecken et al. 2021. Nature Methods) which systematically compared 68 different method and preprocessing combinations and provided guidelines to choose integration methods in their Figure 5a. Based on that benchmark study, Seurat was a top method for integrating single-cell data, Harmony was also ranked among the top on small/simple tasks, and scVI was among the top on large/complex tasks.

Following the reviewer's recommendation, **we have also rerun Lamian analysis on the HCA bone marrow dataset using Harmony and scVI for data integration.** Using Seurat, Harmony and scVI all resulted in similar branching structure corresponding to the three major hematopoietic differentiation lineages (Fig. S17a,b, S18a,b, which are also pasted below) and similar branch cell proportion (Fig. S17c, S18c). The genes' pseudotemporal expression patterns fitted by Seurat+Lamian showed high Pearson correlation with the fitted expression patterns in the corresponding lineage fitted by Harmony(or scVI)+Lamian (Fig. S17d,e, S18d,e), indicating that different methods yielded similar pseudotemporal gene expression patterns. We also checked what percentage of differential genes reported by Seurat+Lamian (FDR<0.05) can be captured by the top differential genes by Harmony+Lamian (TDE: Fig. S17f; XDE: Fig S17g) or scVI+Lamian(TDE: Fig. S18f; XDE: Fig S18g). In both cases, there were substantial overlap between Seurat and Harmony or scVI, and the overlap is much higher than random expectation.

Figure S17. Lamian analysis of the HCA-BM data based on Harmony integration and comparison with Seurat integration. (a-b) Cells plotted with top two latent variables from Harmony, colored by clusters (a) or pseudotime (b). Pseudotime was inferred using top 10 latent variables output by Harmony. (c) Cell proportion on each lineage. (d) Distribution of the Pearson correlation coefficients between genes' pseudotemporal expression patterns fitted by Seurat+Lamian and those fitted by Harmony+Lamian. Pseudotemporal gene expression patterns were fitted as in TDE analysis. The correlation coefficient was computed for each gene using its fitted group-level curves. (e) Similar to (d) but the pseudotemporal curves are fitted as in XDE analysis. In other words, the correlation was analyzed within each sample group (female or male) separately. (f) The proportion of Seurat+Lamian-reported TDE genes ($FDR < 0.05$) that can be found in the top $100n$ Harmony+Lamian TDE genes ($n = 1, 2, \dots$) ("test results"). As a negative control, the gene list based on Harmony was also permuted to re-calculate the overlap ("permuted"). The vertical bars with "*" denote the $FDR < 0.05$ cutoffs for Harmony+Lamian in the three lineages. (g) Similar to (f) but for XDE analysis.

Figure S18. Lamian analysis of the HCA-BM data based on scVI integration and comparison with Seurat integration. (a-b) Cells plotted with top two latent variables from scVI, colored by clusters (a) or pseudotime (b). Pseudotime was inferred using top 10 latent variables output by scVI. (c) Cell proportion on each lineage. (d) Distribution of the Pearson correlation coefficients between genes' pseudotemporal expression patterns fitted by Seurat+Lamian and those fitted by scVI+Lamian. Pseudotemporal gene expression patterns were fitted as in TDE analysis. The correlation coefficient was computed for each gene using its fitted group-level curves. (e) Similar to (d) but the pseudotemporal curves are fitted as in XDE analysis. In other words, the correlation was analyzed within each sample group (female or male) separately. (f) The proportion of Seurat+Lamian-reported TDE genes ($FDR < 0.05$) that can be found in the top 100 scVI+Lamian TDE genes ($n = 1, 2, \dots$) ("test results"). As a negative control, the gene list based on Harmony was also permuted to re-calculate the overlap ("permuted"). The vertical bars with "*" denote the $FDR < 0.05$ cutoffs for scVI+Lamian in the three lineages. (g) Similar to (f) but for XDE analysis.

In the revised manuscript, we modified Section 2.1 on page 2 as follows to guide users.

"The input for Lamian includes (1) low-dimensional space representation, such as principal components analysis (PCA), of the scRNA-seq data from multiple samples that have been harmonized and embedded into a common space using methods such as Seurat, Harmony, or scVI, and (2) the normalized scRNA-seq gene expression matrices, and (3) sample-level metadata, such as covariate information corresponding to samples' characteristics or biological groups, and samples' batch indicators to allow for batch effect correction. We assume that the data harmonization is done by users and refer readers to a recent benchmark study (Luecken, M. D. et al., 2022) for guidelines on choosing the harmonization methods. The purpose of harmonization is to match cells of the same type across samples so that the same type of cells can be compared across samples. As such, it removes both biological differences of interest (e.g. the same cell type can have differential expression between two sample conditions which is removed by harmonization) and unwanted technical differences (e.g. batch effects) among samples. In downstream analyses where one is interested in studying biological differences across samples and conditions, Lamian will use the original normalized gene expression values rather than harmonization-corrected expression values, and it will use a regression framework to remove unwanted technical variations such as batch effects but retain biological differences across samples.

We also added the following in Discussion:

“For sample harmonization, we used Seurat to embed cells into a common low-dimensional space. One could also use other methods such as Harmony and scVI. For example, in our HCA bone marrow analysis, using Seurat, Harmony and scVI produced similar branching structure and differential genes (Fig. S17, S18). In real applications, different harmonization methods may perform differently. We recommend users compare different harmonization methods and choose the one most consistent with the existing knowledge. A systematic comparison of harmonization methods is beyond the scope of this study. Readers are referred to a recent benchmark study (Luecken, M. D. et al., 2022) for discussions on which harmonization methods to use under different conditions.”

2. For HCA-BM data, the authors mentioned they identified 6 cell clusters after applying TSCAN to the harmonized bone marrow data. However, as a complex tissue type, people can usually identify 10-20 or even more cell subpopulations. Is it an over-simplification of the problem by setting the number of cell clusters as 6 here? In addition, can the authors further instruct if Lamian can accommodate a large number of cell clusters, which form a tree with a large number of branches? [Editor: It would be particularly useful to comment on the scalability of Lamian.]

We thank the reviewer for this question. In our analysis, TSCAN was used to construct pseudotemporal trajectory. TSCAN uses a cluster-based minimum spanning tree approach to balance the bias and variance in order to achieve more accurate trajectory inference. In principle, this approach can handle a large number of cell clusters just like how a vanilla minimum spanning tree approach can handle a large number of vertices (tree nodes). The real problem in applications is that, without knowing the ground truth, how can one tell that the cluster number (i.e. number of tree nodes) is neither too big nor too small and the tree topology is at a right complexity level that represents real biological trajectory structures with enough data support rather than random noise. For that reason, TSCAN uses an elbow method to automatically determine the number of cell clusters (Z. Ji et al. *Nucleic acids research*, 2016). This elbow method analyses the relationship between the number of clusters and the percentage of data variance explained by clusters (i.e., $(\text{between cluster sum of squares})/(\text{total sum of squares})$). Then, it identifies the elbow's point of this function as the optimal cluster number. The elbow's point is optimal in the sense that setting a larger number of clusters will only lead to a marginal increase in the percentage of explained data variance. This method has been shown to work well in the original TSCAN article. In our HCA-BM analysis, the number of clusters 6 was determined using this automatic method.

It is certainly possible that the automatically determined cluster number is not optimal for capturing the underlying true trajectory topology. Therefore, TSCAN also provides an option for users to specify the number of cell clusters based on their existing knowledge about the biological system. Since Lamian allows users to construct the tree using TSCAN, it inherited this flexibility from TSCAN.

To demonstrate that Lamian is capable of handling more complex trees, we applied Lamian module 1 to the HCA-BM data by setting the number of clusters to 100 and 1000, respectively. As shown in Figure S2c (also shown below), **Lamian is capable of completing Module 1, including inferring the tree structures and pseudotime, automatically enumerating all branches and assessing the uncertainty of the branches, with reasonable computational efficiency.** For example, for 1000 cell clusters, it can be completed within 6.75 hours and 6.3GB memory (including the computation for repeated bootstrapping to evaluate detection rates of branches). **However, the larger the number of clusters, the more complicated the tree structure is, resulting in increasing levels of noise and lower detection rates for the branches.** As shown in Figure S2a,b (also shown below), the detection rates for most tree branches detected using 100 or 1000 cell clusters are below 0.5. This illustrates that even though Lamian is capable of constructing more complex trees, a key question in real applications is whether one can trust that the tree structure is real rather than random noise particularly when there is little or no prior knowledge about the underlying biological process. Consideration of reliability of the inferred tree structure often leads to simpler trees. This does not necessarily imply that the inferred tree fully captures the branching structure of the underlying biological process. Instead, it only reflects the fact that the available data can only provide enough

information to support robust conclusion on a relatively simple tree and there is not enough information to draw robust conclusions on a more complex tree structure.

Figure S2. Inferring tree structure (Lamian Module 1) in HCA-BM data by setting a large number of cell clusters in TSCAN. (a) Distribution of detection rates of all tree branches when the number of clusters is 100. **(b)** Same as (a) except that the number of clusters is 1000. **(c)** Computational time and memory usage for Lamian Module 1 with different numbers of clusters.

We have addressed this point in the revised manuscript as follows:

- (1) We added the following comments in Section 2.2.1.

“Note that although TSCAN is scalable to a large number of clusters as tree nodes and can handle more complex tree structures, increasing tree complexity can also introduce noise and produce many unreliable branches with low detection rates (Fig. S2). Therefore, we proceed with the three branches here as their presence is robustly supported by the available data and also consistent with known biology.”

- (2) We added the above figure to Supplementary Figure S2.

- (3) We added the following comments on scalability in the Discussion.

“In Lamian, TSCAN is used as the default method to construct pseudotime due to its flexibility and scalability. TSCAN uses the cluster-based MST approach to reduce the number of tree nodes (e.g. clustering 1 million cells into 1000 clusters will result in only 1000 tree nodes instead of 1 million tree nodes) and hence can handle a large number of cells. In terms of flexibility, while TSCAN by default determines the number of cell clusters automatically via an elbow method, users have the option to specify their own cluster number if they are not satisfied with the default cluster number. Increasing the cluster number may create a more complex tree with a more detailed view of the biological process. However, the increased complexity could also introduce noise and false branches (Fig. S2). In real applications, even though one may construct a more complex tree, a key question is whether one can trust that the tree structure is real rather than random noise. Answering this question is challenging when there is little or no prior knowledge about the underlying biological process. Lamian addresses this issue via bootstrap and detection rate. A low confidence tree branch can be reflected by its low detection rate. Based on our experience, applying this criterion often leads to relatively simple tree structure. This does not imply that the underlying tree structure is necessarily simple. Instead, it only reflects the fact that the available data can only provide enough information to support robust conclusion on a relatively simple tree and there is not enough information to draw conclusions on a more complex tree structure. If users have prior knowledge that supports a more complex tree structure, they can use the manual option provided by TSCAN to choose a larger cluster number to define more detailed tree structure. Besides cluster number, TSCAN also allows users to manually specify the order of cell clusters in the trajectory, providing another way to adjust the trajectory based on users’ prior knowledge. These options in TSCAN allow users to conveniently perform analysis on more complex tree structures. Once the tree topology is given, all the remaining analyses including those in modules 2 to 4 (differential topology, XDE, TDE, XCD, TCD) can be

carried out as usual. Users can also choose to replace the default TSCAN method using other methods. For example, our TB analysis illustrated Lamian using user-provided pseudotime. ”

3. As for computational burden, the authors mentioned Lamian is computationally tractable. However, with the example of HCA bone marrow dataset (32,819 cells and 8 samples), it requires 4.1 hours with 25 CPUs and 163 GB RAM (page 11, line 19-21), which doesn't seem much friendly to users, and cannot be handled with a laptop. Since Lamian is designed for multi-batch data, which could scale up to 100K or 1M cells coming from a larger number of samples, will Lamian still work? Some discussion on computational complexity is needed.

Thank you for this great question. As the reviewer said, scalability is an important need for today's single cell analysis. Lamian is indeed capable of handling large datasets. To demonstrate this, in this revised manuscript, we have added an analysis of a new TB data set with 337,191 cells. We have also updated the Lamian software package to improve its computational efficiency for large datasets. In particular, for atlas-sized data (defined here as a dataset with $>10^5$ cells) the upgraded Lamian now supports HDF5 file format to store the data and perform all the calculations, which significantly increases the computational efficiency.

We note that for handling large datasets with $\sim 10^5$ cells, depending on the computational task, it is not unusual to have a few days of computation time and hundred GBs of memory. For example, previous benchmark (Benchmarking atlas-level data integration in single-cell genomics. Luecken et al. 2021. Nature Methods, Extended Data Fig. 7) has shown that around half of the integration methods, such as Seurat v3, MNN and scGen, failed to integrate datasets with $>10^5$ cells in a 1-week and 1TB platform. Compared to those tasks, the computational burden of Lamian actually is not high.

Most importantly, Lamian is computationally more efficient than other alternative pseudotime methods for handling similar tasks. For the TB dataset with 337,191 cells from 184 samples, Lamian with the default permutation test can finish the XDE test (the most time consuming task) with 62.605 hour and 243.041 GB, while the other competing methods such as PhenoPath, condiments and tradeSeq are all timed out in a 7-day-max 400GB-max computational platform. Moreover, taking reviewer 1's advice, in our upgraded Lamian we also added an asymptotic chi-squared distribution based p -value option for differential expression test. This new option provided reasonable FDR control and substantially reduced the computation time. If users use this option, they can finish computation using 8.926 hours and 4.896 GB RAM. For the COVID dataset (55,953 cells) which is smaller than the TB dataset, Phenopath also timed out. Since XDE test is the most time-consuming module in the Lamian differential analysis framework, this result shows that Lamian is fully capable of dealing with large datasets with better computational efficiency compared to the other available methods.

In the revised manuscript, we updated the following figure (Figure S19) to reflect this new information.

Figure S19. Comparison of computational time and memory usage of different methods. (a) Computational time required for Lamian (both Lamian.chisq and Lamian.pm) and other methods. Values more than one week (168 hours) are marked as *timeout*. (b) Same as (a) but for memory usage in GB. Values more than 400GB are marked as *out of memory*. (c) Since the TB data analysis only involved Lamian modules 3 and 4, we further benchmarked computational time required by modules 1 and 2. COVID-19 data with 161 samples were used to generate datasets with different numbers of cells. (d) Same as (c) but for memory usage in GB.

4. This paper lacks a comparison with a recent competing method called *condiments* (<https://www.biorxiv.org/content/10.1101/2021.03.09.433671v1>).

Thank you for this great point. We have now included *condiments* in all the comparisons. These comparisons show that overall Lamian outperforms *condiments*. Please see the **updated Fig. 4, 5, 6** (*condiments* cannot provide results within 1 week and 400GB in this TB data), **S9, S11, S12, and S19**.

5. The authors showed “decreasing the number of cells decreased the detection rate”. This might be a limitation: as different datasets may vary in cell numbers, how can we believe the detection rate would be high enough to be different from random testing, especially when the dataset has limited number of cells?

This is a good question. It suggests partial misunderstanding of the “detection rate”. The detection rate is developed to help people judge to what extent they should trust the tree branch structure. If there are a limited number of cells, the data would not have enough information or provide enough evidence to support the existence of the branch, therefore the detection rate will be low. That is the purpose of the detection rate and analogous to the concept of variance, which is developed to characterize the uncertainty of a parameter estimate (e.g. sample mean). If the sample size is small, one has larger variance, indicating that there is large uncertainty associated with the parameter estimate. Thus, variance is a measure to describe uncertainty. It is not a limitation that variance increases when sample size decreases, as this is exactly what variance is trying to tell about the data. Similarly, the fact that detection rate decreases when cell number decreases is not a limitation of detection rate, since detection rate is a measure developed to reflect the fact that one will have lower confidence about the tree branch when there are not many cells supporting the tree inference. For detection rate, we do not provide a cutoff to tell users which detection rate is high enough (just like when people use variance to describe uncertainty of a sample mean, they often do not set a cutoff for variance). Our goal is to only provide a summary statistic to help users decide whether to further study a branch given the evidence in the single cell data and their existing knowledge about the biological system in question.

6. In the validation of XDE test (P7), the author mentioned the identified XDE genes are enriched in sex chromosomes, as the argument to show their method can accurately captured the DEGs between male and female. Did the author check whether the identified XDE genes on ChrX are the known genes escaping X-inactivation? If yes, this can further support the authors' results.

This is an excellent point. We have now conducted a new analysis to compare the XDE genes detected by our methods to the X-chromosome genes escaping X-inactivation. We collected the list of genes escaping X-inactivation (XCI) from a publication: Balaton, B.P., Cotton, A.M. and Brown, C.J., 2015. Derivation of consensus inactivation status for X-linked genes from genome-wide studies. *Biology of sex differences*, 6(1), pp.1-11. Paper link: <https://bsd.biomedcentral.com/track/pdf/10.1186/s13293-015-0053-7.pdf>. This paper aggregated three published studies that have examined XCI status of genes across the X chromosome,

We found that among the X-chromosome XDE genes between male and female identified using Lamian in the erythroid lineage, more than 95% are also the genes escaping XCI. This percentage is significantly higher than other competing methods. Similarly, in the lymphoid and myeloid lineages, the percentage of XDE genes escaping XCI in Lamian is also higher than the other competing methods, as shown in the following plot below. This analysis further supports our results.

We have now added this figure to **Supplementary Figure S12**.

Figure S12. Comparison of Lamian and other methods based on percentage of sex-associated XDE genes escaping X-chromosome inactivation (XCI) in HCA-BM data. XCI status is obtained from Balaton et al².

We also added the following sentence to Section 2.4.2:

“Additionally, Lamian also showed the largest overlap with genes escaping X-chromosome inactivation (XCI), further demonstrating its top performance in detecting sex-associated XDE (Fig. S12).”

7. XCD test is not very attractive in my opinion, just as the author mentioned “the change can be due to technical sampling bias or real biology”. In addition, the changes in cell cycle status or proliferation rate may be a major reason leading to the changes in cell density along pseudotime. Can the authors check whether there are any relationships between their detected changes in cell density and the distribution of cell cycle status along pseudotime?

We thank the reviewer for this question. We believe that XCD here is a typo, and the reviewer actually wanted to say “TCD” test instead of “XCD” test since (1) the sentence “the change can be due to technical sampling bias or real biology” is used both in the original manuscript and in the revised manuscript to discuss results from the TCD instead of the XCD test; (2) XCD test was able to find meaningful differences between

different sample groups in our real data examples (and hence useful), but TCD test did not result in easily interpretable results (and hence less attractive). Therefore, below we will respond with respect to TCD.

- (1) For TCD analysis (changes of cell density along pseudotime), we acknowledge that we did not find a dataset that produced easily interpretable biological findings. In HCA bone marrow, COVID, and TB data, we all found significant cell density changes along the pseudotime. However, it is unclear whether the cell density change was due to technical sampling bias (e.g. certain cell types are easier to sample) or real biology, which makes the interpretation challenging. Despite this, we cannot rule out that some users will need this function in their own data analysis. Therefore, we decided to provide the TCD function in Lamian for completeness of the package so that users will have a tool to support such analysis if they need it in their own projects.
- (2) Taking the reviewer's advice, we also checked whether there is any relationship between the detected changes in cell density by TCD test and the distribution of cell cycle status along pseudotime. We scored each cell's cell cycle status using three existing gene sets: cell cycle gene lists of S-phase (43 genes) and G2-M-phase (54 genes) retrieved from Seurat v.3.2.1, and the gene list of G1-phase retrieved from GSEA (https://www.gsea-msigdb.org/gsea/msigdb/cards/G1_PHASE, 15 genes). For each cell, S-, G2-M- and G1-phase scores were calculated using the average of the normalized log₂-transformed gene expression across genes in the corresponding gene list. In this way, with the cellular pseudotime, the pseudotemporal pattern of the cell cycle was obtained. Pearson correlation coefficients between cell cycle scores (B-spline fitting to get the same length as the pseudotime window in cell density) and cell density were calculated to assess the association between them. The boxplot below shows the distribution of the correlation coefficients (each dot is a sample). One can see that in all three real data examples (HCA-BM, COVID, TB), the cell density has low correlation with all S-, G2-M-, and G1-phase cell cycle scores (median Pearson correlation is around 0), suggesting that the cell density changes in these datasets cannot be explained by cell cycle. **We added this figure to Supplementary Figure S6.**

Figure S6. Relationship between cell cycle and cell density along pseudotime. The cell cycle gene lists of S-phase (43 genes) and G2-M-phase (54 genes) were retrieved from Seurat v.3.2.1. The gene list of G1-phase was retrieved from GSEA G1_PHASE (15 genes). For each cell, S-, G2-M- and G1-phase scores were calculated using the average of the normalized log₂-transformed gene expression across genes in the corresponding gene lists. In this way, with the cellular pseudotime, the pseudotemporal pattern of the cell cycle was obtained. For each sample, Pearson correlation coefficients between cell cycle scores and cell density along pseudotime were calculated. The boxplots show the distributions of correlation coefficients for different datasets and tree branches (each data point represents a sample).

Regarding XCD test, the fact that we found interesting differential cell densities in both COVID and TB data demonstrates the usefulness of this test. For example:

For COVID, we commented XCD in section 2.5, as follows.

*“The analysis of cell density using XCD test shows that the abundance of activated effector T cells is significantly increased in moderate compared to mild disease ($FDR = 1.38 * 10^{-61}$), Fig. 5d.”*

For TB data, it was reported that there are cell abundance changes between males and females (Reshef et al. Nature Biotechnology, 2022). Thus, we performed a XCD test using the TB data comparing male vs female samples, and found that the cell density along pseudotime is significantly associated with sample sex, which is consistent with the previous report by Reshef et al. This provides an example demonstrating that the XCD test can be useful in real data. **We commented it in section 2.6**, as follows.

“Consistent with previous report (Reshef et al. Nature Biotechnology, 2022), XCD test revealed a significant cell abundance changes between male and female along the pseudotime. T cells from females were more enriched towards naive status (early pseudotime) compared to T cells from males. By contrast, male cells were more enriched towards terminal activation status (late pseudotime) (Fig. 6c, $p=10^{-5}$).”

8. In Section 2.1.3, the authors mentioned unsupervised k- means clustering is applied to DE genes (page 4, line 1). Does the result differ much if other clustering methods are used, such as Louvain or GMM?

We thank the reviewer for this question. We want to clarify that the main focus of module 3 is to appropriately model the temporal patterns of the genes and identify the differential genes with good control of Type I error. Unsupervised *k*-means clustering is an auxiliary function used to group DE genes based on their similar DE patterns to help users more conveniently explore different temporal patterns. The clustering itself will not change which genes will be called as DE genes and thus will not change our main results (i.e. the DE genes list), and therefore it is also not the focus of our study. However, we agree that users may want to try different clustering methods in their downstream analysis. Therefore, we have now added the Louvain and Gaussian mixture model (GMM) to the clusterGene() function in Lamian so that users have these options in addition to *k*-means clustering. We have also compared the clustering results of *k*-means, GMM, and Louvain in both the TDE and XDE tests using the HCA bone marrow data. Results showed that the clustering results are largely consistent in terms of the main temporal patterns identified from the analysis. In real applications, users can choose which clustering method to use according to their preference or practical situation.

We have added the following new supplementary figures (Figs. S7,S8). The confusion matrices suggest that the clustering results are overall similar and the similarity has an increasing trend when the signal strength is higher.

Figure S7. Comparison of k -means and Gaussian mixture model clustering results of differential genes. (a)-(d) Clustering XDE genes in the *in silico* spike-in datasets used in Fig. 4b-d. Heatmaps show the confusion matrix of k -means and Gaussian mixture model (GMM) clustering results for data with increasing signal strength ((a) to (d) represents signal strength from 1 to 4, respectively). (e)-(h) Clustering TDE genes in the *in silico* spike-in datasets used in Fig. S13. Similar to (a)-(d) but on TDE genes. Overall, the clustering results from the two clustering methods become more similar as signal strength increases.

Figure S8. Comparison of k -means and Louvain clustering results of differential genes. (a)-(d) Clustering XDE genes in the *in silico* spike-in datasets used in Fig. 4b-d. Heatmaps show the confusion matrix of k -means and Louvain clustering results for data with increasing signal strength ((a) to (d) represents signal strength from 1 to 4, respectively). (e)-(h) Clustering TDE genes in the *in silico* spike-in datasets used in Fig. S13. Similar to (a)-(d) but on TDE genes. Overall, the clustering results from the two clustering methods become more similar as signal strength increases.

We have added the following sentences to Section 2.4.1:

“Using Gaussian mixture model and Louvain clustering yielded similar results (Figs. S7, S8).”

9. In Section 4.1 Data (page 14, lines 5-9), the authors stated the raw HCA-BM data consist of 290,861 cells, but after screening, the final data for analysis consist of only 32,819 cells, which means nearly 90% of cells are screened out. The authors should justify why such a strict screening is necessary and provide descriptive plots to illustrate the choice of 5,000 reads, 1,000 expressed genes and 10% mitochondrial gene expression are reasonable thresholds.

This is a good point. The following descriptive plots show the number of expressed genes, the proportion of mitochondrial reads and number of reads in the dataset along with the QC cutoffs applied to them. We excluded cells with more than 10% mitochondrial reads. This filter is quite standard and most of the cells are retained after the filtering. Thus, this is not a strong filter and is not the main reason that nearly 90% of cells are screened out.

The number of expressed genes cutoff was set as 1000, which is a strong filter based on the distribution. This filter removes the majority of cells. However, if we apply a lower cutoff, the cell type annotation and the low dimension representation became noisy and did not clearly capture known biology. For example, setting the number of expressed gene cutoff to 500 allowed us to retain 255,547 cells. However, the three cell differentiation lineages were unclear in both Seurat- and Harmony-integrated results. Moreover, both methods were not able to identify any cluster annotated as hematopoietic stem cell (HSC), and a number of clusters were annotated as NA (unidentifiable cell types, colored by grey) (Fig. S20 e,f). These indicate that for this well-studied system, setting the cutoff to be 500 was not a good choice. Similarly, we also tried to set the cutoff to be the median number of expressed genes across cells. The results were also noisy and did not clearly capture the three lineages (Fig.S20 g,h). Since HCA-BM data serves as an example dataset for method illustration, we decided to apply the stringent cutoff 1000 so that we can capture the known biology and hence use the known biology to benchmark downstream differential expression analysis.

Figure S20. Quality control (QC) plots and the integration results from Harmony and Seurat with relaxed quality filters. (a)-(d) Current filtering criteria. **(e)-(f)** Integration results from Harmony (e) and Seurat RPCA (f) after relaxing the number of expressed genes per cell cutoff from 1000 to 500. In Harmony (e), 6/27 clusters are unidentifiable cell types (NA) which occupy 10831 out of 255547 cells. In Seurat (f), 7/28 clusters are unidentifiable cell types (NA) which occupy 11409 out of 255547 cells. **(g)-(h)** Integration results from Harmony (g) and Seurat RPCA (h) after changing the number of expressed genes cutoff from 1000 to the median number of expressed genes. In Harmony (g), 9 out of 29 clusters are NA, i.e. 11995 out of 164917 cells. In Seurat (h), 8 out of 27 clusters are NA, i.e. 9673 out of 164917 cells.

For the number of reads cutoff, the number of reads and the number of expressed genes are highly correlated. We choose 5000 as the reads cutoff simply because most cells with >1000 expressed genes have >5000 reads (see figure below). In other words, 5000 reads cutoff is not a strong filter on top of the number of expressed gene filter and does not influence the filtering results much.

We have now added this discussion as a new section to supplementary notes, and added the analysis results to Supplementary Figure S20. In addition, we have added the following to the Data section.

“See Supplementary Notes and Fig. S20 for a more detailed discussion of filtering parameters and additional quality control (QC) plots.”

10. It is unclear what Lamian stands for as an abbreviation. Some explanation is needed. Both Lamian (majority) and Lamain were used in the text.

Our method Lamian is inspired by the process of making Lamian, a traditional Chinese hand-pull noodle. Here we show one way to make Lamian noodles. First, we make a firm dough using wheat flour, and then pull many thinner noodles from the dough. Finally, we coat the noodles with extra flour so that the noodles don't stick to each other.

In our model, the population-level pattern, the orange curve, represents the overall gene expression dynamics in all samples, which resembles the dough. The sample-level gene expression dynamics of each sample is the population pattern plus some sample-level random variation, which resembles the thinner noodles coming from the dough. Finally, the observed cell-level gene expression values are generated by the sample-level pattern plus some cell-level random variation, which resembles the flour. Because the whole data generating mechanism resembles the process of making Lamian noodles, we name our statistical framework as “Lamian”. We created a plot to summarize this idea. Since this is not directly relevant to science, we added this explanation to the Lamian website (<https://github.com/Winnie09/Lamian>).

Also, we have fixed typos and changed “Lamain” to “Lamian”.

11. Regarding impact: Nature Methods is appropriate for the manuscript given the novelty of the statistical approach, the broad application of the paper, and convincing biological examples.

We sincerely thank the reviewer for recognizing the potential impact of this work.

12. Regarding reproducibility: All data are publicly available. Code has been deposited to GitHub.

We thank the reviewer for this comment.

Reviewer Comments, Second Version:

Reviewer #1 (Remarks to the Author: Overall significance):

The response letter is very comprehensive with detailed answers to each question. In the revised manuscript, the authors have addressed most of my concerns. Specifically, the authors now added the asymptotic test, which seems to be very helpful for computational efficiency.

We have two minor points:

For Q1, "First, can Lamian be used as a downstream pipeline for other popular pseudotime inference methods, e.g., Slingshot and Monocle3?" It seems like the Lamian is based on TSCAN and I think this is fine since different pseudotime methods may have very different properties. To benefit other method developers, I wonder if the authors can briefly summarize some other popular methods and point out why they are not appropriate for Lamian/some possible directions to include them in Lamian. For instance, Slingshot also relies on MST but is not currently applicable for Lamian.

For Q5, "Module 2 uses the two-sample t test to compare branch proportions between two conditions. The t test is inappropriate for two reasons: (1) branch proportions are not normally distributed; (2) proportions of different branches are not i.i.d." We are happy to see that the authors now add the multinomial regression model as an alternative. However, for the statements from the authors, "For example, a typical report of the result would look like "changing sample covariate X1 by one unit will result in an increase of log ratio of cell proportions between the myeloid lineage (branch 1) and lymphoid lineage (branch 2) by β_1 ". It is not straightforward to know whether the cell proportion of the myeloid lineage (branch 1) is increased or decreased, since the answer will also depend on how branch 3 or other branches' cell proportion change relative to branch 1 and branch 2 (i.e. the ratios of branch3/branch2, branch4/branch2, etc.)." I think if we have the baseline rate, the odds ratio can be converted into a percentage. Odds ratio has been used in subjects like Biostat/Epidemiology for quite a long time, and I hope it can be appropriately interpreted by biologists, the potential users of Lamian

Reviewer #2 (Remarks to the Author: Overall significance):

please see my attached file.

Reviewer #3 (Remarks to the Author: Overall significance):

This revision has addressed all of my comments. They added more results from different data integration strategies, improved Lamina's computational efficiency for large datasets, added new comparisons with competing method condiments, and deposited codes to GitHub. We have run through all the functions of the R package "Lamian" on the provided cleaned dataset. They provide a very clear document for the package. We are able to replicate all the results following the document. Lamina is very useful in our ongoing single cell projects with multiple samples and conditions. This comprehensive statistical package will fill the gap in understanding dynamic changes of gene expression in population-based single cell studies.

Reviewer #4 (Remarks to the Author: Overall significance):

This revision has largely improved the quality of the manuscript. I appreciated the addition of a new case study, and in depth benchmarking with other methods, which are convincing. Lamian provides a new approach for pseudotime analysis that will complement (or surpass) other methods in the field.

My comments below are fairly minor to improve the presentation of the manuscript following this revision.

Reviewer #4 (Remarks to the Author: Strength of the claims):

1 - Now that a study demonstrating that Lamian can adjust for batch effects, you should consider adding this information in the abstract, as I think it will be of interest to many researchers.

2 - The added description and justification on harmonisation (2.1) seems a bit out of place ('the purpose ...'). Consider moving this to the Methods section.

3 - The purpose of the first case study (HCA-BM) is quite clear as it illustrates all modules of Lamian and provide some benchmarking (2.2 - 2.4). For the other case studies, the purpose is less clear. Consider adding a short introduction to both to explain what are the characteristics of these studies (eg. COVID19: detecting differences between covariates; TB: accounting for batch effects).

4 - The discussion meanders a bit. Consider tightening the text (e.g 'Consequently, it is not always clear what it means if one says that "a gene is differential along a trajectory lineage" since the lineage does not always exist in all trees.'). The computational part should be moved to the results section, with a table stating # cells, samples, compute time) to ease of reading. Generally, I did find that the added text did not flow very well with the rest of the original version.

5 - I do think that the authors should mention the origin of the name of the method either by text of picture (graphical abstract) in the manuscript!

Minor:

Introduction: 'Recently, PseudotimeDE is developed' -> has been proposed ?
p2 'In summaryvariability'. This information could be more impactful if moved before the description of other methods ('Although'). the added text is a bit clunky.

Section 2.1.3 makes it sound like the two tests come from different models (I think this is because one sentence starts with 'First', but no 'Second' follows).

'separated' appears twice (Fig 4 & 4.1.4)

Fig 6 caption: only mention once at the end of the caption 'Methods that cannot produce output within 1 week and 400 GB RAM are not shown'

Reviewer #4 (Remarks to the Author: Reproducibility):

I found that the GitHub repository for reproducing the analyses is quite obscure (https://github.com/Winnie09/trajectory_variability). As a user, I would not know where to start. Could you reorganise and rename the folders, and perhaps give a brief summary in the readMe file about what each case study is about (see my comments above regarding the aims of each case study).

Response to Reviewers:

Reviewer #1 (Remarks to the Author: Overall significance):

The response letter is very comprehensive with detailed answers to each question. In the revised manuscript, the authors have addressed most of my concerns. Specifically, the authors now added the asymptotic test, which seems to be very helpful for computational efficiency.

We have two minor points:

For Q1, “First, can Lamian be used as a downstream pipeline for other popular pseudotime inference methods, e.g., Slingshot and Monocle3?” It seems like the Lamian is based on TSCAN and I think this is fine since different pseudotime methods may have very different properties. To benefit other method developers, I wonder if the authors can briefly summarize some other popular methods and point out why they are not appropriate for Lamian/some possible directions to include them in Lamian. For instance, Slingshot also relies on MST but is not currently applicable for Lamian.

We thank the reviewer for the suggestion. Now we have added several sentences in the Discussion section to summarize why some other popular methods, including Monocle2, Monocle3, and Slingshot, are not supported in Lamian Modules 1 and 2, while noting that the pseudotime produced by these methods can still be used with Lamian Modules 3 and 4:

Besides TSCAN, one also has an option to use user-provided pseudotemporal trajectories as illustrated in the TB analysis. In fact, one may use Lamian modules 3 and 4 as downstream analysis tools for other pseudotime methods such as Monocle2, Monocle3 and slingshot. However, Lamian modules 1 and 2 which construct trajectory and quantify its uncertainty and variation currently do not support other pseudotime methods due to various issues including scalability and implementation challenges (Supplementary Notes). For example, slingshot, a popular MST method similar to TSCAN, does not scale well to bootstrap due to its time-consuming principal curve fitting. For Monocle2 and Monocle3, modifying Lamian’s trajectory topology uncertainty module to support them is non-trivial due to lack of interoperability between the data structures used by different methods to represent trajectory topology. A future direction is to tailor these methods to improve their scalability and/or interoperability to allow their seamless connection to Lamian modules 1 and 2.

We have also added a section in the Supplementary Notes to discuss the above issue in detail, as follows:

7 Applying other pseudotime methods in Lamian

Lamian Modules 3 and 4 allow one to use pseudotemporal trajectories generated by other methods to replace TSCAN as input for multi-sample differential gene expression and differential cell abundance analysis. Our online user manual (https://winnie09.github.io/Wenpin_Hou/pages/Lamian.html) provides a step-by-step tutorial for this using slingshot as an example. Lamian Modules 1 and 2 which quantify trajectory topology uncertainty and changes currently do not support the other pseudotime methods due to issues such as scalability and other technical complications. For example, slingshot is another popular MST-based method similar to TSCAN, but it has a more time-consuming principal curve fitting step. It takes TSCAN 0.158 seconds to construct the pseudotime (running the main function `TSCANorder()` after model-based clustering) with the HCA-BM data, while slingshot needs 857.102 seconds (running the main function `slingshot()` after model-based clustering). To quantify tree uncertainty, we apply the same trajectory construction method repeatedly to bootstrapped datasets. For 100 bootstrap datasets, it will take more than 23 hours for slingshot to finish. Increasing the number of bootstraps will further increase the time. Furthermore, each pseudotime method (e.g. slingshot, Monocle2, Monocle3, etc.) has its own data structure and trajectory algorithm. Since we are the developer of TSCAN, we are able to modify its code to be compatible with Lamian. For the other pseudotime methods, directly calling these methods or modifying their code to fit into Lamian is difficult as we do not have complete information on how to access and operate on the relevant data structure (e.g. pseudotemporal trajectory topology) in their code. Implementing them from scratch in Lamian is also non-trivial. For this reason, our Modules 1 and 2 currently do not support these methods, but it will be useful to further investigate in the future to find ways to support them in Lamian Modules 1 and 2.

In addition, we have added a start-to-finish tutorial to demonstrate how users can use slingshot's pseudotime in Lamian's Modules 3 and 4. Please see the online manual: https://winnie09.github.io/Wenpin_Hou/pages/Lamian.html

For Q5, "Module 2 uses the two-sample t test to compare branch proportions between two conditions. The t test is inappropriate for two reasons: (1) branch proportions are not normally distributed; (2) proportions of different branches are not i.i.d." We are happy to see that the authors now add the multinomial regression model as an alternative. However, for the statements from the authors, "For example, a typical report of the result would look like "changing sample covariate X1 by one unit will result in an increase of log ratio of cell proportions between the myeloid lineage (branch 1) and lymphoid lineage (branch 2) by β_1 ". It is not straightforward to know whether the cell proportion of the myeloid lineage (branch 1) is increased or decreased, since the answer will also depend on how branch 3 or other branches' cell proportion change relative to branch 1 and branch 2 (i.e. the ratios of branch3/branch2, branch4/branch2, etc.)." I think if we have the baseline rate, the odds ratio can be converted into a percentage. Odds ratio has been used in subjects like Biostat/Epidemiology for quite a long time, and I hope it can be appropriately interpreted by biologists, the potential users of Lamian

We thank the reviewer for raising this point. We agree with the reviewer that the odds ratio can be converted into a percentage. Indeed, many of our biologist collaborators understand the concept of odds ratio. Despite this, we found that in practice it is inconvenient to do such a conversion when there are multiple branches, making it less efficient to interpret the analysis results in real time when we and collaborators sit together to explore the data.

Below we use a simple example to demonstrate the challenge. Suppose one compares a treated sample with a control sample, and the samples have three branches.

In the control sample, *Sample1*, the cell proportions in three branches are: 0.3 (Branch 1), 0.1 (Branch 2), 0.6 (Branch 3). Using Branch 3 as the baseline branch, the odds for Branch 1 against Branch 3 (baseline) is $0.3/0.6 = 0.5$.

In the treated sample, *Sample2*, the cell proportions in three branches are: 0.2 (Branch 1), 0.6 (Branch 2), 0.2 (Branch 3). Using Branch 3 as the baseline branch, the odds for Branch 1 against Branch 3 (baseline) is $0.2/0.2 = 1$.

Comparing the treated sample to the control sample, the odds ratio for Branch 1 against Branch3 (baseline) is $1/0.5 = 2$. In other words, the **odds** for Branch 1 against baseline is **increased** in the treated sample (*Sample 2*) compared to the control sample (*Sample 1*). However, the actual **cell proportion** for Branch 1 is **decreased** from 0.3 in the control sample (*Sample 1*) to 0.2 in the treated sample (*Sample 2*).

Many of our collaborators want to study cell proportions (e.g. COVID-19 disease severity is associated with lymphopenia, that is, decreased lymphocyte proportion in blood), and they want to look at the cell proportion individually for each branch. For them, looking at the odds ratio (i.e., the regression coefficients of the multinomial logistic regression) does not "directly" tell one how cell proportion of each branch will change from one sample to another, as illustrated above. Although we can convert odds ratio to cell proportion, such a conversion often cannot be processed in real time by human brain when there are many categories or branches, and we will need to use a computer or a calculator for this calculation. This decreases the efficiency in information processing and communication when we and our collaborators sit together to explore data and discuss analysis results. This is analogous to a situation where many people prefer to use \log_2 over natural logarithm

(ln) in microarray or RNA-seq differential gene expression analysis. When one sees a \log_2 (fold change) = 2, 3, or 4, one can immediately convert the result to the original scale which gives fold change = 4, 8, or 16. But when one sees $\ln(\text{fold change}) = 2, 3$ or 4, one cannot quickly convert it to the original scale in the brain (i.e. what is the value for fold change = e^2, e^3 , or e^4 ?).

Thus, while multinomial logistic regression may be more statistically rigorous for cell proportion analysis, it also incurs cost for information processing and communication which can decrease the efficiency of data exploration.

To help one more conveniently explore the cell proportion changes for each branch and circumvent the limitations of t-test, we have now replaced the two-sample t-test by a binomial logistic regression fitted for each individual branch. For each branch, the regression models $\log(p/(1-p))$ as a function of the sample covariates. Here p is the underlying true branch cell proportion for the branch, and $1-p$ is the cell proportion of all other branches combined, $\log(p/(1-p))$ is log odds. Compared to multinomial logistic regression, the interpretation of the branch-specific binomial logistic regression results is more straightforward for studying cell proportion changes across samples. A positive regression coefficient means that log odds of cell proportion of the given branch (i.e., $\log(p/(1-p))$) has increased from one sample group to another sample group. It also means that cell proportion (i.e. p) of the branch has increased since log odds is a monotone function of cell proportion. Similarly, a negative regression coefficient means that cell proportion is decreased. This can be interpreted instantaneously by human brain. Thus, binomial logistic regression and multinomial logistic regression each has its own pros and cons.

Our results in **Figure 2h** show that both the binomial logistic regression (fitted for each branch separately) and multinomial regression (fitted for all branches jointly) can provide reasonable performance – they controlled the type I error rate and had increasing power with increasing changes in branch cell proportions.

For developing a software tool with good user experience, statistical rigor certainly is an important factor to consider, but it is not the only factor. Software developers will also need to consider and balance many other factors. In our case, how to present the data and analysis in a way that users can conveniently interpret is an important factor we consider. For this reason, we provide both the binomial logistic regression (fitted for each branch separately) and the multinomial logistic regression (fitted for all branches jointly). Users can choose which method to use based on their own needs. For users who want to look at odds ratios in multinomial logistic regression and are experienced at translating odds ratios to cell proportions, they can use multinomial logistic regression. For users interested in how cell proportion of a given branch changes across samples (i.e. increases or decreases), they can use binomial logistic regression to more conveniently see the patterns in the data.

We have now added a section (“**8. Options for analyzing branch cell proportion changes**”) in **Supplementary Notes** to discuss this point to keep Lamian users informed of the pros and cons of these two methods. Readers are referred to the Supplementary Notes in the main manuscript section 2.1.2 as follows:

covariates. To facilitate convenient exploration of each individual branch, one can use a binomial logistic regression to evaluate covariate-associated branch cell proportion changes for each branch separately. Alternatively, users can also use a multinomial logistic regression to analyze covariate-associated changes of cell proportion ratios between branches by considering all branches jointly (Supplementary Notes). These regression-based methods allow one to identify tree topology changes between

This is also briefly discussed in the main manuscript Methods section “**4.4 Tree variability across samples and differential topology analysis**”.

Reviewer #2 (Remarks to the Author: Overall significance):

I appreciate that the author did in-depth analyses to address most of my questions / comments.

We really appreciate the reviewer's comments.

1. For the second question, the benchmark with PhenoPath and Monocle 2 is comprehensive and I am pleased to see their efforts on it. I also buy most of the authors' points. However, my original suggestions on "providing a supplementary table to summarize existing pseudotime analyses tools and how lamina fills this gap" will be still helpful. It will be also important for the author to respond on my original comment on "Fundamentally, inclusion of the sample covariate when performing a generalized linear regression is a rather conventional practice in biostatistics and thus the novelty of this approach is limited."

We thank the reviewer's suggestion on providing a supplementary table. In the Supplementary Table **TableS1_methods.csv**, we have summarized existing pseudotime analysis tools and statistical tools that are top performers in pseudotime, differential expression, or cell abundance analyses, including Monocle2, Monocle3, tradeSeq, slingshot, TSCAN, limma, milo, DAseq, pseudotimeDE, condiments, and PhenoPath. We have compared our method Lamian with these methods in different aspects of the analyses and clarified how Lamian fills the gaps in the pseudotime analyses. As the table shows, among all the existing popular methods for pseudotime analyses, only Lamian takes into account multi-sample variation (unlike monocle2, monocle3, condiments, etc. which group samples in the same condition altogether) and also provides a comprehensive package of functions to support tree construction, tree structure uncertainty evaluation, and XDE/TDE/XCD/TCD analyses. Also, Lamian's XDE analysis deals with arbitrary differences between conditions, which may be nonlinear functions of pseudotime (unlike PhenoPath which assumes that gene expression changes linearly). Our simulation and real data analysis results have also shown that Lamian has an overall superior performance compared to existing methods because the inclusion of sample-level variability.

Regarding the second comment on novelty, we define the novelty of our study as follows:

- **In terms of single-cell pseudotime analysis, we are the first to systematically illustrate, benchmark and report the serious impact of ignoring sample-level variability on downstream differential expression analysis.** We have shown that ignoring sample-level variability by existing methods can create a substantial number of false discoveries. This is a serious issue that so far has received little attention in the single-cell pseudotime community and literature.
- **To address this gap, we developed Lamian, the first comprehensive and statistically-rigorous framework that systematically addresses different sources of variability, particularly sample-level variability, in the multi-sample pseudotime analysis.** It offers a comprehensive package of solutions to analyzing trajectory topology uncertainty, differential topology, differential gene expression and differential cell abundance. The regression model mentioned by the reviewer is only one component of this whole systematic framework.
- **For the regression model per se, the notion that all regression models are basically the same and therefore using regression is not novel oversimplifies the reality.** First, while including sample covariate in regression is a conventional practice in biostatistics, **sample covariate is only one component of the Lamian model. The Lamian model also has many other important components that are not considered in this review comment**, including how to handle sample-level and cell-level variability via multi-level random effects, modeling random effects of functional curves, and the algorithm to efficiently fit the model. Second, there are a wide variety of different regression models and they have different complexity levels. Unlike linear regression (LM) or generalized linear model (GLM) which can be conveniently handled using standard statistical software tools (e.g. lm or glm function in

R), the model used by Lamian is a functional linear mixed model that cannot be handled directly using existing software. The Lamian model contains both fixed effects (to model effects of sample covariates) and two-levels of random effects (to model sample-level and cell-level random variations). The sample-level random effects, in particular, model the random variation of functional curves (rather than a scalar random variable). **For differential pseudotime analysis, Lamian is the first method that models the variability and difference of functional curves of pseudotemporal gene expression patterns across multiple samples using mixed effects models.** The model offers flexibility to handle non-linear pseudotemporal patterns. Compared to LM and GLM with only fixed effects, fitting the functional mixed effects model in Lamian is not straightforward due to the multiple layers of random effects and model complexity. In fact, Lamian cannot be directly fitted using existing statistical software. In order to be able to use this model, we have developed a computationally efficient algorithm based on expectation-maximization (EM) that can fit the model to large single cell datasets. The model fitting algorithm is presented in **Supplementary Note Section 1**. As one can see from that section (or even better – try to derive and implement it by oneself), developing and implementing this algorithm is non-trivial. **Collectively, formulating and implementing the functional mixed effects model in Lamian require substantial amounts of new intellectual work and cannot be achieved through a simple application of existing tools.** This may not be readily appreciated if one only looks at the name “regression” without looking into technical details.

- Even if one believes that the mixed effects model is not new, **the application of such a model to the multiple-sample differential pseudotime analysis is new.** Many influential methods and tools in computational biology and bioinformatics are applications of well-known models or algorithms to new problems. For example, dynamic programming is not new, but its novel application to sequence alignment resulted in the influential Needleman-Wunsch and Smith–Waterman alignment algorithms. Similarly, minimum spanning tree is an old concept in graph theory, but its novel application to single-cell trajectory analysis resulted in the influential Monocle method. One would not say that Needleman-Wunsch/Smith-Waterman algorithms and Monocle have limited novelty simply because the dynamic programming and minimum spanning tree are “rather conventional practice” in computer science and operations research. On the contrary, it is well-recognized that **these algorithms are highly novel because they have made new connections between existing methods and important but unsolved problems.** For differential pseudotime analysis, we are in a similar situation. Here, the state-of-the-art methods ignore sample-level variability in multi-sample analysis. This can result in a large number of false positives in the literature (e.g. thousands of false discoveries in our example datasets), which is a critical problem that need to be addressed if we as the scientific community is serious about addressing the replicability crisis. **By connecting the functional mixed effects model to this important but unsolved problem, Lamian provides a novel solution to the problem.** This new solution will provide the community with **a tool to change the current unsatisfactory state of data analysis to a new state** where one can substantially reduce false discoveries and increase the power for detecting true differential signals. We would argue that this is a significant novel development of differential pseudotime analysis rather than a development with limited novelty.

Given that the manuscript is now considered by *Nature Communications* and *Communications Biology* instead of *Nature Methods*, and note that the recommendations from the other reviewers were *Nature Communications* and *Nature Methods*, we believe that the novelty summarized here is well above the bar of *Nature Communications* and *Communications Biology*.

In the revised **Discussion** section, we have now edited the first paragraph to describe how Lamian fills in the current gap of multi-sample pseudotime analysis as follows:

3 Discussion

In summary, Lamian provides a systematic solution to multi-sample pseudotime analysis capable of detecting topology, gene expression and cell density differences between different conditions. In biomedical research, while making new discoveries is exciting, ensuring that the discoveries are real and replicable is equally important. One challenge the scientific community faces today is that many “significant” findings cannot be replicated or validated in independent studies. One important contributor to this problem is the flawed statistical analyses which can produce a large number of false discoveries. Such irreplicable false discoveries can be detrimental by distracting investigators from real signals and misleading subsequent research efforts, resulting in substantial waste of precious human and financial resources. In the context of pseudotime analysis, our results demonstrate that, due to lack of appropriate consideration of cross-sample variability, existing pseudotime methods can report thousands of false differential genes in null simulations where the data do not contain any true differential signals. This highlights a critical gap in the pseudotime literature and an open challenge that needs to be addressed. Lamian fills in this gap by introducing a comprehensive statistical framework, including a functional mixed effects model, to account for cross-sample variability in the multi-sample differential pseudotime analysis. In order to benchmark this new method, we applied it to both simulated and real data. We note that the analyses of three real datasets (HCA-BM, COVID, TB) mainly serve the purpose of illustrating and evaluating Lamian, and that making new biological discoveries *per se* is not the focus of this study. The orthogonal information (e.g. sex chromosome genes) and large sample size (e.g. COVID and TB data) available in these data make it possible to objectively and robustly compare different methods by quantifying the overlap with orthogonal information or between independent data partitions. Our results in these simulated and real data show that the new solution provided by Lamian substantially outperforms other existing methods to prioritize true discoveries and filter out false discoveries that are not generalizable to new samples.

2. Firstly I apologize since it is my fault to use the words “at all” and “ever” in my comments on “Overall, the author needs to justify why ...”. — there is no intention on my side to be negative for authors’ scientific work. My goal as a review is to provide direct, critical and honest comments and ensure papers have biological relevance rather than ad hoc computational exercises as it is not published for a pure statistical journal. I appreciate they addressed most of my comments and explained why comparing male/ female is meaningful. These points they listed should be included in the main text to “justify why the HCA- BM dataset was used for the analyses and why comparing the male and female is meaningful and interesting”. In addition, I appreciate the authors’ efforts to find the papers on the sex difference on HSCs. My previous point is that I don’t “expect the hematopoiesis to differ across men and women.” The paper you listed are most about “hematopoietic stem and progenitor cells” which is relevant but not directly related to hematopoiesis. Providing proper justification for the choice of your dataset in the analyses is important. This applies to the analysis for the Covid19 dataset too. You can justify the choice based on your response to my comments which are most computational. In the long term though, I would suggest the author to really take efforts to understand relevant biology to make a real impact with your computational tool. This is a good piece of advice for anyone who moves into biology from outside.

We greatly appreciate the comments from the reviewer. We are grateful for your critical input which has helped us to substantially improve this manuscript.

We have now incorporated our previous responses in the revised manuscript to justify why HCA-BM dataset was used and why we choose to analyse the male/female differences. In addition, we would like to reiterate that because sex difference was understudied in the past in many scientific investigations, a lot remains unknown about such differences. Therefore, the lack of prior knowledge about sex difference in a particular biological system (e.g. hematopoiesis) should not be interpreted as there is no sex difference in the system. For the sex difference analysis in the HCA bone marrow dataset, if there is no real sex difference, then what we found would be pure noise and a priori one would not expect that random noise obtained from such an unbiased genome-wide analysis to be enriched in sex chromosomes. The fact that the XDE genes we found are indeed enriched in sex chromosomes suggest that they contain real differential signals between the sexes.

Below we summarize the revisions we have made in the manuscript to explain this and justify the use of HCA-BM data:

Section 2.2

2.2 Lamian estimates tree topology stability and accurately detects differential tree topology

We begin with illustrating Modules 1 and 2 of Lamian using a Human Cell Atlas (HCA)^{37,38} 10x Genomics scRNA-seq dataset, referred to as HCA-BM, consisting of bone marrow samples from 8 donors (4 females and 4 males) and a total of 32,819 cells. Bone marrow contains hematopoietic stem cells (HSCs) differentiating into different blood cell types, creating a natural branching structure. This dataset along with the existing biological knowledge about this system therefore can be used to demonstrate and evaluate Lamian's ability to analyze a trajectory with branches.

Section 2.3.2

2.3.2 Detect differentially expressed genes associated with a covariate along pseudotime using HCA-BM data

Next, we tested whether there are differential gene expression patterns along pseudotime associated with sex as a covariate (Module 3: XDE test). Currently, which genes are sex-associated XDE genes in this system is not completely known. However, we reasoned that if there is no sex-associated XDE gene, then any XDE gene reported by the algorithm would be noise, and a priori one would not expect genes that are random noise to be associated with sex chromosomes. On the other hand, if XDE genes reported by Lamian in a genome-wide analysis are found to be enriched in sex chromosomes, it would suggest that sex-associated XDE genes exist and the algorithm is able to detect true XDE signals. For each gene, Lamian reports three

Note that our analysis later in this section showed that the XDE genes found by Lamian are indeed enriched in sex chromosomes, suggesting that sex-associated XDE genes do exist in this HCA-BM dataset.

Section 2.4

2.4 Lamian is more powerful than existing methods to detect differences while controlling the FDR by accounting for sample-level variation

In this section, we demonstrate that Lamian is more powerful than existing methods to detect gene expression differences that are associated with a covariate (Module 3: XDE test). Robustly comparing methods requires datasets with a sufficiently large number of known differential and non-differential genes to serve as the ground truth. Unfortunately, such datasets are not widely available. To address this, we combine simulations with the real HCA-BM data for method evaluation. The HCA-BM dataset is unique in that its male and female samples allow a between-sex comparison. Since there are many sex chromosome genes, the enrichment of sex-associated XDE genes in sex chromosomes can provide an objective and relatively robust benchmark to compare different methods. Thus, the HCA-BM data is used in this article for both method demonstration and evaluation.

For the COVID dataset, we also provided explanation on why it was used:

2.5 Lamian analysis of COVID-19 scRNA-seq data identifies differential CD8 T cell transcriptional programs during a critical stage of disease severity transition

To further demonstrate and evaluate Lamian's ability to detect differences associated with sample covariates, we applied Lamian to a COVID-19 peripheral blood mononuclear cell (PBMC) 10x Genomics scRNA-seq dataset obtained from a recent study⁴³. The COVID-19 disease severity of a patient may progress from mild to moderate to severe. It was reported that the mild to moderate transition is a critical stage with rapid immune landscape changes that may determine the trajectory of disease progression⁴³. CD8+ T cell activation is an important component of COVID-19 patients' immune response to the infection. By analyzing scRNA-seq data from 66 mild and 48 moderate COVID-19 patients, we examined the CD8+ T cell activation program in these patients and asked how it changes during the mild-to-moderate disease severity transition. The relatively large sample size of this dataset also allowed us to partition samples into non-overlapping subsets and systematically benchmark different methods' ability to detect XDE genes by evaluating the detection consistency between different sample subsets.

Finally, in the first paragraph of the **Discussion** section, we added:

variability in the multi-sample differential pseudotime analysis. In order to benchmark this new method, we applied it to both simulated and real data. We note that the analyses of three real datasets (HCA-BM, COVID, TB) mainly serve the purpose of illustrating and evaluating Lamian, and that making new biological discoveries *per se* is not the focus of this study. The orthogonal information (e.g. sex chromosome genes) and large sample size (e.g. COVID and TB data) available in these data make it possible to objectively and robustly compare different methods by quantifying the overlap with orthogonal information or between independent data partitions. Our results in these simulated and real data show that the new solution provided by Lamian substantially outperforms other existing methods to prioritize true discoveries and filter out false discoveries that are not generalizable to new samples.

Last but not least, we agree with the reviewer that it is important to take efforts to understand relevant biology to make a real impact with computational tools. It is indeed always our goal as computational biologists to make a real impact with computational tools, and therefore we will never stop learning.

3. “It will be interesting to check whether this low detection rate is biologically relevant or whether it is purely driven by uneven capture of cell number for the myeloid lineage.”

My point on this question is that if the detection rate of the myeloid lineage is low in the simulation study, it may imply the differentiation into myeloid is less frequent / robust comparing to other lineages. Imagine this situation: for a progenitor population to differentiate into branch A and branch B, if branch A is more robust and dominates the cell population, the downsampling simulation may lead to lower detection rate of branch B simply because the cells in branch B is much less and uniformly downsampled may rarely capture branch B and thus you cannot detect branch B. I hope this makes sense.

We thank the reviewer for clarifying the question. The reviewer’s comment is based on assuming “uniformly downsample”. We are sorry if we have not made it crystal clear in the old manuscript that the downsampling in our simulations is not uniform for all branches. Instead, we only downsampled cells in the myeloid lineage but did not change the other two lineages. All cells in the other two lineages are retained. Thus, the decreasing detection rate of the myeloid lineage when the cell number in that lineage decreases is **a pattern one would expect based on the simulation design** – if we throw away most of the cells in the myeloid branch to create a simulation dataset while keeping the other two branches unchanged, we would expect that the detection rate of the myeloid branch will be low in that simulation dataset. We use this pattern for method benchmark. If a method for characterizing branch detection uncertainty really works, it should show such a pattern in this simulation. If a method fails to show this decreasing detection rate, it will suggest that the method lacks the ability to accurately characterize branch detection uncertainty.

In summary, since the cell number is artificially reduced in the myeloid lineage while the other two lineages are not affected, the low detection rate in the myeloid lineage in this simulation does not have a real biological interpretation. One cannot interpret it as “differentiation into myeloid is less frequent / robust comparing to other lineages”.

To help avoid this confusion, we have now added the following clarifications:

2.2.3 Lamian accurately characterizes stability and differences in tree topology

To demonstrate the validity of Lamian’s topology stability and differential topology analysis, we performed two sets of simulations. In Simulation 1, we subsampled cells in the myeloid lineage in the HCA-BM data to reduce the myeloid cell number **while retaining all cells in the erythroid and lymphoid lineages** (Fig. 2e,f). As expected, decreasing the number of cells decreased the detection rate for the myeloid branch (Fig. 2g). For example, when 80% cells **in the myeloid lineage** were reduced, the detection rate dropped to 0.106 (Fig. 2c,g). Hence, the detection rate provides a reasonable measure for quantifying the certainty (or uncertainty) conveyed by the data about the presence of a branch.

In Simulation 2, we reduced the number of cells in the myeloid lineage in four out of the eight samples **while retaining all cells in the other two lineages** (Fig. 2f). As the number of cells decreased, the detection rate of the myeloid branch again

Reviewer #3 (Remarks to the Author: Overall significance):

This revision has addressed all of my comments. They added more results from different data integration strategies, improved Lamina’s computational efficiency for large datasets, added new comparisons with competing method conditions, and deposited codes to GitHub. We have run through all the functions of the R package "Lamian" on the provided cleaned dataset. They provide a very clear document for the package. We are able to replicate all the results following the document. Lamina is very useful in our ongoing single cell projects with multiple samples and conditions. This comprehensive statistical package will fill the gap in understanding dynamic changes of gene expression in population-based single cell studies.

We greatly appreciate the reviewer’s positive comments.

Reviewer #4 (Remarks to the Author: Overall significance):

This revision has largely improved the quality of the manuscript. I appreciated the addition of a new case study, and in depth benchmarking with other methods, which are convincing. Lamian provides a new approach for pseudotime analysis that will complement (or surpass) other methods in the field.

My comments below are fairly minor to improve the presentation of the manuscript following this revision.

We greatly appreciate the reviewer's positive comments.

Reviewer #4 (Remarks to the Author: Strength of the claims):

1 - Now that a study demonstrating that Lamian can adjust for batch effects, you should consider adding this information in the abstract, as I think it will be of interest to many researchers.

We thank the reviewer for this great suggestion. We have now added this information in the abstract as follows:

Pseudotime analysis with single-cell RNA-sequencing (scRNA-seq) data has been widely used to study dynamic gene regulatory programs along continuous biological processes. While many methods have been developed to infer the pseudotemporal trajectories of cells within a biological sample, it remains a challenge to compare pseudotemporal patterns with multiple samples (or replicates) across different experimental conditions. *Lamian* is a comprehensive and statistically-rigorous computational framework for differential multi-sample pseudotime analysis. It can be used to identify changes in a biological process associated with sample covariates, such as different biological conditions **while adjusting for batch effects**, and to detect changes in gene expression, cell density, and topology of a pseudotemporal trajectory. Unlike existing methods that ignore sample variability, *Lamian* draws statistical inference after accounting for cross-sample variability and hence substantially reduces sample-specific false discoveries that are not generalizable to new samples. Using both real scRNA-seq and simulation data, including an analysis of differential immune response programs between COVID-19 patients with different disease severity levels, we demonstrate the advantages of *Lamian* in decoding cellular gene expression programs in continuous biological processes.

2 - The added description and justification on harmonisation (2.1) seems a bit out of place ('the purpose ...'). Consider moving this to the Methods section.

We appreciate the reviewer's suggestion. We have now moved these sentences to the Methods section 4.2. Data harmonization and processing as follows:

4.2 Data harmonization and preprocessing

Before *Lamian* analysis, one needs to first harmonize data from different samples. The purpose of harmonization is to match cells of the same type across samples so that the same type of cells can be compared across samples. As such, it removes both biological differences of interest (e.g. the same cell type can have differential expression between two sample conditions which is removed by harmonization) and unwanted technical differences (e.g. batch effects) among samples. In downstream analyses, since one is interested in biological variation across samples and conditions, *Lamian* will use the original normalized gene expression values instead of the harmonization-corrected expression values, and it will use a regression framework to remove unwanted technical variations such as batch effects but retain biological differences across samples. In this study, we used Seurat(v.3.2.1)³² to integrate (or harmonize) multiple samples in each dataset. For differential expression (DE) analysis, SAVER was used to impute gene expression values to address the drop-outs in the data. All DE methods used imputed values except tradeSeq and condiments since they require count values as inputs. Principal Component Analysis (PCA) and Uniform Manifold Approximation and Projection (UMAP)⁵³ were used for visualization, and they were both run using default settings.

3 - The purpose of the first case study (HCA-BM) is quite clear as it illustrates all modules of Lamian and provide some benchmarking (2.2 - 2.4). For the other case studies, the purpose is less clear.

Consider adding a short introduction to both to explain what are the characteristics of these studies (eg. COVID19: detecting differences between covariates; TB: accounting for batch effects).

This is a great suggestion. We have now added the following sentence (pink) at the beginning of subsection 2.5 about COVID-19:

2.5 Lamian analysis of COVID-19 scRNA-seq data identifies differential CD8 T cell transcriptional programs during a critical stage of disease severity transition

To further demonstrate and evaluate Lamian's ability to detect differences associated with sample covariates, we applied Lamian to a COVID-19 peripheral blood mononuclear cell (PBMC) 10x Genomics scRNA-seq dataset obtained from a recent study⁴³. The COVID-19 disease severity of a patient may progress from mild to moderate to severe. It was reported that the mild to moderate transition is a critical stage with rapid immune landscape changes that may determine the trajectory of disease progression⁴³. CD8+ T cell activation is an important component of COVID-19 patients' immune response to the infection. By analyzing scRNA-seq data from 66 mild and 48 moderate COVID-19 patients, we examined the CD8+ T cell activation program in these patients and asked how it changes during the mild-to-moderate disease severity transition. The relatively large sample size of this dataset also allowed us to partition samples into non-overlapping subsets and systematically benchmark different methods' ability to detect XDE genes by evaluating the detection consistency between different sample subsets.

We also have added the following sentence at the beginning of subsection 2.6 about TB:

2.6 Lamian analysis of tuberculosis data

To demonstrate and evaluate Lamian's ability to analyze large datasets and detect differences associated with sample covariates while adjusting for potential confounders such as batch effects, we analyzed an atlas-size dataset consisting of 337,191 memory T cells from 184 donors (100 females and 84 males) in a tuberculosis (TB) progression cohort⁵⁰ (Fig. 6a,b). This dataset has recently been used for demonstrating co-varying neighborhood analysis and biologically meaningful cell abundance differences between males and females were reported along the second principal component of the co-varying neighborhood abundance matrix (NAM-PC2)⁵¹. Consistent with that study, we provided NAM-PC2 as cells' pseudotime and conducted differential analysis (Fig. 6a). Samples in this dataset are profiled in multiple batches (Fig. 6b). We added batch indicators to the Lamian regression model to account for batch effects.

4 - The discussion meanders a bit. Consider tightening the text (e.g 'Consequently, it is not always clear what it means if one says that "a gene is differential along a trajectory lineage" since the lineage does not always exist in all trees.'). The computational part should be moved to the results section, with a table stating # cells, samples, compute time) to ease of reading. Generally, I did find that the added text did not flow very well with the rest of the original version.

We appreciate the reviewer's suggestion. We have now tightened the following sentences

"Consequently, it is not always clear what it means if one says that "a gene is differential along a trajectory lineage" since the lineage does not always exist in all trees. It is also unrealistic to enumerate and report all branches that occurred in bootstrapped trees and differential genes for each bootstrapped branch. ... This sequential procedure avoids the complication in comparing different trees and makes it easier to summarize and report analysis results to the end users."

and revised them as follows (see texts in pink):

difficult to implement and make the results difficult to summarize and report. This is because trees reconstructed from different bootstrap samples can have different topologies due to the randomness. A branch that appears in one tree may not exist in another tree, and often it is unclear how one should align branches of different trees. It is unrealistic to enumerate all branches that occurred in bootstrapped trees, and the meaning of differential expression along a branch can be unclear if the branch does not always exist. For this reason, Lamian separated the evaluation of uncertainties of the inferred pseudotemporal trajectory (i.e. the construction of minimum spanning tree) and the evaluation of uncertainties of gene expression using a sequential "conditional" procedure. In other words, our module 1 evaluates the uncertainty of pseudotime (MST) construction. Next, conditional on a tree lineage and conditional on the corresponding inferred pseudotime, modules 3 and 4 perform differential analyses using bootstrap sampling to account for the cell-level uncertainty, followed by modelling sample- and cell-level variability to account for gene expression variability and uncertainty. This sequential procedure avoids the complication of comparing different trees, making it easier for summarizing the analysis results to end users. Thus, while it may be imperfect, it

We have also moved the computational part to Results section 2.7 “**Computational efficiency**”. We have now added a supplementary table Table S2 showing the #cell, #sample, time, and memory, as suggested by the reviewer.

			Computational Time (Hour)					
	NumberOfSamples	NumberOfCells	condiments	Lamian.chisq	Lamian.pm	monocle2TrajTestCorr	phenopath	tradeSeq
HCA.Simu	8	13k	2.961	0.228	2.344	0.1496	14.1616	0.2448
HCA	8	13k	4.0775	0.233	2.70877778	0.072333333	14.17666667	0.309333333
COVID	161	56k	10.497	1.529	27.12725	0.187	NA	0.578
TB	184	337k	NA	8.926	62.605	10.275	NA	NA
			Memory (GB)					
	NumberOfSamples	NumberOfCells	condiments	Lamian.chisq	Lamian.pm	monocle2TrajTestCorr	phenopath	tradeSeq
HCA.Simu	8	13k	28.2649872	5.406388	91.9243463	4.8552312	16.2346992	132.3098592
HCA	8	13k	28.103318	5.692792	122.82056	7.924004	17.42598667	170.492868
COVID	161	56k	77.20384	4.002128	279.96936	28.895608	NA	318.775632
TB	184	337k	NA	4.89568	243.04102	43.28332	NA	NA

Please note that in the previous version of the manuscript, we have displayed the computational time and memory in **Supplementary Figure S18**. In this revision, we have retained that figure to facilitate visual comparisons.

5 - I do think that the authors should mention the origin of the name of the method either by text or picture (graphical abstract) in the manuscript!

We thank the reviewer for the suggestion. We have now added the following texts to the Introduction.

To address **these gaps**, we introduce a comprehensive and integrative statistical framework, referred to as Lamian, for differential multi-sample pseudotime analysis. **Lamian** is named after a traditional Chinese hand-pulled noodle. The name is chosen based on the similarity between the process of making Lamian noodles and our statistical model in which multi-sample single-cell data are described using multiple smooth noodle-like functional curves (Figure S1). Given scRNA-seq data from

We have also added the following figure to the **Supplementary Figure S1**.

Figure S1. The statistical model is named as Lamian, a traditional Chinese hand-pulled noodle, based on the similarity between the model and the process of making Lamian. To make the Lamian noodles, one first makes a firm dough using wheat flour, and then pulls many thinner noodles from the dough. Finally, one coats the noodles with extra flour to prevent the noodles from sticking to each other. In the method Lamian, the population-level pattern (the thick orange curve) of gene expression dynamics in all samples resembles the dough. The sample-level gene expression dynamics is modeled as the population-level pattern plus some sample-level random variation, resulting in a curve for each sample (the thin curves). The sample-level curves resemble the thinner noodles coming from the dough. Finally, the observed cell-level gene expression values are modeled as the sample-level curve plus some cell-level random variation. Here the cells resemble the flour.

Minor:

Introduction: 'Recently, PseudotimeDE is developed' -> has been proposed ?

p2 'In summaryvariability'. This information could be more impactful if moved before the description of other methods ('Although'). the added text is a bit clunky.

We have changed "is developed" to "has been proposed", as follows:

Pseudotime inference itself also has uncertainties. Recently, PseudotimeDE³⁰ has been proposed to account for pseudotime reconstruction uncertainties in single-sample pseudotime analysis via subsampling cells and permuting pseudotime.

The following sentences (pink color) have been moved before the description of other methods ("Although..."):

However, there currently does not exist a comprehensive integrative framework that identifies all three types of changes in pseudotemporal trajectories (topology, cell density, and gene expression) across experimental conditions with multiple samples per condition, while also accounting for sample-level variability.

Although there exist pseudotime analysis methods to detect changes in gene expression along pseudotime (e.g. Monocle²⁰⁻²², TSCAN²³, Slingshot²⁴), in cell abundance along pseudotime (e.g. milo²⁵, DAseq²⁶), and in trajectory lineages (e.g. tradeSeq²⁷), most methods do not investigate changes across conditions. Almost all methods ignore sample-to-sample

Section 2.1.3 makes it sound like the two tests come from different models (I think this is because one sentence starts with 'First', but no 'Second' follows).

We have added 'Second' to the following sentence:

along pseudotime ($H_1 : f(t) \neq c$). Here, TDE refers to *pseudotime differential expression*. **Second**, the XDE test evaluates for each gene whether the pseudotemporal activity $f(t)$ is associated with a sample-level covariate, such as whether $f(t)$ is different between healthy and disease samples. Here, XDE refers to *covariate X differential expression*. Currently, existing pseudotime

'seperated' appears twice (Fig 4 & 4.1.4)

We have proofread to make sure "separated" is correctly spelled.

Fig 6 caption: only mention once at the end of the caption 'Methods that cannot produce output within 1 week and 400 GB RAM are not shown'

We have now mentioned it only once in the Fig 6 caption.

Reviewer #4 (Remarks to the Author: Reproducibility):

I found that the GitHub repository for reproducing the analyses is quite obscure (https://github.com/Winnie09/trajectory_variability). As a user, I would not know where to start. Could you reorganise and rename the folders, and perhaps give a brief summary in the readMe file about what each case study is about (see my comments above regarding the aims of each case study).

We thank the reviewer for this great suggestion. We have now re-organized the folders and added a new README file for users to navigate the codes corresponding to each analysis. We will keep

updating the Github repository and the README file in response to users' inquires. Please see the screenshots as follows.

Introduction

Codes for generating the results in the paper: a statistical framework for differential pseudotime analysis in multiple single-cell RNA-seq

by Wenpin Hou, Zhicheng Ji, Zeyu Chen, E. John Wherry, Stephanie C. Hicks*, Hongkai Ji*

Folders

function: main R functions used in this repository.

h5func: hdf5-related R functions.

tree_variability: module 1 analysis.

hca_bone_marrow_data_analysis: HCA-BM single-cell RNA-seq data analysis. The purpose of this HCA-BM case study is to illustrate all modules of Lamian and provide some benchmarking (2.2 - 2.4).

hca_bone_marrow_data_integration_cutoff: integration results using two other cutoffs for HCA-BM data.

covid_data_analysis: COVID19 data analysis. The purpose of this TB case study is to illustrate that Lamian can identify differential CD8 T cell transcriptional programs during a critical stage of disease severity transition (2.5).

tb_data_analysis: TB (tuberculosis) single-cell RNA-seq data analysis. The purpose of this TB case study is to demonstrate and evaluate that Lamian can be applied to detect differences with respect to sample covariates while adjusting for batch effects (2.6).

harcohen_data_analysis: Harcohen data analysis. For back up purpose only. This data is not presented.

age_differential: identify age-associated genes.

cellcycle_gene_analysis: cellcycle gene analysis.

efficiency: computational efficiency.

Citation

A statistical framework for differential pseudotime analysis with multiple single-cell RNA-seq samples. Wenpin Hou, Zhicheng Ji, Zeyu Chen, E John Wherry, Stephanie C Hicks*, Hongkai Ji*. bioRxiv 2021.07.10.451910; doi: <https://doi.org/10.1101/2021.07.10.451910>.

This manuscript is now under revision in a peer-review journal.

Contact

Should you encounter any bugs or have any suggestions, please feel free to contact Wenpin Hou wh2526@cumc.columbia.edu, or open an issue on the Github page <https://github.com/Winnie09/Lamian/issues>.

© 2023 GitHub, Inc. Terms Privacy Security Status Docs Contact GitHub Pricing API Training

Reviewer Comments, Third Version:

Reviewer #1 (Remarks to the Author: Overall significance):

The authors have addressed most of my comments, so I would suggest acceptance. Although for point 2, the authors still decided to use the t-test mainly because it is easy to understand by biologists, which is still a compromise in my opinion.

Reviewer #2 (Remarks to the Author: Overall significance):

the author made efforts to address my comments, I don't have further comments

Reviewer #4 (Remarks to the Author: Overall significance):

The authors have addressed all my comments, and the GitHub page has been improved.

Author Rebuttal, Third Version

Reviewer reports:

Reviewer #1:

The authors have addressed most of my comments, so I would suggest acceptance. Although for point 2, the authors still decided to use the t-test mainly because it is easy to understand by biologists, which is still a compromise in my opinion.

We thank the reviewer for all the helpful comments and recommendation for acceptance. For the point 2 regarding the t-test, we note that the statement that “the authors still decided to use the t-test” is inaccurate. Indeed, in our previous revision [REDACTED] we have already replaced the original two-sample t-test by a new binomial logistic regression function which overcomes the limitations of t-test. The binomial logistic regression considers the fact that the cell proportions are bounded between 0 and 1 rather than unbounded as in t-test. The binomial logistic regression also keeps the interpretation of the results relatively simple.

Besides the binomial logistic regression, we also provided the multinomial logistic regression function in our previous revision for rigorous evaluation of differential branch cell proportions. Multinomial logistic regression considers the facts that not only the cell proportions are bounded between 0 and 1 but also the proportions of different branches are not independent. Thus, this function fully addresses the limitations of t-test mentioned by the reviewer. A drawback of multinomial logistic regression is that interpretation of its results is more complicated. This was explained in detail in the last section of Supplementary Notes in our previous submission.

Importantly, our results in **Figure 2h** (unchanged from our previous revision) have also shown that both the binomial logistic regression (fitted for each branch separately) and multinomial regression (fitted for all branches jointly) can provide reasonable performance – they both controlled the type I error rate and had increasing power with increasing changes in branch cell proportions.

In Lamian, we decided to provide both of these functions so that users have the flexibility to choose the solution that is best suitable for their needs. If statistical rigor is the priority, users can use the multinomial logistic regression. If ease of interpretation is what they need, users can use the binomial logistic regression. With both these functions, we believe that this reviewer’s concern about our original two-sample t-test can be fully addressed by our current implementation of Lamian.

In this submission [REDACTED], we highlighted the fact that multinomial logistic regression is also provided in addition to binomial logistic regression, just in case the reviewer has missed it:

Module 2 of Lamian first identifies variation in tree topology across samples and then assesses if there are differential topological changes associated with sample covariates (Fig. 1b). For each sample, Lamian calculates the proportion of cells in each tree branch, referred to as *branch cell proportion*. Because a zero or low proportion can reflect absence or depletion of a branch, changes in tree topology can be described using branch cell proportion changes. With multiple samples, Lamian characterizes the cross-sample variation of each branch by estimating the variance of the branch cell proportion across samples. Furthermore, regression models can be fit to test whether the branch cell proportion is associated with sample covariates. To facilitate convenient exploration of each individual branch, one can use a binomial logistic regression to evaluate covariate-associated branch cell proportion changes for each branch separately. **Alternatively, users can also use a multinomial logistic regression to analyze covariate-associated changes of cell proportion ratios between branches by considering all branches jointly (Supplementary Notes).** These regression-based methods allow one to identify tree topology changes between different conditions, for example in a case-control cohort, accounting for sample-level variability. They are functions not provided by methods such as PhenoPath, condiments and PseudotimeDE.

Details about the binomial logistic regression and multinomial logistic regression and their pros and cons are provided in Methods section “**Tree variability across samples and differential topology analysis**” and **Supplementary Notes “Options for analyzing branch cell proportion changes**”, which remain the same as the previous submission [REDACTED].

Reviewer #2:

the author made efforts to address my comments, I don't have further comments

We thank the reviewer for all the helpful comments and the positive feedback.

Reviewer #3:

The authors have addressed all my comments, and the GitHub page has been improved.

We thank the reviewer for all the helpful comments and the positive feedback.